# Towards Accurate Validation in Deep Clustering through Unified Embedding Learning

## Abstract

Deep clustering integrates deep neural networks into the clustering process, simultaneously learning embedding spaces and cluster assignments. However, significant challenges remain in evaluating and comparing the performance of different deep clustering algorithms—or even different training runs of the same algorithm. First, evaluating the clustering results from different models in the same high-dimensional input space is impractical due to the curse of dimensionality. Second, comparing the clustering results of different models in their respective learned embedding spaces introduces discrepancies, as existing validation measures are designed for comparisons within the same feature space. To address these issues, we propose a novel evaluation framework that learns a unified embedding space. This approach aligns different embedding spaces into a common space, enabling accurate comparison of clustering results across different models and training runs. Extensive experiments demonstrate the effectiveness of our framework, showing improved consistency and reliability in evaluating deep clustering performance.

## 1 Introduction

Deep clustering methods (Min et al., 2018; Yang et al., 2016; Ghasedi Dizaji et al., 2017) have seen extensive development in enhancing the scalability of traditional clustering techniques. By transforming high-dimensional data into a lower-dimensional latent feature space (also known as the embedding space) using deep neural networks, these methods make the clustering process more efficient and manageable. Most deep clustering approaches optimize a clustering objective based on the learned embedding space, addressing the challenges associated with high-dimensional data. Despite these advancements, accurately evaluating and validating the model performance remains a significant challenge, particularly due to the absence of labels. Proper evaluation is crucial for both model training and comparison, yet it remains an under-explored aspect of deep clustering research.

Clustering results are often assessed using two main types of validation approaches: *external measures* and *internal measures* (Liu et al., 2010). External measures are used when true labels are available, allowing direct comparison between predicted clusters and actual labels. Examples include normalized mutual information (NMI) and clustering accuracy (ACC), which respectively measure the similarity between cluster assignments and the proportion of correctly matched labels. However, their reliance on true labels limits their use in many cases. Internal measures (Rousseeuw, 1987; Caliński & Harabasz, 1974; Davies & Bouldin, 1979; Sarle, 1983; Dunn, 1974; Hubert & Levin, 1976; Halkidi & Vazirgiannis, 2001; 2008), on the other hand, evaluate clustering based solely on the data's inherent characteristics, with metrics like the Silhouette score, Calinski-Harabasz index, and Davies-Bouldin index serving as key tools when labels are unavailable.

Given the input data $\mathbf{X}$ and an estimated partition $\rho$, the internal validation score, denoted $\pi(\rho|\mathbf{X})$, is traditionally used to assess how well the partition $\rho$ fits the structure of the data $\mathbf{X}$. In many deep clustering tasks, such as image clustering, the high dimensionality of $\mathbf{X}$ makes direct calculation of $\pi(\rho|\mathbf{X})$ in the original data space (referred to as raw space) challenging, where distances lose meaning and computation becomes costly. Since deep clustering algorithms generate lower-dimensional embedded data $\mathbf{Z} := g(\mathbf{X})$ via an encoder $g$ and perform clustering in this embedded space, many studies (Wang et al., 2018; 2021; Huang et al., 2021a;b; Ronen et al., 2022; Hadipour et al., 2022; Li et al., 2023) use $\pi(\rho|\mathbf{Z})$ as a validation criterion based on the coupled embedded data (see Figure 1 for more details about the difference between raw space and coupled space-based evaluation).

However, using coupled embeddings, now a mainstream approach for validation in deep clustering tasks, faces the issue that the embedding data $\mathbf{Z}$ and the corresponding embedding space can vary between different clustering algorithms or even within the same algorithm when using different hyperparameters or initializations. This variability creates a discrepancy because internal validation measures typically assume a consistent feature space, thereby undermining the accuracy and reliability of clustering assessment and comparison.

In this work, we start by providing a theoretical analysis to identify and discuss the pitfalls of two widely adopted approaches for applying internal validation measures in deep clustering evaluation. First, we analyze how the curse of dimensionality diminishes the effectiveness of internal validation when applied directly to high-dimensional raw data. Second, we demonstrate that comparing internal measure scores calculated on coupled embedding spaces can lead to inconsistent evaluation of clustering results. To address these challenges, we argue that the ideal solution would involve comparison within a single, optimal low-dimensional embedding space that accurately preserves the similarity and distance relationships among data points. This inspires us to propose a novel approach that estimates an optimal space by aligning and unifying the embedding data from multiple embedding spaces generated from deep clustering results into a common, consistent representation. Our method involves developing an algorithm based on unified embedding learning to achieve this unification. With the unified space, internal measure scores can be computed to reliably compare clustering results. Empirical studies demonstrate that our framework significantly improves the accuracy of internal validation in deep clustering, offering a more consistent and precise evaluation of clustering outcomes.

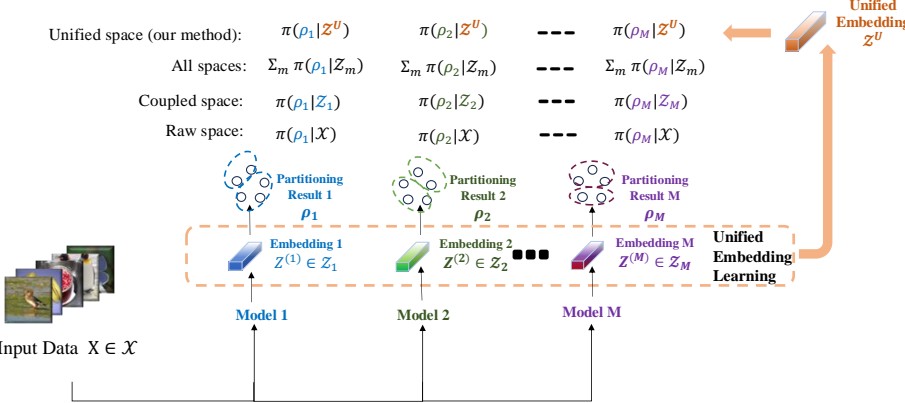

Figure 1: Comparison of four internal validation approaches based on different choices of evaluated spaces. $\pi(\rho|\mathcal{X})$ represents the internal measure score of the estimated partition $\rho$ on the data $\mathbf{X}$ in a space $\mathcal{X}$. "All spaces" refers to a baseline that uses the simple average of scores across all available embedding spaces, represented as $\sum_{m=1}^{M} \pi(\rho|\mathcal{Z}_m)$, for evaluation.

## 2 PITFALLS OF INTERNAL VALIDATION

Let $\mathbf{X} = \{\mathbf{x}_1, \cdots, \mathbf{x}_n\}$ represent a set of $n$ observations from a high-dimensional feature space $\mathcal{X}$ and $Y = \{y_1, \cdots, y_n\}$ denote the corresponding unknown true labels. Clustering techniques aim to find a mapping $\phi : \mathbf{X} \to \{1, \ldots, K\}$ that partitions the data into $K$ clusters. Denote $C_k := \{i \in \{1, ..., n\} | \phi(\mathbf{x}_i) = k\}$ as the index set for the $k$-th cluster. Consequently, $\rho := \{C_1, \ldots, C_K\}$ forms a partition of the index set $\{1, \ldots, n\}$. As we mentioned in Section 1, the internal measure of the clustering outcome $\rho$ based on the original data $\mathbf{X}$ is denoted as $\pi(\rho|\mathbf{X})$. In this section, we instead use the notation $\pi(\phi|\mathcal{X})$ to emphasize that the partition $\rho$ is generated by the algorithm $\phi$, and the measure is evaluated on the feature space $\mathcal{X}$. In addition to the estimated partition $\rho$, a deep clustering algorithm also converts the data $\mathbf{X}$ into lower-dimensional representations denoted as $\mathbf{Z} := \{\mathbf{z}_1, \cdots, \mathbf{z}_n\}$ in the low-dimensional embedding space $\mathcal{Z}$. Thus, $\pi(\phi|\mathcal{Z})$ denotes the internal measure of the partition generated by $\phi$ in the embedding space $\mathcal{Z}$.

**Theorem 1.** *[Distance Meaningless in High Dimensional Spaces (Beyer et al., 1999)] Let* $\{X_1, ..., X_n \in \mathbb{R}^p\}$ *be $n$ random points and $X_0$ be a random query point that is independent*

*from $\{X_1, ..., X_n\}$. Let $f$ be the probability density function of any fixed distribution on $\mathbb{R}$. For any distance function $d$, define $d_{\max} = \max_{i \in \{1,...,n\}} d(X_i, X_0)$ and $d_{\min} = \min_{i \in \{1,...,n\}} d(X_i, X_0)$. Given a fixed $n$, for any $\epsilon > 0$, we have $\lim_{p \to \infty} \mathbb{P}(\frac{d_{\max}}{d_{\min}} \leq 1 + \epsilon) = 1$, where the expectation is taken over the product distribution $f \times \cdots \times f$.*

Theorem 1 suggests that, as the dimensionality increases, the pairwise distance between data points in the input space $\mathcal{X}$ becomes indiscernible. Thus, any distance-based measure is unreliable and even misleading because of the curse of dimensionality. Since nearly all commonly used internal measures are based on distance calculations, this is particularly relevant when applying these measures (e.g., Silhouette score, Calinski-Harabasz index, and Davies-Bouldin index) in deep clustering evaluations, where the input data $\mathbf{X}$ often exhibits extremely high dimension. In such cases, relying on $\pi(\phi|\mathcal{X})$ can result in failed evaluations.

A widely adopted alternative in deep clustering evaluation is to compute internal measures in the lower-dimensional embedding space $\mathcal{Z}$, where distances more accurately reflect data similarity. However, unlike $\mathcal{X}$, the embedding space $\mathcal{Z}$ is influenced by the mapping function $\phi$. Comparing partitioning results $\rho$ based on their coupled embedding spaces (i.e., comparing $\pi(\phi_m|\mathcal{Z}_m)$) violates the assumption of internal measures that data should lie in the same feature space, leading to potentially inaccurate conclusions.

Recall that $\mathcal{Z}$ represents the embedding space where the input data $\mathbf{X}$ is transformed into the embedding data $\mathbf{Z} := \{\mathbf{z}_1, \cdots, \mathbf{z}_n\}$. Let $S_{i,j}$ be the similarity between $\mathbf{z}_i$ and $\mathbf{z}_j$ for any $i, j \in \{1, ..., n\}$, which satisfies $S_{i,j} \geq 0$, $S_{i,i} \geq 0$ and $\sum_i S_{i,j} = 1 \geq 0$.

**Definition 2.1.** We call a space $\mathcal{Z}$ an *informative space* for the data $\mathbf{X} := \{\mathbf{x}_1, \cdots \mathbf{x}_n\}$ if its corresponding similarity matrix $S$ satisfies that $S_{i,j_1} > S_{i,j_2}$ for any $i, j_1, j_2 \in \{1, ..., n\}$ where $y_i = y_{j_1}$ and $y_i \neq y_{j_2}$.

**Theorem 2.2.** *For a data $\mathbf{X} := \{\mathbf{x}_1, \cdots \mathbf{x}_n\}$, consider two informative spaces $\mathcal{Z}_1, \mathcal{Z}_2$. Assume that the partition $\phi_1(X)$ is as good as $\phi_2(X)$ in the sense that $\mathbb{P}(\pi(\phi_1(X)|\mathcal{Z}) \geq \pi(\phi_2(X)|\mathcal{Z})) \to 1$ as $n \to \infty$ for any informative space $\mathcal{Z}$. Then $\mathbb{P}(\pi(\phi_1(X)|\mathcal{Z}_1) \geq \pi(\phi_2(X)|\mathcal{Z}_2))$ does not always converge to 1.*

Theorem 2.2 indicates that comparing internal measure scores calculated on coupled embedding spaces does not ensure consistent evaluation of clustering results. This conclusion is evident in practical scenarios. For example, one deep clustering model may produce clusters that are more widely separated on the embedding space but have some misclassifications at the cluster boundaries, while another model might generate tighter clusters with perfect classification. Despite the boundary inaccuracies, the first model could still obtain a higher score from an internal measure like the Silhouette score, which emphasizes cluster separation. Theorem 2.2 underscores the necessity of a low-dimensional space that preserves the similarity structure between data points for reliable internal validation, while also highlighting the importance of a unified or common space for such validation. These insights drive our pursuit of a unified low-dimensional embedding space that effectively maintains similarity relationships among data points for internal validation.

## 3 UNIFIED EMBEDDING LEARNING

Given $M$ clustering results, our goal is to construct the unified embedding data, denoted as $\mathbf{Z}^u$, that optimally preserves the similarity structure of the original data by integrating embeddings $\{\mathbf{Z}^{(m)}\}_{1 \leq m \leq M}$ from these results. Many techniques have been developed for unified embedding learning in tasks such as multi-view clustering (Wang et al., 2019; Nie et al.; Zhu et al., 2018), multilingual alignment (Duong et al., 2017), and knowledge integration (Hwang & Sigal, 2014), aiming to align and integrate embeddings from diverse data sources. A common approach to achieving the unification of embeddings is by learning a common similarity (or affinity) matrix across multiple sources. To meet our objective, we first compute a similarity matrix $S^{(m)}$ for each embedding $\mathbf{Z}^{(m)}$ and then learn a unified similarity matrix by combining the individual $S^{(m)}$ matrices. Finally, we use an optimization approach akin to that used in stochastic neighbor embedding (Hinton & Roweis, 2002; Van der Maaten & Hinton, 2008) to estimate the low-dimensional embeddings $\mathbf{Z}^u$ in the unified space. The detailed steps are outlined as follows.

**S1: Develop a Unified Similarity Matrix**  Given any embedding space $\mathcal{Z}$ with embedded data $\mathbf{Z} := \{\mathbf{z}_1, \cdots, \mathbf{z}_n\}$, we calculate the similarity between $\mathbf{z}_i$ and $\mathbf{z}_j$ as

$$s_{i,j} = \frac{\exp(-\|\mathbf{z}_i - \mathbf{z}_j\|^2 / 2\sigma_i^2)}{\sum_{k \neq i} \exp(-\|\mathbf{z}_i - \mathbf{z}_k\|^2 / 2\sigma_i^2)}, \forall i \neq j \in \{1, ..., n\}. \tag{1}$$

The parameter $\sigma_i$ is a variance item and controls the spread of similarity around each data point. The tuning of $\sigma_i$ is further discussed in Section 4.3. Note that for all $i$, the sum of similarities satisfies $\sum_j s_{i,j} = 1$, with $s_{i,j} \geq 0$ for all $i, j$, and we set $s_{i,i} = 0$ to exclude self-similarity.

Denote $S^{(m)}$ as the similarity matrix defined in Eq. (1) that corresponds to the embedding $\mathbf{Z}^{(m)}$, $m = 1, \ldots, M$. We construct the similarity matrix for the unified embedding space by minimizing the following objective function:

$$\min_{U, \{w^{(m)}\}_{m=1}^M} \sum_{m=1}^M w^{(m)} \left\| U - S^{(m)} \right\|_F^2 \tag{2}$$

$$\text{subject to } \forall i, j, u_{ij} \geq 0, \mathbf{u}_i \mathbf{1}_N = 1, \tag{3}$$

where $w^{(m)}$ is the weight representing the importance of each embedding space, and $\mathbf{u}_i \in \mathbb{R}^{1 \times n}$ is the $i$-th row of $U$. The term $\|\cdot\|_F$ denotes the Frobenius norm of a matrix. A similar optimization problem has been explored in the context of multi-view clustering (Nie et al.; Zhu et al., 2018). Drawing inspiration from this work, we propose an iterative re-weighting approach, in which $w^{(m)}$ and $U$ are updated alternately. Differentiating Eq. (2) with respect to $U$ and setting the derivative to zero yields:

$$w^{(m)} = \frac{1}{2\|U - S^{(m)}\|_F} \tag{4}$$

This provides a method to update each $w^{(m)}$ while keeping $U$ fixed. Given each $w^{(m)}$, we can absorb them into the norm, allowing us to rewrite the optimization problem as: $\min_U \left\| U - \frac{\sum_{m=1}^M w^{(m)} S^{(m)}}{\sum_{m=1}^M w^{(m)}} \right\|_F^2$ subject to $\forall i, j, u_{ij} \geq 0, \mathbf{u}_i \mathbf{1}_N = 1$.

Recall that each $S^{(m)}$ is a non-negative matrix with row vectors that sum to one, i.e., $\mathbf{s}_i^{(m)} \mathbf{1}_N = 1$. Consequently, the solution to this optimization problem is straightforward and can be expressed as:

$$U = \sum_{m=1}^M \frac{w^{(m)}}{\sum_{m=1}^M w^{(m)}} S^{(m)} \tag{5}$$

The solution in Eq. (5) is a weighted combination of $S^{(1)}, ..., S^{(M)}$ ( hereafter referred to as the candidate similarity matrices), so we rewrite $U = \sum_{m=1}^M \mathsf{w}_m S^{(m)}$, where $\mathsf{w}_m = \frac{w^{(m)}}{\sum_{m=1}^M w^{(m)}}$.

The two steps can be iterated until the algorithm converges. Detailed update procedures are outlined in Algorithm 1. A convergence analysis of the algorithm is provided in Appendix B.

---

**Algorithm 1** Iterative re-weighted procedure

---

**Input:** Similarity matrices $\{S^{(1)}, S^{(2)}, \cdots, S^{(M)}\}$

1: Initialize each $w^{(m)} = \frac{1}{M}$
2: **repeat**
3:     Update $U$ according to Eq. (5)
4:     Update $w^{(m)}$ according to Eq. (4)
5: **until** the objective function converges

**Output:** $U$ **and** $\{w^{(m)}\}_{m=1}^M$

---

After obtaining $U$, we perform a normalization step to ensure that the resulting matrix is symmetric and that all entries sum to 1, thereby mitigating the issue of outliers (Van der Maaten & Hinton, 2008). Specifically, we define the normalized matrix $U^{\text{norm}}$ with the $(i, j)$-th entry $u_{ij}^{\text{norm}} = \frac{u_{ij} + u_{ji}}{2 \sum_{i,j} u_{ij}}$, where $u_{ij}$ is the $(i, j)$-th entry of $U$. The normalized value $u_{ij}^{\text{norm}}$ represents joint probabilities that reflect the similarities associated with the unified embedding space (Van der Maaten & Hinton, 2008).

**S2: Learn a Unified Embedding Space**   We follow the optimization strategy used in stochastic neighbor embedding methods(Hinton & Roweis, 2002; Van der Maaten & Hinton, 2008). For any given $\{\mathbf{z}_i^u\}_{i=1}^n$, we calculate the pairwise similarity

$$q_{ij} = \frac{(1 + \|\mathbf{z}_i^u - \mathbf{z}_j^u\|^2)^{-1}}{\sum_k \sum_{l \neq k}(1 + \|\mathbf{z}_k^u - \mathbf{z}_l^u\|^2)^{-1}}, \tag{6}$$

based on a Cauchy distribution, and we set $q_{ii}$ to zero. Then, we estimate the embedding $\mathbf{Z}^u$ by aligning the distributions $u_{ij}^{\text{norm}}$ with $q_{ij}$ in the sense that the Kullback-Leibler divergence between $u_{ij}^{\text{norm}}$ and $q_{ij}$ across all data points is minimized. In particular, we have the unified embedding vector $\hat{\mathbf{z}}_i^u$ as

$$(\hat{\mathbf{z}}_1^u, ..., \hat{\mathbf{z}}_n^u) = \arg\min_{\mathbf{z}_i^u} \sum_{i \neq j} u_{ij}^{\text{norm}} \log \frac{u_{ij}^{\text{norm}}}{q_{ij}} \tag{7}$$

The objective function in Eq. (7) is minimized using a gradient descent method with momentum. We then carry out internal evaluations on the unified embedding data $\hat{\mathbf{z}}_i^u$.

In our method, Step **S1** introduces a weighting scheme to derive this unified similarity matrix, which is crucial in determining the quality of the final learned unified embedding space. Given the unified similarity matrix, Step **S2** follows a well-established manifold learning technique, as consolidated in numerous previous works. We justify the use of linear aggregation for the similarity matrices $\{S^{(m)}\}_{m=1}^M$ (see Eq. (5)) in **S1** with the following theoretical analysis.

**Definition 3.1.**  Denote the value $a_{i,j_1,j_2} := \mathrm{I}_{S_{i,j_1} > S_{i,j_2}}$ where $\mathrm{I}(\cdot)$ is the indicator function, and $S_{t_1,t_2}$ is the similarity between $\mathbf{z}_{t_1}$ and $\mathbf{z}_{t_2}$ for any $t_1, t_2 \in \{1, ..., n\}$. We call the the set

$$A_{\mathbf{X}, \mathcal{Z}} := \{(i, j_1, j_2) : i, j_1, j_2 \in \{1, ..., n\}, y_i = y_{j_1}, y_i \neq y_{j_2}, S_{i,j_1} > S_{i,j_2}, S_{i,j_1} > S_{j_1,j_2}\}$$

the *similarity index set* of $\mathbf{X}$ generated by $\mathcal{Z}$. For notation convenience, we omit the subscript $\mathbf{X}$ and instead use $A_{\mathcal{Z}}$ when there is no confusion.

*Remark* 3.2.  Intuitively, in an informative space as in Definition 2.1, two points within the same cluster should have higher similarity than that of two points from two clusters. In general, the set $A_{\mathbf{X}, \mathcal{Z}}$ contains the triplets of points in $\mathcal{Z}$ where the similarity matrix aligns with that of an informative space.

To demonstrate the consistency of the unified similarity matrix (Eq. (5)), we start with the following definitions. For any set $A$, let $|A|$ denote its cardinality. For any two sets $A$ and $B$, denote $A \nabla B := \{x : x \in A \cup B, x \notin A \cap B\}$.

**Definition 3.3.**  Given any weighted similarity matrix $\sum_{m=1}^M \mathsf{w}_m S^{(m)}$, the weight $\mathbf{w} = \{\mathsf{w}_1, ..., \mathsf{w}_M\}$ is *weakly consistent* if there exists an informative space $\mathcal{Z}$ such that $\frac{\sum_m \mathsf{w}_m \cdot |A_{\mathcal{Z}^{(m)}} \nabla A_{\mathcal{Z}}|}{|A_{\mathcal{Z}}|} \xrightarrow{p} 0$ as $n \to \infty$, and $\mathbf{w}$ is *consistent* if $\sum_m \mathsf{w}_m \cdot |A_{\mathcal{Z}^{(m)}} \nabla A_{\mathcal{Z}}| \xrightarrow{p} 0$ as $n \to \infty$.

*Remark* 3.4.  The (weak) consistency of the weights makes sure that the weighting of the candidate similarity index sets is centered around the true similarity index set to some degree.

**Definition 3.5.**  Given the candidate embedding spaces $(\mathcal{Z}^{(1)}, ..., \mathcal{Z}^{(M)})$ and the weight $\mathbf{w}$, define the *importance* of the triplet $(i, j_1, j_2)$ as $v_{i,j_1,j_2} := \sum \mathsf{w}_m I((i, j_1, j_2) \in A_{\mathbf{X}, \mathcal{Z}^{(m)}})$ for $i, j_1, j_2 \in \{1, ..., n\}$.

*Remark* 3.6.  If the weights satisfy $\mathsf{w}_m \geq 0$ and $\mathsf{w}_1 + ... + \mathsf{w}_M = 1$, we have $0 \leq v_{i,j_1,j_2} \leq 1$. The importance $v_{i,j_1,j_2}$ is the accumulated weights of the candidate embedding spaces that contain the triplet, which reflects how much the unified embedding space agrees with an informative space on the triplet $(i, j_1, j_2)$. In an extreme example, if all the candidate similarity matrices agree with the truth on a triplet $(i, j_1, j_2)$, i.e., $(i, j_1, j_2) \in A_{\mathcal{Z}^{(m)}}$ for all $m$, then the unified similarity matrix will also agree with the truth on the triplet, and the importance $v_{i,j_1,j_2} = 1$ in this case.

Next, we show that a reasonable aggregating scheme enables us to build a unified (which takes the form of linear combination) similarity matrix that converges to the true similarity matrix.

**Theorem 3.7.**        *(a) Assume that the weight* $\mathbf{w}$ *is weakly consistent, we have*

$$\sum_{(i,j_1,j_2) \in A_{\mathcal{Z}}} v_{i,j_1,j_2} / |A_{\mathcal{Z}}| \xrightarrow{p} 1$$

*and*

$$\sum_{(i,j_1,j_2)\notin A_{\mathcal{Z}}} v_{i,j_1,j_2}/|A_{\mathcal{Z}}| \xrightarrow{p} 0$$

*as $n \to \infty$.*

(b) *Assume that the weighting $\mathbf{w}$ is consistent, we have*

$$\min_{(i,j_1,j_2)\in A_{\mathcal{Z}}} v_{i,j_1,j_2} \xrightarrow{p} 1$$

*and*

$$\max_{(i,j_1,j_2)\notin A_{\mathcal{Z}}} v_{i,j_1,j_2} \xrightarrow{p} 0$$

*as $n \to \infty$.*

Under the weak consistency (consistency, respectively) assumption of the weights, the sum of the importance of the true triplets will tend to the number of triplets in $A_{\mathcal{Z}}$ (1, respectively), while the sum of the importance of the triplets excluded by the true similarity index set converges to 0 (0, respectively). That is, the unified similarity matrix $\sum \mathbf{w}_m S^{(m)}$ (with a weakly consistent weight $\mathbf{w}$) agrees with some informative space $\mathcal{Z}$ on its similarity index set, thus correctly capturing the similarity structure of the input data. It is worth pointing out that if all the candidate embedding spaces are non-informative, we may not be able to find a good weight $\mathbf{w}$. Theorem 3.7 guarantees the consistency of our unified approach in estimating an informative space for reliable evaluation. In this regard, although applying other multi-view techniques (e.g., Zhu et al. (2018); Lin et al. (2021)) may also produce unified similarity matrices, we do not anticipate the same theoretical guarantee.

## 4 EMPIRICAL STUDY

### 4.1 STUDY DESIGN

**Evaluation Metrics**   To compare the performance of different validation approaches, we conducted experiments to assess their abilities to accurately rank partitioning results from different runs based on their similarity to ground truth labels. We use external measures as an oracle and evaluate the performance of different internal validation approaches by comparing their ranking consistency with these external measures. Specifically, we use two widely adopted external measures, normalized mutual information (NMI) and clustering accuracy (ACC), as described in Section 1 and defined in Appendix D. To quantify ranking consistency, we report Spearman's rank correlation coefficient ($r_s$) and Kendall's rank correlation coefficient ($\tau_B$), as defined in Appendix E.4. Our experiments include the performance of internal validation methods using three commonly applied measures: the Silhouette score, Calinski-Harabasz index, and Davies-Bouldin index, whose definitions can be found in Appendix C.

**Evaluated Deep Clustering Methods**   Deep clustering methods are generally divided into two main approaches (Min et al., 2018): autoencoder-based (Song et al., 2013; Yang et al., 2017; Ghasedi Dizaji et al., 2017; Vincent et al., 2008; Masci et al., 2011; Ronen et al., 2022) and clustering deep neural network (CDNN)-based (Yang et al., 2016; Ghasedi Dizaji et al., 2017; Caron et al., 2018; Wang et al., 2021). The primary distinction is that CDNN-based methods learn image clusters and embeddings without relying on an autoencoder. From these two categories, we selected two prominent methods: *DEPICT* (Ghasedi Dizaji et al., 2017)[1], representing the autoencoder-based approach, and *JULE* (Yang et al., 2016)[2], a leading CDNN-based method. DEPICT uses a multinomial logistic regression layer atop a convolutional autoencoder to map data into a embedding space, minimizing both clustering and reconstruction losses. JULE creates a recurrent framework that iteratively merges clusters through agglomerative clustering, optimizing a weighted triplet loss to jointly estimate cluster labels and embeddings. Further details on these two methods can be found in Appendix E.3.

---

[1]`https://github.com/herandy/DEPICT`
[2]`https://github.com/jwyang/JULE.torch`

Table 1: Rank consistency between the NMI scores and those generated by the evaluation regime using different spaces for hyperparameter tuning. The coefficients $r_s$ and $\tau_B$ represent the Spearman and Kendall rank correlation coefficients, respectively, used to measure this consistency. Empty cells indicate cases where results are unavailable. The best results are highlighted in bold.

| | USPS | | YTF | | FRGC | | MNIST-test | | CMU-PIE | | UMist | | COIL-20 | | COIL-100 | | Average | |
|---|---|---|---|---|---|---|---|---|---|---|---|---|---|---|---|---|---|---|
| | $r_s$ | $\tau_B$ | $r_s$ | $\tau_B$ | $r_s$ | $\tau_B$ | $r_s$ | $\tau_B$ | $r_s$ | $\tau_B$ | $r_s$ | $\tau_B$ | $r_s$ | $\tau_B$ | $r_s$ | $\tau_B$ | $r_s$ | $\tau_B$ |
| JULE: Calinski-Harabasz index | | | | | | | | | | | | | | | | | | |
| Raw space | 0.58 | 0.47 | 0.79 | 0.62 | -0.44 | -0.28 | 0.81 | 0.62 | -0.99 | -0.93 | -0.57 | -0.40 | -0.30 | -0.18 | 0.32 | 0.21 | 0.02 | 0.02 |
| Coupled space | 0.17 | 0.13 | 0.52 | 0.40 | -0.13 | -0.10 | 0.49 | 0.34 | -0.14 | -0.08 | **0.70** | **0.50** | 0.53 | 0.38 | 0.20 | 0.19 | 0.29 | 0.22 |
| All spaces | **0.85** | **0.68** | **0.91** | **0.79** | **0.31** | **0.23** | 0.82 | 0.67 | 0.90 | 0.77 | 0.63 | 0.44 | 0.62 | 0.47 | 0.91 | 0.76 | **0.75** | 0.60 |
| **Unified space** | 0.84 | 0.68 | 0.81 | 0.66 | 0.17 | 0.12 | **0.86** | **0.69** | **0.98** | **0.93** | 0.58 | 0.40 | **0.77** | **0.62** | **0.97** | **0.85** | 0.75 | **0.62** |
| JULE: Davies-Bouldin index | | | | | | | | | | | | | | | | | | |
| Raw space | -0.48 | -0.30 | -0.47 | -0.32 | -0.43 | -0.30 | -0.83 | -0.67 | -0.97 | -0.89 | -0.70 | -0.50 | -0.57 | -0.39 | -0.79 | -0.61 | -0.66 | -0.50 |
| Coupled space | -0.10 | -0.03 | -0.32 | -0.21 | -0.08 | -0.05 | -0.13 | -0.06 | 0.26 | 0.19 | **0.62** | **0.44** | **0.61** | **0.43** | 0.43 | 0.35 | 0.16 | 0.13 |
| All spaces | -0.26 | -0.13 | -0.46 | -0.34 | **0.12** | 0.08 | -0.15 | -0.06 | 0.92 | 0.79 | -0.35 | -0.24 | -0.24 | -0.16 | -0.46 | -0.35 | -0.11 | -0.05 |
| **Unified space** | **0.41** | **0.35** | **-0.09** | **-0.08** | 0.12 | **0.10** | **0.77** | **0.57** | **0.94** | **0.82** | -0.22 | -0.16 | 0.50 | 0.39 | **0.83** | **0.62** | **0.41** | **0.33** |
| JULE: Silhouette score | | | | | | | | | | | | | | | | | | |
| Raw space | 0.81 | 0.62 | 0.85 | 0.70 | 0.07 | 0.04 | 0.71 | 0.53 | 0.32 | 0.29 | -0.45 | -0.32 | -0.12 | -0.05 | 0.23 | 0.15 | 0.30 | 0.24 |
| Coupled space | 0.27 | 0.20 | 0.72 | 0.55 | 0.04 | 0.03 | 0.56 | 0.41 | 0.41 | 0.30 | **0.70** | **0.50** | **0.64** | **0.47** | 0.55 | 0.41 | 0.49 | 0.36 |
| All spaces | 0.70 | 0.57 | **0.90** | **0.77** | **0.41** | **0.28** | 0.78 | 0.63 | 0.95 | 0.84 | 0.64 | 0.43 | 0.26 | 0.16 | 0.71 | 0.54 | 0.67 | 0.53 |
| **Unified space** | **0.87** | **0.70** | 0.87 | 0.69 | 0.36 | 0.24 | **0.84** | **0.68** | **0.98** | **0.91** | 0.45 | 0.31 | 0.60 | 0.45 | **0.98** | **0.88** | **0.74** | **0.61** |
| DEPICT: Calinski-Harabasz index | | | | | | | | | | | | | | | | | | |
| Raw space | -0.05 | -0.10 | **0.73** | **0.62** | 0.43 | 0.25 | 0.43 | 0.35 | -0.95 | -0.83 | | | | | | | 0.12 | 0.06 |
| Coupled space | 0.76 | 0.57 | 0.44 | 0.26 | 0.76 | 0.57 | 0.89 | 0.72 | 0.49 | 0.44 | | | | | | | 0.67 | 0.51 |
| All spaces | **0.96** | **0.84** | 0.53 | 0.41 | **0.90** | **0.77** | **0.96** | **0.87** | 0.73 | 0.59 | | | | | | | 0.82 | 0.70 |
| **Unified space** | 0.95 | 0.84 | 0.65 | 0.52 | 0.89 | 0.75 | 0.96 | 0.84 | **0.95** | **0.80** | | | | | | | **0.88** | **0.75** |
| DEPICT: Davies-Bouldin index | | | | | | | | | | | | | | | | | | |
| Raw space | 0.05 | -0.10 | **0.63** | **0.48** | 0.48 | 0.32 | -0.01 | -0.03 | -0.14 | -0.18 | | | | | | | 0.20 | 0.10 |
| Coupled space | 0.81 | 0.59 | 0.45 | 0.31 | **0.90** | **0.74** | 0.89 | 0.72 | 0.63 | 0.59 | | | | | | | 0.73 | 0.59 |
| All spaces | **0.95** | **0.84** | 0.49 | 0.35 | 0.65 | 0.50 | 0.50 | 0.36 | 0.23 | 0.06 | | | | | | | 0.56 | 0.42 |
| **Unified space** | 0.92 | 0.78 | 0.60 | 0.42 | 0.81 | 0.66 | **0.92** | **0.80** | **0.99** | **0.92** | | | | | | | **0.85** | **0.72** |
| DEPICT: Silhouette score | | | | | | | | | | | | | | | | | | |
| Raw space | 0.50 | 0.36 | 0.76 | **0.61** | 0.57 | 0.41 | 0.74 | 0.59 | -0.21 | -0.12 | | | | | | | 0.47 | 0.37 |
| Coupled space | 0.73 | 0.50 | 0.47 | 0.36 | 0.79 | 0.65 | 0.86 | 0.69 | 0.59 | 0.52 | | | | | | | 0.69 | 0.54 |
| All spaces | 0.96 | 0.84 | 0.65 | 0.53 | 0.94 | 0.82 | **0.97** | **0.90** | 0.95 | 0.86 | | | | | | | 0.89 | 0.79 |
| **Unified space** | **0.98** | **0.91** | **0.78** | 0.59 | **0.95** | **0.84** | 0.97 | 0.90 | **0.97** | **0.88** | | | | | | | **0.93** | **0.82** |

**Datasets** We evaluated the methods DEPICT and JULE on the datasets referenced in their original papers, respectively. These datasets include two handwritten digit datasets: USPS and MNIST-test (LeCun et al., 1998), two multi-view object image datasets: COIL-20 and COIL-100 (Nene et al., 1996), and four face image datasets UMist, FRGC-v2.02, CMU-PIE, and YouTube-Face (YTF) (Graham & Allinson, 1998; Sim et al., 2002; Wolf et al., 2011). The datasets USPS, MNIST-test, FRGC, CMU-PIE, and YTF are common to both JULE and DEPICT studies, while COIL-20, COIL-100, and UMist are unique to JULE. Information on sample sizes, image dimensions, and the number of classes for each dataset can be found in Appendix E.1.

**Evaluated Tasks** Our study focuses on two critical aspects of deep clustering: (1) *hyperparameter tuning*, where different runs are generated using different hyperparameter configurations, and (2) *cluster number determination*, where runs are performed with varying numbers of clusters $K$. For the hyperparameter tuning experiments, in the JULE algorithm, we construct a search space of $6 \times 7 = 42$ combinations of the hyperparameter pair (learning rate, unfolding rate $\eta$). For the DEPICT algorithm, the search space consists of $6 \times 3 = 18$ combinations of the hyperparameter pair (learning rate, balancing parameter in the reconstruction loss function). For the cluster number determination experiments, we explore $K$ across 10 evenly spaced values that include the true $K$ or a nearby value. Specifically, we use $\{5, ..., 50\}$ for the MNIST-test, USPS, FRGC, UMist, YTF, and COIL-20 datasets; $\{10, ..., 100\}$ for CMU-PIE; and $\{20, ..., 200\}$ for COIL-100. For all experiments, if a training run fails, the clustering results are considered missing, and the corresponding configuration is excluded from the final evaluation.

## 4.2 COMPARISON OF DIFFERENT VALIDATION APPROACHES

We evaluated the performance of four validation approaches: raw space, coupled space, all spaces, and unified space (our method), as illustrated in Figure 1. Here "all spaces" refers to the straightforward idea of using a simple average of the scores across all available embedding spaces, i.e., $\sum_{m=1}^{M} \pi(\rho | \mathcal{Z}_m)$, as a score of the partition $\rho$. In running our method, the step of unifying the similarity matrix does not involve any hyperparameters that require tuning. For embedding optimization, we use the default hyperparameter values based on the implementation in Pedregosa et al. (2011).

We report the performance of all approaches based on the rank consistency between their generated scores and NMI scores for both tasks under evaluation (Tables 1 and 2). The results show that the

Table 2: Rank consistency between the NMI scores and those generated by the evaluation regime using different spaces for cluster number determination.

| | USPS | | YTF | | FRGC | | MNIST-test | | CMU-PIE | | UMist | | COIL-20 | | COIL-100 | | Average | |
|---|---|---|---|---|---|---|---|---|---|---|---|---|---|---|---|---|---|---|
| | $r_s$ | $\tau_B$ | $r_s$ | $\tau_B$ | $r_s$ | $\tau_B$ | $r_s$ | $\tau_B$ | $r_s$ | $\tau_B$ | $r_s$ | $\tau_B$ | $r_s$ | $\tau_B$ | $r_s$ | $\tau_B$ | $r_s$ | $\tau_B$ |
| JULE: Calinski-Harabasz index | | | | | | | | | | | | | | | | | | |
| Raw space | 0.44 | 0.56 | 0.95 | 0.89 | -0.93 | -0.83 | 0.43 | 0.51 | -0.37 | -0.24 | -0.33 | -0.24 | 0.74 | 0.64 | 0.53 | 0.47 | 0.18 | 0.22 |
| Coupled space | 0.65 | 0.64 | 0.1 | 0.06 | -0.93 | -0.83 | 0.64 | 0.6 | -0.03 | -0.02 | -0.13 | -0.07 | 0.76 | **0.71** | **0.74** | 0.56 | 0.22 | 0.21 |
| All spaces | 0.55 | 0.6 | 0.9 | 0.78 | -0.87 | -0.72 | 0.64 | 0.64 | 0.88 | 0.73 | -0.14 | -0.11 | 0.74 | 0.64 | 0.72 | **0.64** | 0.43 | 0.40 |
| **Unified space** | **0.98** | **0.91** | **1.0** | **1.0** | **0.83** | **0.67** | **0.96** | **0.87** | **0.95** | **0.87** | **0.43** | **0.24** | **0.83** | 0.71 | 0.61 | 0.51 | **0.82** | **0.72** |
| JULE: Davies-Bouldin index | | | | | | | | | | | | | | | | | | |
| Raw space | -0.27 | -0.29 | **0.92** | **0.78** | **0.87** | **0.72** | -0.46 | -0.42 | 0.72 | 0.47 | **0.19** | **0.16** | -0.88 | -0.79 | -0.92 | -0.82 | 0.02 | -0.02 |
| Coupled space | 0.54 | 0.38 | 0.15 | 0.17 | 0.85 | 0.67 | 0.43 | 0.29 | 0.78 | 0.56 | -0.08 | 0.02 | -0.26 | -0.14 | -0.9 | -0.78 | 0.19 | 0.15 |
| All spaces | **0.88** | **0.73** | 0.83 | 0.67 | 0.82 | 0.61 | **0.81** | **0.64** | 0.82 | 0.64 | 0.12 | 0.11 | -0.67 | -0.5 | -0.92 | -0.82 | 0.34 | 0.26 |
| **Unified space** | 0.47 | 0.33 | 0.55 | 0.39 | 0.18 | 0.17 | 0.54 | 0.47 | **0.92** | **0.82** | -0.28 | -0.2 | **0.43** | **0.43** | **0.9** | **0.78** | **0.46** | **0.40** |
| JULE: Silhouette score | | | | | | | | | | | | | | | | | | |
| Raw space | 0.56 | 0.47 | **1.0** | **1.0** | -0.18 | -0.17 | 0.61 | 0.47 | 0.55 | 0.38 | 0.19 | 0.16 | -0.41 | -0.36 | 0.39 | 0.2 | 0.34 | 0.27 |
| Coupled space | 0.85 | 0.73 | 0.33 | 0.28 | **0.72** | **0.61** | 0.88 | 0.69 | 0.96 | 0.87 | 0.07 | 0.16 | 0.55 | 0.43 | 0.44 | 0.29 | 0.60 | 0.51 |
| All spaces | **0.98** | **0.91** | 0.97 | 0.89 | 0.68 | 0.56 | **0.93** | **0.82** | 0.98 | 0.91 | 0.21 | 0.16 | 0.36 | 0.21 | 0.47 | 0.33 | 0.70 | 0.60 |
| **Unified space** | 0.84 | 0.69 | 0.87 | 0.72 | 0.63 | 0.5 | 0.92 | 0.78 | **0.99** | **0.96** | **0.42** | **0.29** | **0.93** | **0.86** | **0.95** | **0.87** | **0.82** | **0.71** |
| DEPICT: Calinski-Harabasz index | | | | | | | | | | | | | | | | | | |
| Raw space | 0.46 | 0.6 | -0.69 | -0.56 | -0.88 | -0.78 | 0.46 | 0.6 | -0.92 | -0.82 | | | | | | | -0.31 | -0.19 |
| Coupled space | 0.46 | 0.6 | -0.99 | -0.96 | -0.85 | -0.72 | 0.44 | 0.56 | -0.92 | -0.82 | | | | | | | -0.37 | -0.27 |
| All spaces | 0.46 | 0.6 | -0.98 | -0.91 | -0.85 | -0.72 | 0.46 | 0.6 | 0.44 | 0.56 | | | | | | | -0.09 | 0.03 |
| **Unified space** | **0.77** | **0.64** | **0.89** | **0.73** | **0.73** | **0.61** | **0.99** | **0.96** | **0.85** | **0.69** | | | | | | | **0.85** | **0.73** |
| DEPICT: Davies-Bouldin index | | | | | | | | | | | | | | | | | | |
| Raw space | -0.39 | -0.42 | **0.99** | **0.96** | 0.68 | 0.39 | -0.22 | -0.16 | **0.92** | **0.82** | | | | | | | 0.40 | 0.32 |
| Coupled space | 0.46 | 0.6 | -0.78 | -0.64 | -0.85 | -0.72 | 0.44 | 0.56 | -0.1 | 0.02 | | | | | | | -0.17 | -0.04 |
| All spaces | 0.7 | **0.64** | 0.88 | 0.73 | -0.13 | -0.17 | **0.94** | **0.82** | 0.92 | 0.82 | | | | | | | **0.66** | **0.57** |
| **Unified space** | **0.84** | 0.64 | 0.73 | 0.6 | 0.27 | 0.22 | 0.83 | 0.69 | 0.64 | 0.42 | | | | | | | 0.66 | 0.51 |
| DEPICT: Silhouette score | | | | | | | | | | | | | | | | | | |
| Raw space | -0.34 | -0.29 | **1.0** | **1.0** | 0.3 | 0.11 | 0.39 | 0.33 | -0.43 | -0.33 | | | | | | | 0.18 | 0.16 |
| Coupled space | 0.44 | 0.56 | -0.61 | -0.47 | -0.85 | -0.72 | 0.44 | 0.56 | -0.12 | -0.02 | | | | | | | -0.14 | -0.02 |
| All spaces | 0.74 | 0.64 | 0.98 | 0.91 | 0.07 | 0.06 | 0.81 | 0.73 | **0.99** | **0.96** | | | | | | | 0.72 | 0.66 |
| **Unified space** | **0.93** | **0.87** | 0.95 | 0.87 | **0.55** | **0.44** | **0.99** | **0.96** | 0.99 | 0.96 | | | | | | | **0.88** | **0.82** |

proposed approach achieves the highest average rank consistency compared to the three competing methods in most scenarios, underscoring its effectiveness. These findings indicate that scores computed from embedding spaces generally exhibit stronger rank correlations with external validation measures than scores derived from the raw space, aligning with Theorem 1. Furthermore, the comparison between the coupled space embeddings and the ensemble-based methods (using all spaces and unified space) confirms the validity of Theorem 2.2, as the ensemble scores demonstrate significantly higher rank correlations with external measures. Our evaluations are based on three widely used internal validation measures. While the relative performance of the four methods remains consistent across these measures, the reported consistency values vary considerably between them. This highlights that the choice of measure $\pi$ critically influences rank consistency, making it a crucial factor in internal validation. Results comparing rank consistency with ACC scores are provided in the Appendix. F.1, revealing similar findings. Additionally, using the unified space tends to select $K$ values closer to the true number of clusters. For instance, in the case of JULE (Figure 2) on the CMU-PIE dataset, with a true $K = 68$, the proposed method selects $K = 70$, which is the closest to the actual value, whereas using coupled spaces yield significantly less accurate estimates. Similarly, for the COIL-20 and COIL-100 datasets, the proposed method identifies highly accurate $K$ values, while other approaches deviate considerably. For DEPICT on the YTF dataset (Figure 2), the proposed approach selects $K = 45$ and $K = 50$ based on different measures, both of which are close to the true value of $K = 41$, while other methods suggest $K = 5$ in some scenarios. The optimal number of clusters detected for all datasets is reported in Figures A1 and A2.

### 4.3 ANALYSIS OF THE PROPOSED VALIDATION APPROACH

Figure 3 visualizes the final embeddings generated by the proposed approach, demonstrating that the low-dimensional embeddings effectively distinguish data points with distinct cluster labels across the displayed datasets and tasks. Figures A3 to A6 provide these visualizations for all datasets and tasks. In some cases, however, the embeddings generated from the unified space do not clearly separate the classes. Upon examining the t-SNE plots (Van der Maaten & Hinton, 2008) for individual clustering outputs in these problematic cases, we found that most candidate spaces fail to retain the local structure (see a more detailed discussion in Appendix F.3). This suggests that when the candidate spaces struggle to preserve local structure, it becomes difficult for the unified embedding space to maintain that structure as well.

Stochastic neighbor embedding methods select $\sigma_i$ in Eq. (1) by controlling the perplexity, which is a smooth measure of the effective number of neighbors (Van der Maaten & Hinton, 2008). We

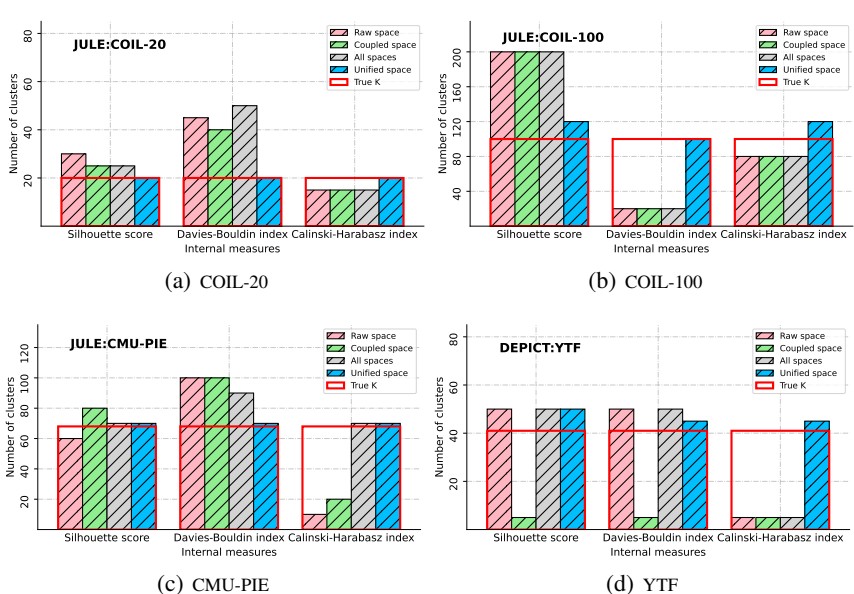

Figure 2: The optimal $K$ identified by each approach is displayed using bar plots, with the true $K$ indicated by a red, outlined, hollow box.

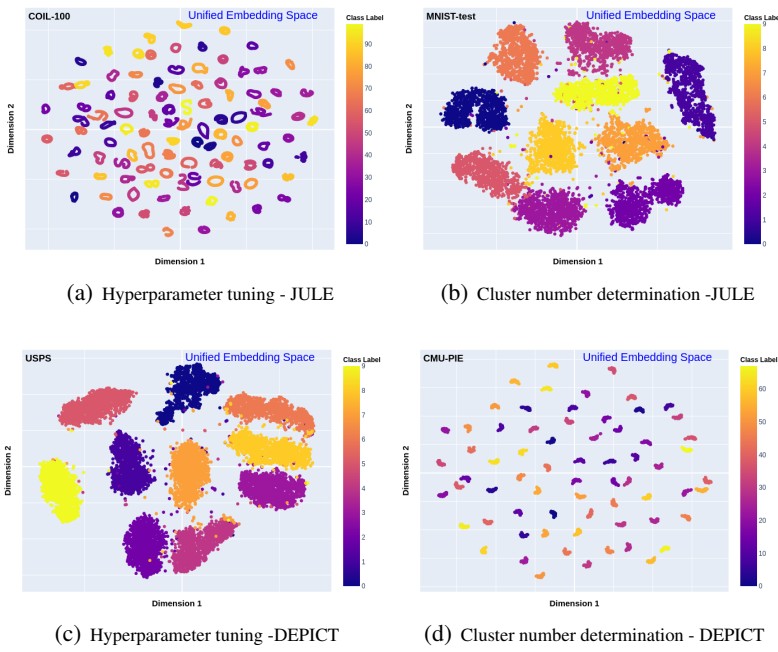

Figure 3: Visualization of low-dimensional embeddings generated by the proposed approach, with points in different colors representing distinct true cluster labels.

investigate the impact of perplexity in our method. For the main results, we selected a commonly used perplexity value of 30 and additionally examined values of 5 and 50, which represent the lower and upper bounds of the recommended range, to assess sensitivity. The results (Tables A4 to A7) indicate that, overall, our approach remains robust across these different perplexity settings. Additionally, we explore the effect of dimension on the generated low-dimensional embeddings. Given that our model employs a Cauchy distribuion, a special case of the Student's $t$-distribution, we

generated two-dimensional embeddings. While higher dimensionalities may improve the recovery of global structure, the heavy tails of the $t$-distribution in such cases can lead to distortions in local structure. Our underlying premise posits that preserving local structure, rather than global structure, facilitates a more accurate alignment of internal measures with external benchmarks (see more discussion in Section 5). We conducted experiments with dimensionalities of $4, 8, 16, 32$, and $128$. The findings reported in Tables A8 through A11 indicate that lower dimensionalities produce similarly good performance, while very high dimensionality negatively impacts the rank consistency between evaluation scores and external measures, thereby supporting our hypothesis.

## 5 DISCUSSION AND TAKE-AWAYS

This paper presents a simple yet effective internal evaluation approach by learning a unified embedding space, which addresses key challenges in deep clustering evaluation. Extensive experiments validate the framework's efficacy across various evaluation settings. Like other approaches that unify similarity matrices, the proposed approach has a computational and memory complexity of $\mathcal{O}(Mn^2)$. In our experiments, we demonstrate its applicability to evaluation tasks involving over 40 clustering results and datasets of more than 10,000 samples, which may represent a sufficiently large scale for many real-world evaluation scenarios. Several key takeaways and insights are highlighted for consideration in future research. In our method, a crucial step involves developing a unified similarity matrix by combining similarity matrices from all candidate embedding spaces. This step assumes the informativenss of the spaces obtained from deep clustering methods in contributing to the overall evaluation. If most candidate embedding spaces fail to accurately preserve the similarity information and clustering structure within the data, the unified space is likely to exhibit similar shortcomings. In such cases, the clustering results generated from these spaces are often untrustworthy, and we argue that comparing subpar results to determine which is "less bad" is not a meaningful evaluation strategy. In future work, we aim to address this issue by proposing a testing procedure to assess the viability of evaluations on the obtained embedding spaces.

In manifold learning, a trade-off often exists between preserving local and global structure during dimension reduction (Van der Maaten & Hinton, 2008; Silva & Tenenbaum, 2002). Our method employs an optimization approach similar to that used in stochastic neighbor embedding (SNE) methods. Consequently, like SNE, our approach prioritizes local structure over global structure in the data. In this work, we focus on local structure because it is generally more crucial for clustering accuracy. Clustering fundamentally involves grouping similar objects together, making the preservation of local data structure more relevant for differentiating between clusters (Rosales et al., 2004; Yang et al., 2016; Guo et al., 2017). Importantly, our goal is to achieve a more accurate evaluation of clustering results rather than simply assessing clustering quality, as these are related but distinct objectives. To achieve this, we benchmark our method against external measures to ensure better alignment with actual performance. Internal measures are typically designed to evaluate clustering quality, which may not fully reflect the correctness of clustering results. This discrepancy underscores our approach: preserving local structure in internal evaluations enhances alignment with external evaluations, given that clustering accuracy is less concerned with global geometry aspects like cluster size and distance.

Additionally, it is important to note that our approach relies on Euclidean distances for similarity calculations, which is the standard case. However, this might not yield optimal unified embeddings when alternative distance metrics, such as cosine similarity, are involved in the clustering evaluation objective.

## REPRODUCIBILITY STATEMENT

To ensure the reproducibility of our results, we provide comprehensive implementation and experimental details throughout the appendices. Appendix E.1 contains data information, while expanded implementation details of our method and specific experimental procedures are outlined in Appendix E.2. The deep clustering methods evaluated and the evaluation metrics employed in our experiments are described in Appendices E.3 and E.4, respectively. For the theorems presented in our paper, detailed proofs can be found in Appendix A. Additionally, a convergence analysis of our algorithm is provided in Appendix B.

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

# Appendix

## A  Technical Proofs

### A.1  Proof of Theorem 2.2

*Proof.* We proceed by considering two cases:

1. If we have $\mathbb{P}(\pi(\phi_1(X)|\mathcal{Z}_1) - \pi(\phi_1(X)|\mathcal{Z}_2) \geq 0) \to 1$,

   Then,

   $$\mathbb{P}(\pi(\phi_1(X)|\mathcal{Z}_1) \geq \pi(\phi_2(X)|\mathcal{Z}_2))$$
   $$\geq \mathbb{P}(\pi(\phi_1(X)|\mathcal{Z}_1) > \pi(\phi_1(X)|\mathcal{Z}_2) \text{ and } \pi(\phi_1(X)|\mathcal{Z}_2) \geq \pi(\phi_2(X)|\mathcal{Z}_2))$$
   $$\geq \mathbb{P}(\pi(\phi_1(X)|\mathcal{Z}_1) > \pi(\phi_1(X)|\mathcal{Z}_2)) + \mathbb{P}(\pi(\phi_1(X)|\mathcal{Z}_2) \geq \pi(\phi_2(X)|\mathcal{Z}_2)) - 1$$
   $$\to 1 + 1 - 1 = 1$$

| Appendix | Description |
|---|---|
| Appendix A | Technical proofs for Theorems 2.2 and 3.7 |
| Appendix B | Convergence analysis of Algorithm 1 |
| Appendix C | Overview of internal validation measures |
| Appendix D | Overview of external validation measures |
| Appendix E | Additional experimental details, including data information, implementation details, descriptions of deep clustering methods, and evaluation metrics |
| Appendix F | Supplementary experimental results: rank consistency with ACC scores, optimal number of clusters, visualizations of unified embeddings, and sensitivity analysis |

as $n \to \infty$.

2. If $\mathbb{P}(\pi(\phi_1(X)|\mathcal{Z}_1) - \pi(\phi_1(X)|\mathcal{Z}_2) \geq 0) \to 1$ does not hold,

i) Consider the case where $\phi_1(X) = \phi_2(X)$, i.e., $\phi_1(X)$ and $\phi_2(X)$ are the same.

$$\begin{aligned}
&\mathbb{P}(\pi(\phi_1(X)|\mathcal{Z}_1) - \pi(\phi_2(X)|\mathcal{Z}_2)) \geq 0) \\
=&\mathbb{P}(\pi(\phi_1(X)|\mathcal{Z}_1) - \pi(\phi_1(X)|\mathcal{Z}_2)) \geq 0) \\
=&1 - \mathbb{P}(\pi(\phi_1(X)|\mathcal{Z}_1) - \pi(\phi_1(X)|\mathcal{Z}_2)) < 0) \\
\to&0.
\end{aligned}$$

So $\mathbb{P}(\pi(\phi_1(X)|\mathcal{Z}_1) - \pi(\phi_2(X)|\mathcal{Z}_2)) \geq 0)$ does not converge to 1.

ii) Consider the case where $\phi_1(X) \neq \phi_2(X)$. Then we have the following decomposition:

$$\pi(\phi_1(X)|\mathcal{Z}_1) - \pi(\phi_2(X)|\mathcal{Z}_2) = [\pi(\phi_1(X)|\mathcal{Z}_1) - \pi(\phi_2(X)|\mathcal{Z}_1)] - [\pi(\phi_2(X)|\mathcal{Z}_2) - \pi(\phi_2(X)|\mathcal{Z}_1)].$$

The first quantity $[\pi(\phi_1(X)|\mathcal{Z}_1) - \pi(\phi_2(X)|\mathcal{Z}_1)]$ represents the clustering difference on space $\mathcal{Z}_1$, and the second quantity $[\pi(\phi_2(X)|\mathcal{Z}_2) - \pi(\phi_2(X)|\mathcal{Z}_1)]$ represents the space difference.

If the clustering difference is larger than the space difference, we then have $\pi(\phi_1(X)|\mathcal{Z}_1) > \pi(\phi_2(X)|\mathcal{Z}_2)$. To give a counterexample, if we have $\mathbb{P}(\max_{\phi_1} [\pi(\phi_1(X)|\mathcal{Z}_1) - \pi(\phi_2(X)|\mathcal{Z}_1)] < [\pi(\phi_2(X)|\mathcal{Z}_2) - \pi(\phi_2(X)|\mathcal{Z}_1)]) \to c$ for some $0 < c < 1$, then

$$\begin{aligned}
&\mathbb{P}(\pi(\phi_1(X)|\mathcal{Z}_1) - \pi(\phi_2(X)|\mathcal{Z}_2) > 0) \\
=&1 - \mathbb{P}(\pi(\phi_1(X)|\mathcal{Z}_1) - \pi(\phi_2(X)|\mathcal{Z}_1) < \pi(\phi_2(X)|\mathcal{Z}_2) - \pi(\phi_2(X)|\mathcal{Z}_1)) \\
\leq&1 - \mathbb{P}(\max_{\phi_1} [\pi(\phi_1(X)|\mathcal{Z}_1) - \pi(\phi_2(X)|\mathcal{Z}_1)] < \pi(\phi_2(X)|\mathcal{Z}_2) - \pi(\phi_2(X)|\mathcal{Z}_1)) \\
\to&1 - c < 1.
\end{aligned}$$

In summary, $\mathbb{P}(\pi(\phi_1(X)|\mathcal{Z}_1) > \pi(\phi_2(X)|\mathcal{Z}_2)) \to 1$ happens only when $\mathbb{P}(\pi(\phi_1(X)|\mathcal{Z}_1) - \pi(\phi_1(X)|\mathcal{Z}_2) \geq 0) \to 1$.

$\square$

## A.2 PROOF OF THEOREM 3.7

*Proof.* (a) Denote by $A_{\mathcal{Z}}$ the set of variables contained in $A_{\mathcal{Z}}$ but not in $A_{\mathcal{Z}^{(k)}}$. Since

$$\frac{\sum_{k=1}^{K} \mathsf{w}_k |A_{\mathcal{Z}} \backslash A_{\mathcal{Z}^{(k)}}|}{|A_{\mathcal{Z}}|} = \frac{\sum_{k=1}^{K} \mathsf{w}_k \sum_{(i,j_1,j_2) \in A_{\mathcal{Z}}} I((i,j_1,j_2) \notin A_{\mathcal{Z}^{(k)}})}{|A_{\mathcal{Z}}|}$$

$$= \frac{\sum_{(i,j_1,j_2) \in A_{\mathcal{Z}}} \sum_{k=1}^{K} \mathsf{w}_k I((i,j_1,j_2) \notin A_{\mathcal{Z}^{(k)}})}{|A_{\mathcal{Z}}|}$$

$$= \frac{\sum_{(i,j_1,j_2) \in A_{\mathcal{Z}}} \sum_{k=1}^{K} \mathsf{w}_k (1 - I((i,j_1,j_2) \in A_{\mathcal{Z}^{(k)}}))}{|A_{\mathcal{Z}}|}$$

$$= \frac{\sum_{(i,j_1,j_2) \in A_{\mathcal{Z}}} (1 - s_{i,j_1,j_2})}{|A_{\mathcal{Z}}|}.$$

and by the definition of weak consistency,

$$0 \le \frac{\sum_{k=1}^{K} \mathsf{w}_k |A_{\mathcal{Z}} \backslash A_{\mathcal{Z}^{(k)}}|}{|A_{\mathcal{Z}}|} \le \frac{\sum_{k=1}^{K} \mathsf{w}_k |A_{\mathcal{Z}^{(k)}} \nabla A_{\mathcal{Z}}|}{|A_{\mathcal{Z}}|} \xrightarrow{p} 0.$$

Hence,

$$\frac{\sum_{(i,j_1,j_2) \in A_{\mathcal{Z}}} (1 - s_{i,j_1,j_2})}{|A_{\mathcal{Z}}|} \xrightarrow{p} 0.$$

On the other hand,

$$\frac{\sum_{(i,j_1,j_2) \notin A_{\mathcal{Z}}} s_{i,j_1,j_2}}{|A_{\mathcal{Z}}|} = \frac{\sum_{(i,j_1,j_2) \notin A_{\mathcal{Z}}} \sum_{k=1}^{K} \mathsf{w}_k I((i,j_1,j_2) \in A_{\mathcal{Z}^{(k)}})}{|A_{\mathcal{Z}}|}$$

$$= \frac{\sum_{k=1}^{K} \mathsf{w}_k \sum_{(i,j_1,j_2) \notin A_{\mathcal{Z}}} I((i,j_1,j_2) \in A_{\mathcal{Z}^{(k)}})}{|A_{\mathcal{Z}}|}$$

$$= \frac{\sum_{k=1}^{K} \mathsf{w}_k |A_{\mathcal{Z}^{(k)}} \backslash A_{\mathcal{Z}}|}{|A_{\mathcal{Z}}|}$$

$$\le \frac{\sum_{k=1}^{K} \mathsf{w}_k |A_{\mathcal{Z}^{(k)}} \nabla A_{\mathcal{Z}}|}{|A_{\mathcal{Z}}|} \xrightarrow{p} 0.$$

(b) We omit the proof since it is similar to that of (a) without the denominator $|A_{\mathcal{Z}}|$.

$\square$

# B CONVERGENCE ANALYSIS

In this section, we provide a convergence analysis of Algorithm 1.

**Lemma B.1.** *[Nie et al. (2010); Nie et al.] For any positive numbers $a$ and $b$, we have the inequality:*

$$a - \frac{a^2}{2b} \le b - \frac{b^2}{2b} \tag{8}$$

**Theorem B.2.** *Each iteration of Algorithm 1 monotonically decreases the objective function in Eq. (2), ensuring convergence to a local optimum of the optimization problem.*

*Proof.* By updating $w^{(m)}$ according to Eq. (4), the objective function in Eq. (2) becomes $\sum_{m=1}^{M} \|U - S^{(m)}\|_F$. We will now prove that Algorithm 1 decreases this function monotonically.

Let $U^t$ and $U^{t-1}$ represent the matrix $U$ after and before the update at each iteration, respectively.

We first show that with $w^{(m)}$ fixed, the solution from Eq. (5) satisfies:

$$\sum_{m=1}^{M} w^{(m)} \left\| U^t - S^{(m)} \right\|_F^2 \leq \sum_{m=1}^{M} w^{(m)} \left\| U^{t-1} - S^{(m)} \right\|_F^2 \tag{9}$$

The optimization problem in Eq. (2) can be rewritten as:

$$\min_{\{u_{ij}\}_{i,j=1}^n} \sum_{i,j=1}^{n} \sum_{m=1}^{M} w^{(m)} (u_{ij} - s_{ij}^{(m)})^2 \tag{10}$$

Since $w^{(m)}$ is fixed and positive, this optimization problem is equivalent to:

$$\min_{\{u_{ij}\}_{i,j=1}^n} \sum_{i,j=1}^{n} (u_{ij} - \sum_{m=1}^{M} w^{(m)} s_{ij}^{(m)} / \sum_{m=1}^{M} w^{(m)})^2, \tag{11}$$

which indicates that $U$ in Eq. (5) is the minimizer. Furthermore, the $U$ from Eq. (4) satisfy the constraints in Eq. 2, since each $S^{(m)}$ is a non-negative matrix with row vectors summing to one, and $w^{(m)}$ is positive. Thus, the updated $U^t$ minimizes the objective function in Eq. (2), leading to the inequality in Eq. (9).

Updating the $(t-1)$-th iteration according to Eq. (4), we have the weight $w^{(m)} = \frac{1}{2\left\| U^{t-1} - S^{(m)} \right\|_F}$. Following a similar proof process as in Nie et al., and using Eq. 9, we can derive the following inequality:

$$\sum_{m=1}^{M} \frac{\left\| U^t - S^{(m)} \right\|_F^2}{2\left\| U^{t-1} - S^{(m)} \right\|_F} \leq \sum_{m=1}^{M} \frac{\left\| U^{t-1} - S^{(m)} \right\|_F^2}{2\left\| U^{t-1} - S^{(m)} \right\|_F} \tag{12}$$

According to Lemma B.1, we futher have

$$\sum_{m=1}^{M} \left\| U^t - S^{(m)} \right\|_F - \sum_{m=1}^{M} \frac{\left\| U^t - S^{(m)} \right\|_F^2}{2\left\| U^{t-1} - S^{(m)} \right\|_F} \tag{13}$$

$$\leq \sum_{m=1}^{M} \left\| U^{t-1} - S^{(m)} \right\|_F - \sum_{m=1}^{M} \frac{\left\| U^{t-1} - S^{(m)} \right\|_F^2}{2\left\| U^{t-1} - S^{(m)} \right\|_F} \tag{14}$$

Thus, we obtain:

$$\sum_{m=1}^{M} \left\| U^t - S^{(m)} \right\|_F - \sum_{m=1}^{M} \left\| U^{t-1} - S^{(m)} \right\|_F$$

$$\leq \sum_{m=1}^{M} \frac{\left\| U^t - S^{(m)} \right\|_F^2}{2\left\| U^{t-1} - S^{(m)} \right\|_F} - \sum_{m=1}^{M} \frac{\left\| U^{t-1} - S^{(m)} \right\|_F^2}{2\left\| U^{t-1} - S^{(m)} \right\|_F}$$

Together with Eq. (12), we have:

$$\sum_{m=1}^{M} \left\| U^t - S^{(m)} \right\|_F \leq \sum_{m=1}^{M} \left\| U^{t-1} - S^{(m)} \right\|_F$$

This shows that each iteration results in a monotonic decrease of the non-negative objective function, thus guaranteeing the convergence of the algorithm to a local minimum.

$\square$

## C    INTERNAL VALIDATION MEASURES

In this section, we provide further details on the three internal validation measures discussed and applied in the paper: the Silhouette score (Rousseeuw, 1987), the Calinski-Harabasz index (Caliński & Harabasz, 1974; Desgraupes, 2013), and the Davies-Bouldin index (Davies & Bouldin, 1979).

**Notation.**    Denote the dataset in $\mathbb{R}^p$, used for both clustering and evaluation, by $\{x_1, \cdots, x_N\}$. Denote the $k$-th cluster by $C_k$, with $n_k$ representing the cardinality of $C_k$. Following the notation in Desgraupes (2013), let $\mu^{\{k\}}$ be the centroid of the cluster $C_k$, and $\mu$ be the centroid of all observations. That is,

$$\mu^{\{k\}} = \frac{1}{n_k} \sum_{i \in C_k} x_i$$

$$\mu = \frac{1}{N} \sum_{i=1}^{N} x_i$$

(15)

**Silhouette Score (Rousseeuw, 1987)**    For any two observations $x_i$ and $x_j$, let

$$a(i) = \frac{1}{|C_I| - 1} \sum_{j \in C_I, i \neq j} d(i, j)$$

(16)

denote the average distance between the $i$-th observation and all other observations within its cluster $C_I$, where $d(i, j) := d(x_i, x_j)$ and $d(\cdot)$ is a distance function (we choose Euclidean distance, a commonly used metric, for this work). Let

$$b(i) = \min_{J \neq I} \frac{1}{|C_J|} \sum_{j \in C_J} d(i, j)$$

(17)

denote the smallest distance between the $i$-th observation and any other cluster. The Silhouette value for $x_i$ is defined as:

$$s(i) = \frac{b(i) - a(i)}{\max\{a(i), b(i)\}}.$$

(18)

The Silhouette score is defined as:

$$\pi_{Silhouette} = \frac{1}{K} \sum_{k=1}^{K} \frac{1}{n_k} \sum_{i \in C_k} s(i).$$

(19)

A higher Silhouette score generally signifies better clustering quality.

**Davies-Bouldin index (Davies & Bouldin, 1979)**    Let

$$\delta_k = \frac{1}{n_k} \sum_{i \in C_k} \left\| x_i - \mu^{\{k\}} \right\|$$

(20)

represent the average Euclidean distance of points within cluster $C_k$ to the centroid $\mu^{\{k\}}$. Let

$$\Delta_{kk'} = d\left(\mu^{\{k\}}, \mu^{\{k'\}}\right) = \left\| \mu^{\{k'\}} - \mu^{\{k\}} \right\|$$

(21)

denote the Euclidean distance between $\mu^{\{k\}}$ and $\mu^{\{k'\}}$.

For each cluster $k$, define

$$M_k = \max_{k' \neq k} \left( \frac{\delta_k + \delta_{k'}}{\Delta_{kk'}} \right).$$

(22)

The Davies-Bouldin index is the average of $M_k$ across all clusters:

$$\pi_{Davies-Bouldin} = \frac{1}{K} \sum_{k=1}^{K} M_k. \tag{23}$$

A lower Davies-Bouldin index generally indicates higher clustering quality. Therefore, when using rank correlation with external measures to evaluate the performance based on the Davies-Bouldin index, we apply a negative sign to the calculated value to ensure proper alignment with the evaluation criteria.

**Calinski-Harabasz index (Caliński & Harabasz, 1974)** The within-cluster dispersion is defined as

$$WGSS^{\{k\}} = \sum_{i \in C_k} ||x_i - \mu^{\{k\}}||^2 = \frac{1}{n_k} \sum_{i < j \in C_k} |x_i - x_j|^2. \tag{24}$$

The pooled within-cluster sum of squares (WGSS) is then defined as the total of the within-cluster dispersions over all clusters:

$$WGSS = \sum_{k=0}^{K} WGSS^{\{k\}}. \tag{25}$$

The between-group dispersion (BGSS) is defined as

$$BGSS = \sum_{k=1}^{K} n_k \left\| \mu^{\{k\}} - \mu \right\|^2. \tag{26}$$

The Calinski-Harabasz index is expressed as:

$$\pi_{Calinski-Harabasz} = \frac{BGSS/(K-1)}{WGSS/(N-K)}. \tag{27}$$

A higher Calinski-Harabasz index generally indicates better clustering quality.

# D  EXTERNAL VALIDATION MEASURE

**Normalized Mutual Information** For two distinct cluster assignments $Y_1$ and $Y_2$, the Normalized Mutual Information (NMI) is defined as:

$$NMI(Y_1; Y_2) = \frac{2 \times I(Y_1; Y_2)}{H(Y_1) + H(Y_2)}. \tag{28}$$

$I$ represents the mutual information between $Y_1$ and $Y_2$, while $H$ denotes the entropy function. The Normalized Mutual Information (NMI) ranges from 0, indicating no mutual information, to 1, indicating perfect correlation. For evaluating clustering results, we use $Y$ to denote the true cluster labels and $\hat{Y}$ to denote the estimated cluster labels. We express this as $NMI(Y; \hat{Y})$.

**Clustering accuracy** The clustering accuracy (ACC) in estimating the true labels $Y$ against the estimated labels $\hat{Y}$ is defined as:

$$ACC(Y, \hat{Y}) = \max_{\text{perm} \in P} \frac{\sum_{i=1}^{N} I\{\text{perm}(\hat{y}_i) = y_i\}}{N} \tag{29}$$

where $P$ represents the set of all possible permutations of the indices. Clustering accuracy computes the proportion of correctly matched pairs up to the best permutation.

# E  ADDITIONAL EXPERIMENTAL DETAILS

## E.1  DATA INFORMATION

Table A1 provides detailed information on the datasets, including sample size, image size, and number of classes for COIL20 (Nene et al., 1996), COIL100 (Nene et al., 1996), CMU-PIE (Sim et al., 2002),

UMist (Graham & Allinson, 1998), FRGC[3], YTF (Wolf et al., 2011), MNIST-test (LeCun et al., 1998), and USPS[4].

Table A1: Data description

| Dataset | Sample Size | Image Dimension | Class Count |
|---|---|---|---|
| COIL20 | 1440 | 128×128 | 20 |
| COIL100 | 7200 | 128×128 | 100 |
| CMU-PIE | 2856 | 32×32 | 68 |
| UMist | 575 | 112×92 | 20 |
| FRGC | 2462 | 32×32 | 20 |
| YTF | 10000 | 55×55 | 41 |
| MNIST-test | 10000 | 28×28 | 10 |
| USPS | 11000 | 16×16 | 10 |

### E.2 EXPANDED EXPERIMENTAL AND IMPLEMENTATION DETAILS

We provide additional experimental details to ensure full reproducibility of our results. For our method implementation, we set the perplexity to 30 and reduce the data to two dimensions, as recommended in the original work on stochastic neighbor embedding (Van der Maaten & Hinton, 2008). We also conduct a sensitivity analysis with different values, reported in Appendix F.4. The step of unifying the similarity matrix does not require hyperparameter tuning. The convergence criterion in Algorithm 1 is set when the absolute difference in the objective function is less than $1e-8$. For embedding optimization, we follow the t-SNE implementation from the *sklearn* library (Pedregosa et al., 2011), adhering to all default settings except for randomly initializing the embeddings. Specifically, the early exaggeration factor is 12, the learning rate is $\max(n/\text{early exaggeration}/4, 50)$, and the momentum is set to 0.5 for exploration and 0.8 for remaining iterations.

We compute tnternal measures, including the Silhouette score, Calinski-Harabasz index, and Davies-Bouldin index from the sklearn library in Python. We run JULE and DEPICT using their source code. For JULE, we explore 42 hyperparameter combinations, selecting the learning rate from $[0.0005, 0.001, 0.005, 0.01, 0.05, 0.1]$ and the unfolding rate ($\eta$) from $[0.2, 0.3, 0.4, 0.5, 0.7, 0.8, 0.9]$ that include the values of 0.2 and 0.9 suggested in the original paper. For DEPICT, we explore 18 combinations, selecting the learning rate from $[0.00005, 0.0001, 0.0005, 0.001, 0.005, 0.01]$ and the reconstruction loss balancing parameter from $[0.1, 1.0, 10.0]$. Failed trials are excluded from the evaluation. For the cluster number determination experiment, we search for $K$ among ten evenly spaced values that encompass the true $K$ or a nearby value. For MNIST-test, USPS, FRGC, UMist, YTF, and COIL-20, we generate a sequence of 10 evenly spaced values ranging from 5 to 50; for CMU-PIE, we generate a sequence of 10 evenly spaced values ranging from 10 to 100; and for COIL-100, we generate a sequence of 10 evenly spaced values ranging from 20 to 200. In cases of failed trials, we either exclude that $K$ or use a nearby value (e.g., $K = 11$ instead of $K = 10$ for JULE on YTF).

### E.3 DEEP CLUSTERING ALGORITHMS

In this section, we provide more details regarding the deep clustering algorithms evaluated in this paper: JULE (Yang et al., 2016) and DEPICT (Ghasedi Dizaji et al., 2017).

**JULE (Yang et al., 2016)** is a widely cited method for joint unsupervised learning that employs agglomerative clustering techniques to perform deep clustering tasks. Unlike approaches that integrate autoencoders, JULE directly trains the feature extractor (encoder) within a deep neural network using a joint learning strategy in a recurrent framework. In this framework, the merging operations of agglomerative clustering are executed as part of the forward pass, allowing the generation of cluster labels. During the backward pass, the network learns deep representations and updates its parameters based on these generated labels. JULE introduces a unified weighted triplet loss function that captures both the affinity between clusters and the local structure surrounding them. Each epoch involves

---

[3] http://www3.nd.edu/~cvrl/CVRL/Data$_$Sets.html
[4] https://cs.nyu.edu/~roweis/data.html

merging two clusters and computing the associated loss, which is optimized in an end-to-end manner to concurrently estimate cluster labels and embed the data. A critical hyperparameter in this algorithm is the unfolding rate, which determines the number of timesteps used for the agglomerative clustering process. A lower unfolding rate results in more frequent updates to the network.

**DEPICT (Ghasedi Dizaji et al., 2017)** is a well-cited method for deep clustering that operates within an autoencoder framework. This algorithm features a design that integrates a multinomial logistic regression function on top of a multilayer convolutional autoencoder. DEPICT introduces a clustering loss function that effectively maps data into a discriminative embedding subspace, enhancing the quality of the learned representations. The optimization objective is formulated by minimizing relative entropy (KL divergence), supplemented with regularization to account for the frequency of cluster assignments. In addition to the clustering task, DEPICT incorporates an auxiliary reconstruction task, employing a reconstruction loss to ensure the fidelity of the learned representations. Utilizing a joint learning framework, DEPICT concurrently minimizes both the clustering loss and the reconstruction loss, enabling accurate predictions of cluster assignments while simultaneously improving the learning of feature embeddings. A key hyperparameter in this algorithm is the balancing parameter for the reconstruction loss, which adjusts the trade-off between the clustering and reconstruction losses to optimize overall performance.

### E.4 EVALUATION METRICS

**Spearman's rank correlation coefficient (Spearman, 1961; Zwillinger & Kokoska, 1999; Kiefer, 1964)** Spearman's rank correlation is a nonparametric statistic that evaluates the strength and direction of monotonic relationships between two random variables.

Given $n$ pairs of values $(X_i, Y_i)$, where $i = 1, \ldots, n$, let $\mathrm{R}(X_i)$ represent the rank of $X_i$ among $\{X_1, \ldots, X_n\}$, and define $\mathrm{R}(Y_i)$ in the same way for $\{Y_1, \ldots, Y_n\}$. The Spearman's rank correlation $r_s$ is then calculated as the Pearson correlation between the ranked values $\{\mathrm{R}(X_i)\}_{i=1}^n$ and $\{\mathrm{R}(Y_i)\}_{i=1}^n$:

$$r_s = r_{\mathbb{P}}(\mathrm{R}(X), \mathrm{R}(Y)) = \frac{\mathrm{cov}(\mathrm{R}(X), \mathrm{R}(Y))}{\sigma_{\mathrm{R}(X)} \sigma_{\mathrm{R}(Y)}}, \tag{30}$$

where $\mathrm{cov}(\mathrm{R}(X), \mathrm{R}(Y))$ represents the covariance of the ranked variables, and $\sigma_{\mathrm{R}(X)}$ and $\sigma_{\mathrm{R}(Y)}$ are their respective standard deviations.

**Kendall rank correlation coefficient (Kendall, 1938; Agresti, 2010; Knight, 1966)** Let $(x_1, y_1), \cdots, (x_n, y_n)$ denote the set of observations corresponding to the random variables $(X, Y)$. For any pair of observations $(x_i, y_i)$ and $(x_j, y_j)$ with $i < j$, they are deemed concordant if the sort order of $(x_i, x_j)$ and $(y_i, y_j)$ aligns, i.e., $(x_i - x_j) \cdot (y_i - y_j) > 0$. We say $(x_i, y_i)$ and $(x_j, y_j)$ form a tied pair if either $x_i = x_j$ or $y_i = y_j$. We say $(x_i, y_i)$ and $(x_j, y_j)$ are discordant if they are neither concordant nor tied.

With these definitions, the Kendall coefficient $\tau_B$ is defined as:

$$\tau_B = \frac{n_c - n_d}{\sqrt{(n_0 - n_1)(n_0 - n_2)}} \tag{31}$$

where $n_0 = n(n-1)/2$, $n_1 = \sum_i t_i(t_i - 1)/2$, $n_2 = \sum_j u_j(u_j - 1)/2$. Here, $n_c$ and $n_d$ denote the number of concordant and discordant pairs, respectively. $t_i$ represents the number of tied values in the $i$-th group of ties for the first variable (e.g., $X$ in the pair $\{X, Y\}$), while $u_j$ corresponds to the number of tied values in the $j$-th group of ties for the second variable (e.g., $Y$ in the pair $\{X, Y\}$). The count of discordant pairs is equivalent to the inversion number, which represents how many rearrangements are needed to permute the $Y$-sequence with the same order of the $X$-sequence.

## F ADDITIONAL RESULTS

In this section, we provide additional results, including tables, figures, and sensitivity analysis outcomes that were not included in the paper due to page limitations.

## F.1 RANK CONSISTENCY WITH ACC

We present the performance of the four evaluation approaches in terms of rank consistency with ACC scores for both tasks, as shown in Table A2 and Table A3. The results align with our earlier observations on rank consistency with NMI scores (Tables 1 and 2). The proposed method consistently achieves the highest average rank correlation with ACC scores across most scenarios. In the few cases where it does not, its performance remains very close to the method with the highest rank correlation. Additionally, both the proposed method and the approach using all spaces generally demonstrate stronger rank correlations with ACC scores than those derived from coupled space or raw space, reinforcing the conclusions drawn from rank consistency with NMI scores. These findings further support the effectiveness of the proposed approach as discussed in the main text.

Table A2: Rank consistency between the ACC scores and those generated by the evaluation regime using different spaces for hyperparameter tuning.

| | USPS | | YTF | | FRGC | | MNIST-test | | CMU-PIE | | UMist | | COIL-20 | | COIL-100 | | Average | |
|---|---|---|---|---|---|---|---|---|---|---|---|---|---|---|---|---|---|---|
| | $r_s$ | $\tau_B$ | $r_s$ | $\tau_B$ | $r_s$ | $\tau_B$ | $r_s$ | $\tau_B$ | $r_s$ | $\tau_B$ | $r_s$ | $\tau_B$ | $r_s$ | $\tau_B$ | $r_s$ | $\tau_B$ | $r_s$ | $\tau_B$ |
| JULE: Calinski-Harabasz index | | | | | | | | | | | | | | | | | | |
| Raw space | 0.70 | 0.59 | 0.54 | 0.39 | -0.52 | -0.35 | 0.91 | 0.76 | -0.98 | -0.91 | -0.50 | -0.35 | -0.29 | -0.17 | 0.36 | 0.23 | 0.03 | 0.02 |
| Coupled space | 0.04 | 0.05 | 0.39 | 0.27 | -0.26 | -0.18 | 0.31 | 0.21 | -0.20 | -0.12 | 0.64 | 0.45 | 0.57 | 0.40 | 0.09 | 0.08 | 0.20 | 0.14 |
| All spaces | **0.92** | **0.79** | **0.78** | **0.61** | **0.30** | **0.21** | 0.91 | 0.77 | 0.91 | 0.78 | **0.65** | **0.47** | 0.57 | 0.42 | 0.91 | **0.78** | **0.74** | **0.60** |
| **Unified space** | 0.88 | 0.71 | 0.58 | 0.43 | 0.21 | 0.14 | **0.94** | **0.80** | **0.98** | **0.90** | 0.62 | 0.42 | **0.73** | **0.55** | **0.92** | 0.77 | 0.73 | 0.59 |
| JULE: Davies-Bouldin index | | | | | | | | | | | | | | | | | | |
| Raw space | -0.67 | -0.43 | -0.45 | -0.30 | -0.04 | -0.01 | -0.94 | -0.80 | -0.96 | -0.86 | -0.77 | -0.60 | -0.56 | -0.38 | -0.83 | -0.64 | -0.65 | -0.50 |
| Coupled space | -0.27 | -0.15 | **-0.14** | **-0.09** | -0.23 | -0.14 | -0.35 | -0.19 | 0.20 | 0.16 | **0.53** | **0.36** | **0.63** | **0.44** | 0.33 | 0.26 | 0.09 | 0.08 |
| All spaces | -0.49 | -0.21 | -0.35 | -0.23 | 0.49 | 0.36 | -0.35 | -0.20 | 0.89 | 0.76 | -0.47 | -0.34 | -0.30 | -0.22 | -0.48 | -0.34 | -0.13 | -0.05 |
| **Unified space** | **0.28** | **0.27** | -0.21 | -0.14 | **0.53** | **0.37** | **0.89** | **0.71** | **0.94** | **0.82** | -0.28 | -0.21 | 0.48 | 0.37 | **0.75** | **0.56** | **0.42** | **0.34** |
| JULE: Silhouette score | | | | | | | | | | | | | | | | | | |
| Raw space | 0.92 | 0.77 | 0.59 | 0.43 | 0.27 | 0.19 | 0.83 | 0.66 | 0.35 | 0.32 | -0.35 | -0.24 | -0.14 | -0.05 | 0.14 | 0.08 | 0.33 | 0.27 |
| Coupled space | 0.14 | 0.12 | 0.54 | 0.39 | -0.08 | -0.02 | 0.41 | 0.27 | 0.36 | 0.27 | 0.64 | 0.46 | 0.67 | 0.48 | 0.44 | 0.31 | 0.39 | 0.28 |
| All spaces | 0.74 | 0.68 | 0.66 | 0.49 | 0.71 | **0.53** | 0.89 | 0.72 | 0.96 | 0.87 | 0.64 | 0.43 | 0.19 | 0.10 | 0.62 | 0.45 | 0.68 | 0.53 |
| **Unified space** | **0.93** | **0.79** | **0.80** | **0.63** | 0.72 | 0.53 | **0.94** | **0.80** | 0.96 | 0.87 | 0.45 | 0.29 | 0.57 | 0.39 | **0.91** | **0.76** | **0.79** | **0.64** |
| DEPICT: Calinski-Harabasz index | | | | | | | | | | | | | | | | | | |
| Raw space | -0.10 | -0.19 | **0.65** | **0.50** | 0.54 | 0.38 | 0.59 | 0.47 | -0.95 | -0.83 | | | | | | | 0.14 | 0.07 |
| Coupled space | 0.56 | 0.40 | 0.54 | 0.35 | 0.76 | 0.57 | 0.88 | 0.69 | 0.48 | 0.43 | | | | | | | 0.64 | 0.49 |
| All spaces | **0.94** | **0.83** | 0.54 | 0.45 | 0.92 | 0.79 | 0.95 | 0.86 | 0.74 | 0.62 | | | | | | | 0.82 | 0.71 |
| **Unified space** | 0.87 | 0.70 | 0.57 | 0.42 | **0.93** | **0.80** | **0.96** | **0.88** | **0.95** | **0.81** | | | | | | | **0.86** | **0.72** |
| DEPICT: Davies-Bouldin index | | | | | | | | | | | | | | | | | | |
| Raw space | 0.06 | -0.09 | 0.48 | **0.33** | 0.53 | 0.39 | 0.13 | 0.07 | -0.14 | -0.20 | | | | | | | 0.21 | 0.10 |
| Coupled space | 0.61 | 0.42 | 0.48 | 0.32 | **0.92** | **0.74** | 0.88 | 0.69 | 0.62 | 0.56 | | | | | | | 0.70 | 0.55 |
| All spaces | **0.93** | **0.80** | 0.40 | 0.28 | 0.65 | 0.50 | 0.45 | 0.32 | 0.24 | 0.07 | | | | | | | 0.53 | 0.39 |
| **Unified space** | 0.85 | 0.71 | **0.50** | 0.33 | 0.72 | 0.56 | **0.92** | **0.82** | **0.98** | **0.91** | | | | | | | **0.79** | **0.66** |
| DEPICT: Silhouette score | | | | | | | | | | | | | | | | | | |
| Raw space | 0.45 | 0.27 | **0.75** | **0.59** | 0.69 | 0.51 | 0.79 | 0.63 | -0.23 | -0.13 | | | | | | | 0.49 | 0.37 |
| Coupled space | 0.52 | 0.33 | 0.57 | 0.45 | 0.80 | 0.62 | 0.85 | 0.65 | 0.59 | 0.48 | | | | | | | 0.67 | 0.51 |
| All spaces | **0.95** | **0.86** | 0.72 | 0.57 | 0.94 | 0.82 | 0.96 | 0.88 | 0.95 | 0.85 | | | | | | | **0.91** | **0.80** |
| **Unified space** | 0.88 | 0.74 | 0.69 | 0.53 | **0.95** | **0.84** | 0.96 | 0.88 | **0.96** | **0.87** | | | | | | | 0.89 | 0.77 |

Table A3: Rank consistency between the ACC scores and those generated by the evaluation regime using different spaces for cluster number determination.

| | USPS (10) | | YTF (41) | | FRGC (20) | | MNIST-test (10) | | CMU-PIE (68) | | UMist (20) | | COIL-20 (20) | | COIL-100 (100) | | Average | |
|---|---|---|---|---|---|---|---|---|---|---|---|---|---|---|---|---|---|---|
| | $r_s$ | $\tau_B$ | $r_s$ | $\tau_B$ | $r_s$ | $\tau_B$ | $r_s$ | $\tau_B$ | $r_s$ | $\tau_B$ | $r_s$ | $\tau_B$ | $r_s$ | $\tau_B$ | $r_s$ | $\tau_B$ | $r_s$ | $\tau_B$ |
| JULE: Calinski-Harabasz index | | | | | | | | | | | | | | | | | | |
| Raw space | 0.71 | 0.64 | **1.00** | **1.00** | -0.46 | -0.25 | 0.41 | 0.47 | -0.38 | -0.29 | -0.09 | -0.02 | 0.76 | **0.71** | 0.36 | 0.33 | 0.29 | 0.32 |
| Coupled space | 0.84 | **0.73** | 0.03 | -0.06 | -0.49 | -0.31 | 0.61 | 0.56 | -0.09 | -0.07 | -0.04 | 0.07 | 0.74 | 0.64 | 0.60 | 0.51 | 0.27 | 0.26 |
| All spaces | 0.78 | 0.69 | 0.88 | 0.78 | -0.37 | -0.20 | 0.61 | 0.56 | 0.83 | 0.69 | -0.07 | 0.02 | 0.76 | 0.71 | 0.56 | 0.51 | 0.50 | 0.47 |
| **Unified space** | **0.88** | 0.73 | 0.95 | 0.89 | **0.37** | **0.37** | **0.94** | **0.82** | **0.96** | **0.91** | **0.19** | **0.11** | 0.81 | 0.64 | **0.77** | **0.64** | **0.73** | **0.64** |
| JULE: Davies-Bouldin index | | | | | | | | | | | | | | | | | | |
| Raw space | -0.49 | -0.38 | **0.85** | 0.67 | 0.37 | 0.20 | -0.41 | -0.38 | 0.77 | 0.51 | **0.02** | **-0.16** | -0.86 | -0.71 | -0.82 | -0.78 | -0.07 | -0.13 |
| Coupled space | 0.39 | 0.29 | 0.10 | 0.06 | 0.37 | 0.25 | 0.49 | 0.33 | 0.83 | 0.60 | -0.28 | -0.29 | -0.29 | -0.21 | -0.87 | -0.73 | 0.09 | 0.04 |
| All spaces | **0.77** | **0.56** | 0.80 | 0.67 | **0.71** | **0.54** | **0.84** | **0.69** | 0.85 | 0.69 | -0.06 | -0.20 | -0.69 | -0.57 | -0.79 | -0.69 | 0.30 | 0.21 |
| **Unified space** | 0.53 | 0.42 | 0.43 | 0.28 | 0.49 | 0.37 | 0.53 | 0.42 | **0.93** | **0.87** | -0.58 | -0.33 | 0.41 | 0.36 | **0.85** | **0.64** | **0.45** | **0.38** |
| JULE: Silhouette score | | | | | | | | | | | | | | | | | | |
| Raw space | 0.62 | 0.56 | **0.95** | **0.89** | -0.17 | -0.14 | 0.53 | 0.42 | 0.53 | 0.33 | 0.04 | -0.07 | -0.38 | -0.29 | 0.52 | 0.33 | 0.33 | 0.25 |
| Coupled space | **0.93** | **0.82** | 0.30 | 0.28 | 0.21 | 0.09 | 0.82 | 0.64 | 0.98 | 0.91 | -0.13 | -0.16 | 0.52 | 0.36 | 0.55 | 0.42 | 0.52 | 0.42 |
| All spaces | 0.88 | 0.73 | **0.97** | 0.89 | **0.61** | **0.48** | **0.90** | **0.78** | **0.99** | **0.96** | 0.04 | -0.07 | 0.33 | 0.14 | 0.59 | 0.47 | 0.66 | 0.55 |
| **Unified space** | 0.92 | 0.78 | 0.80 | 0.61 | 0.50 | 0.42 | 0.87 | 0.73 | 0.96 | 0.91 | **0.08** | **0.07** | **0.98** | **0.93** | **1.00** | **1.00** | **0.76** | **0.68** |
| DEPICT: Calinski-Harabasz index | | | | | | | | | | | | | | | | | | |
| Raw space | **0.88** | **0.82** | -0.66 | -0.51 | -0.40 | -0.28 | 0.82 | **0.78** | -0.92 | -0.82 | | | | | | | -0.06 | -0.00 |
| Coupled space | 0.88 | 0.82 | -0.96 | -0.91 | -0.37 | -0.22 | 0.79 | 0.73 | -0.92 | -0.82 | | | | | | | -0.11 | -0.08 |
| All spaces | 0.88 | 0.82 | -0.94 | -0.87 | -0.37 | -0.22 | 0.82 | 0.78 | 0.44 | 0.56 | | | | | | | 0.17 | 0.21 |
| **Unified space** | 0.56 | 0.42 | **0.85** | **0.69** | **0.83** | **0.67** | **0.87** | 0.78 | **0.85** | **0.69** | | | | | | | **0.79** | **0.65** |
| DEPICT: Davies-Bouldin index | | | | | | | | | | | | | | | | | | |
| Raw space | -0.82 | -0.64 | **1.00** | **1.00** | 0.03 | -0.11 | -0.50 | -0.33 | **0.92** | **0.82** | | | | | | | 0.13 | 0.15 |
| Coupled space | **0.88** | **0.82** | -0.77 | -0.60 | -0.37 | -0.22 | 0.79 | **0.73** | -0.10 | 0.02 | | | | | | | 0.09 | 0.15 |
| All spaces | 0.48 | 0.42 | 0.90 | 0.78 | 0.47 | 0.33 | **0.85** | 0.73 | 0.92 | 0.82 | | | | | | | 0.72 | **0.62** |
| **Unified space** | 0.81 | 0.60 | 0.71 | 0.56 | **0.82** | **0.72** | 0.70 | 0.51 | 0.64 | 0.42 | | | | | | | **0.73** | 0.56 |
| DEPICT: Silhouette score | | | | | | | | | | | | | | | | | | |
| Raw space | -0.28 | -0.24 | **0.99** | **0.96** | -0.20 | -0.17 | 0.66 | 0.51 | -0.43 | -0.33 | | | | | | | 0.15 | 0.14 |
| Coupled space | 0.87 | 0.78 | -0.64 | -0.51 | -0.37 | -0.22 | 0.79 | 0.73 | -0.12 | -0.02 | | | | | | | 0.11 | 0.15 |
| All spaces | **0.93** | **0.87** | 0.99 | 0.96 | 0.68 | 0.56 | **0.96** | **0.91** | **0.99** | **0.96** | | | | | | | **0.91** | **0.85** |
| **Unified space** | 0.74 | 0.64 | 0.94 | 0.82 | **0.93** | **0.83** | 0.85 | 0.78 | 0.99 | 0.96 | | | | | | | 0.89 | 0.81 |

## F.2 IDENTIFYING THE OPTIMAL NUMBER OF CLUSTERS

We plot the optimal number of clusters $K$ identified by each evaluation approach across different datasets in the experiment for cluster number determination. Results for JULE are shown in Figure A1, and for DEPICT in Figure A2. The ground truth $K$ is represented by a red, outlined, hollow box, while the solid boxes with hatches—colored in light pink, light green, light gray, and steel blue—indicate the $K$ values identified by the approaches using raw space, coupled space, all spaces, and unified space (the proposed method), respectively.

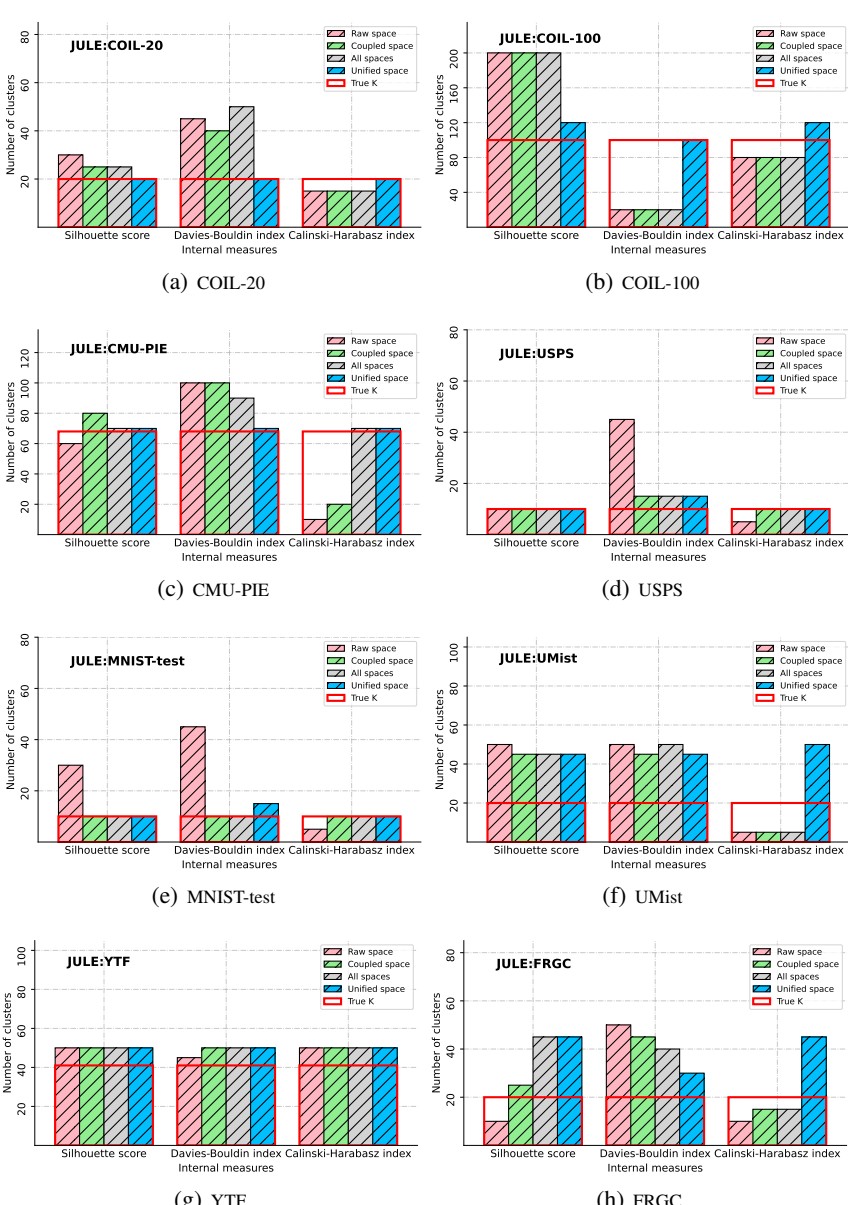

Figure A1: The optimal $K$ identified by each approach for JULE experiment is displayed using bar plots, with the true $K$ indicated by a red, outlined, hollow box.

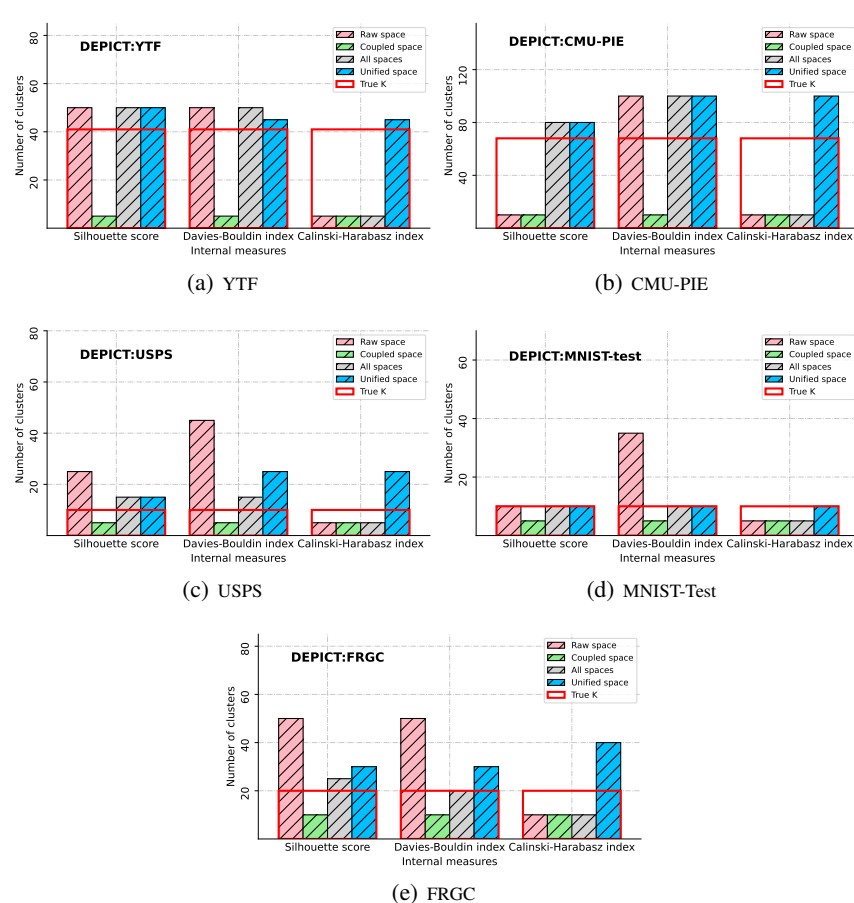

Figure A2: The optimal $K$ identified by each approach for DEPICT experiment is displayed using bar plots, with the true $K$ indicated by a red, outlined, hollow box.

### F.3 EMBEDDING VISUALIZATION

We plot the unified embeddings to visualize the structure of the embedding data in relation to the true clustering groups. Visualizations for the hyperparameter tuning task are shown for JULE in Figure A3 and for DEPICT in Figure A4. For the task of cluster number determination, visualizations are provided for JULE in Figure A5 and for DEPICT in Figure A6.

The unified embedding space effectively separates data points from different clusters across most datasets, including USPS, MNIST-test, CMU-PIE, COIL-20, COIL-100, and YTF for both JULE and DEPICT in the two tasks. Notably, USPS and MNIST-test exhibit well-defined, convex clusters, while COIL-20, COIL-100, and YTF display clusters with more complex, non-convex shapes. We also created t-SNE plots (Van der Maaten & Hinton, 2008), which are well-known for preserving local structure and mapping data to a 2-dimensional feature space, to visualize embedding data from each candidate embedding space (see Supplementary Material). The t-SNE visualizations of individual embedding spaces reveal clusters and patterns consistent with those observed in the unified embedding space. However, for FRGC and UMist, the unified embedding space fails to form clusters corresponding to the true cluster memberships. Upon closer examination of the t-SNE plots for individual clustering outputs in these cases, we found that most candidate spaces struggle to preserve local structure. This suggests that when the candidate spaces fail to maintain local structure, it becomes challenging for the unified embedding space to do so as well.

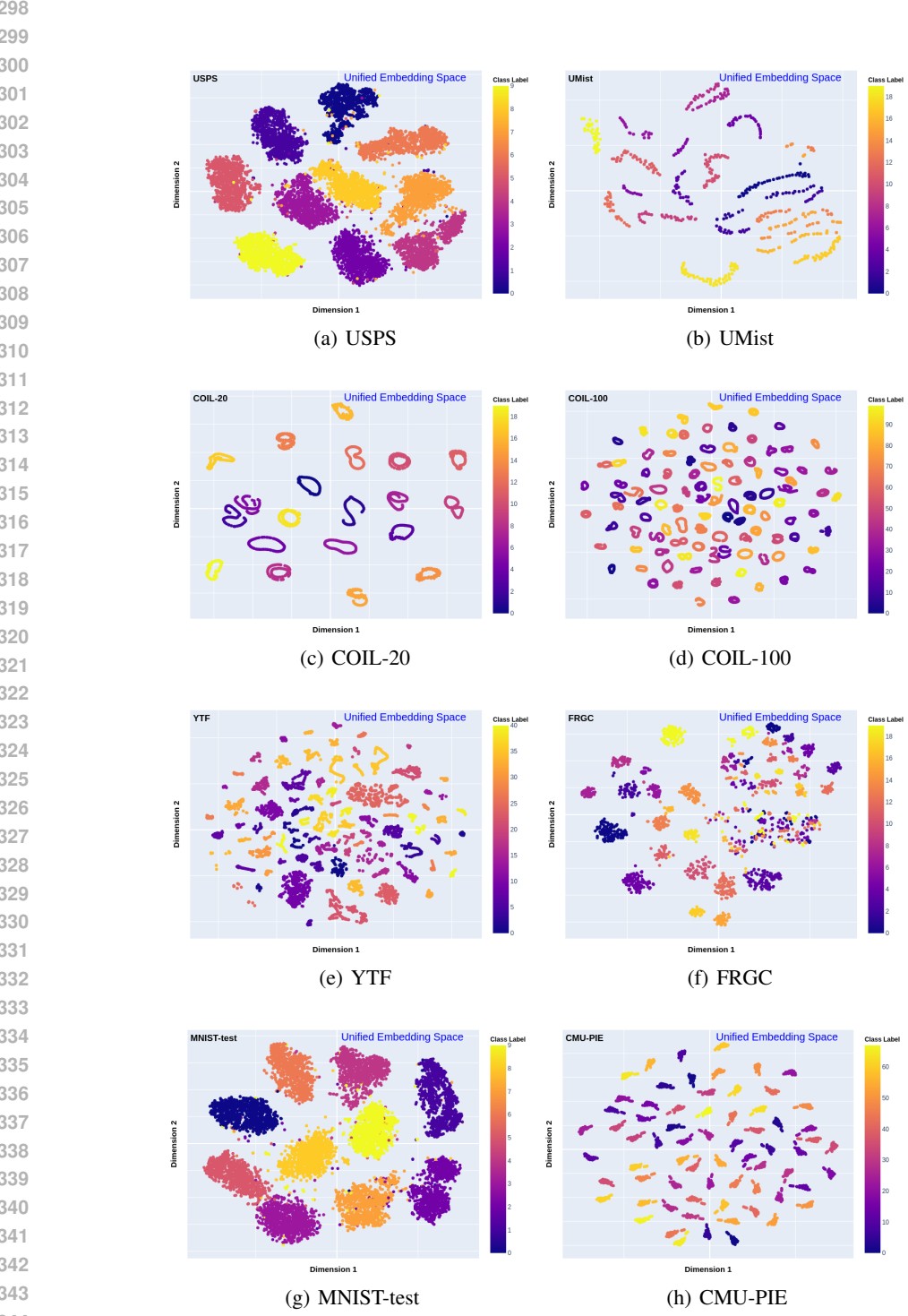

Figure A3: Visualization of low-dimensional embeddings generated by the proposed approach for hyperparameter tuning using JULE.

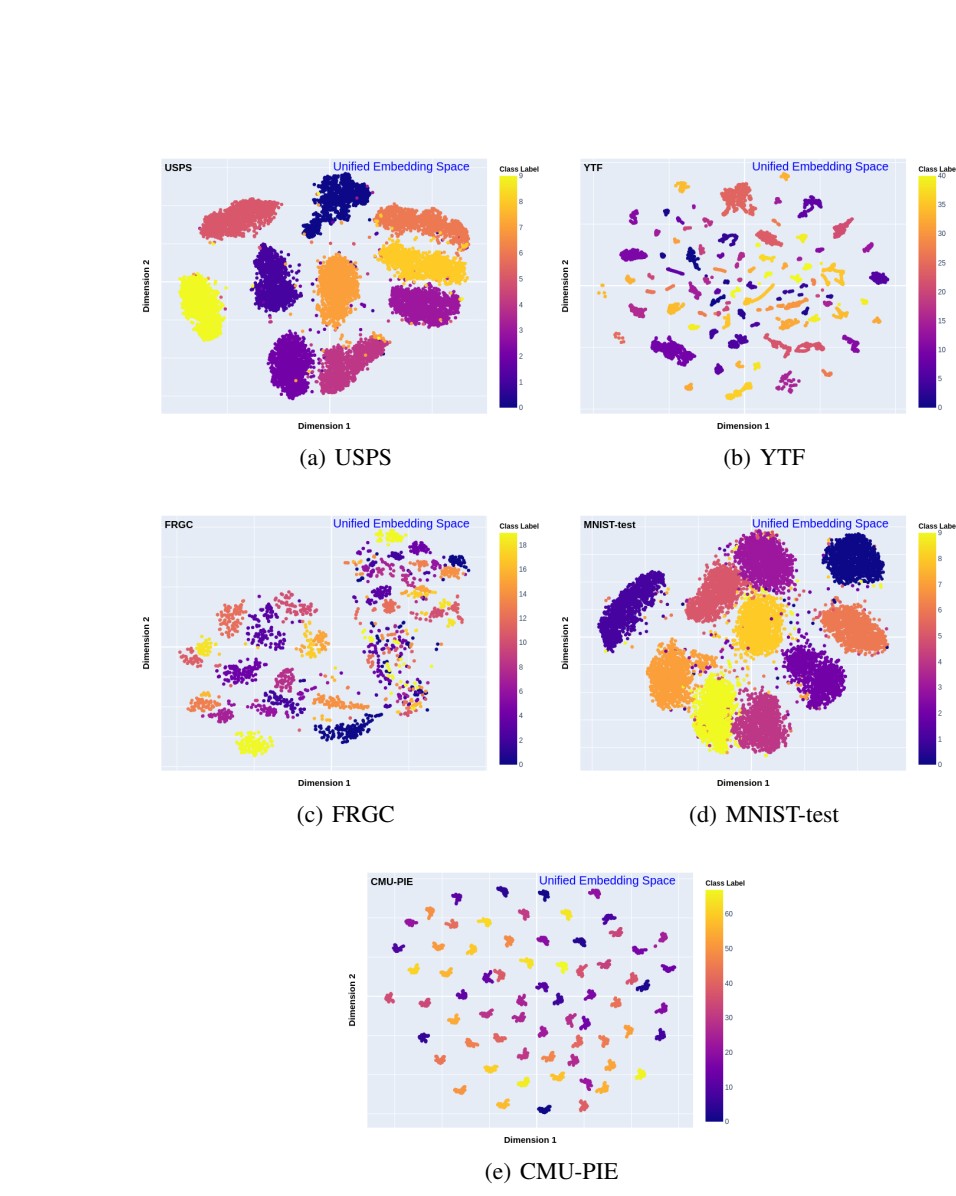

(a) USPS

(b) YTF

(c) FRGC

(d) MNIST-test

(e) CMU-PIE

Figure A4: Visualization of low-dimensional embeddings generated by the proposed approach for for hyperparameter tuning using DEPICT.

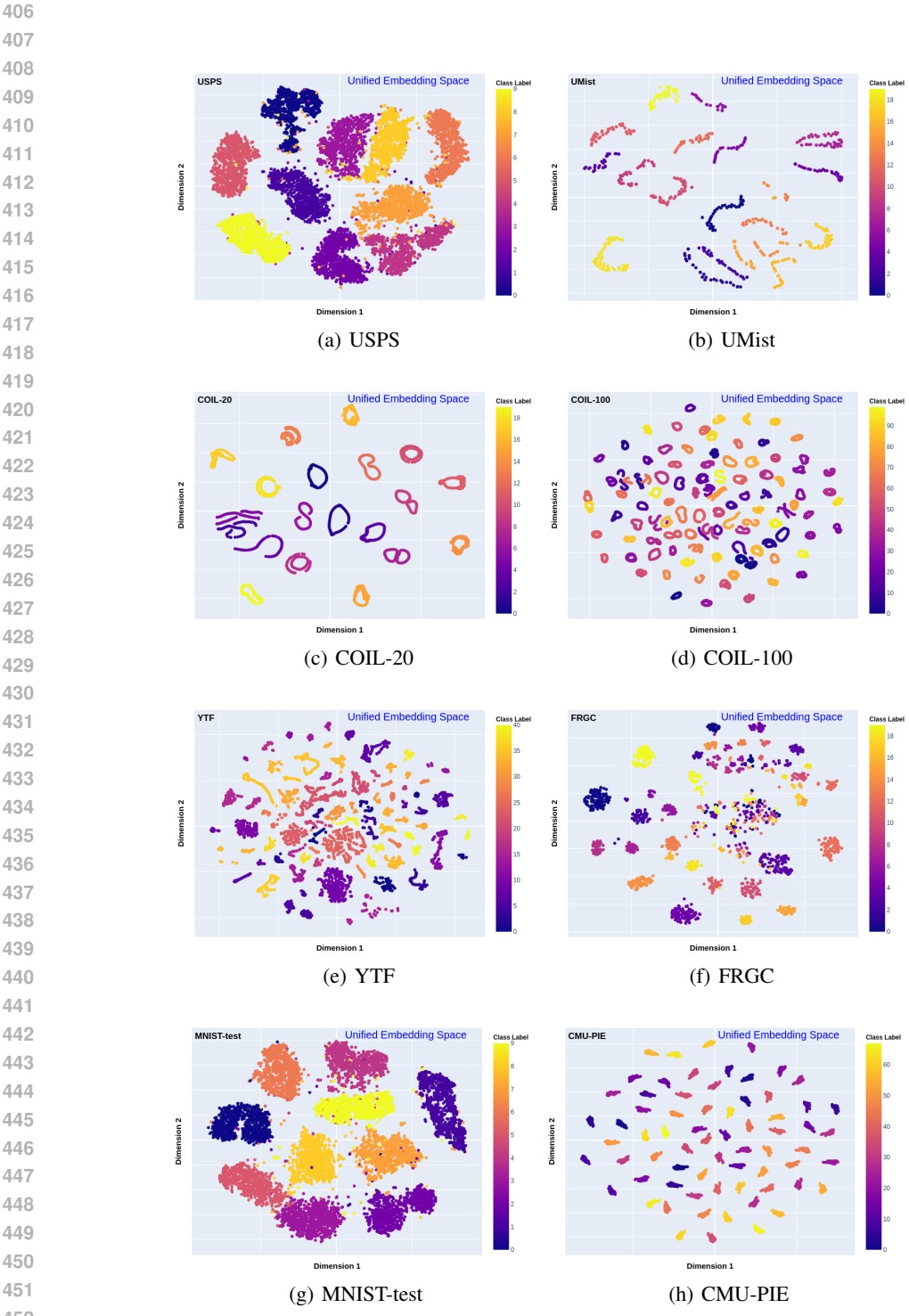

Figure A5: Visualization of low-dimensional embeddings generated by the proposed approach for cluster number determination using JULE.

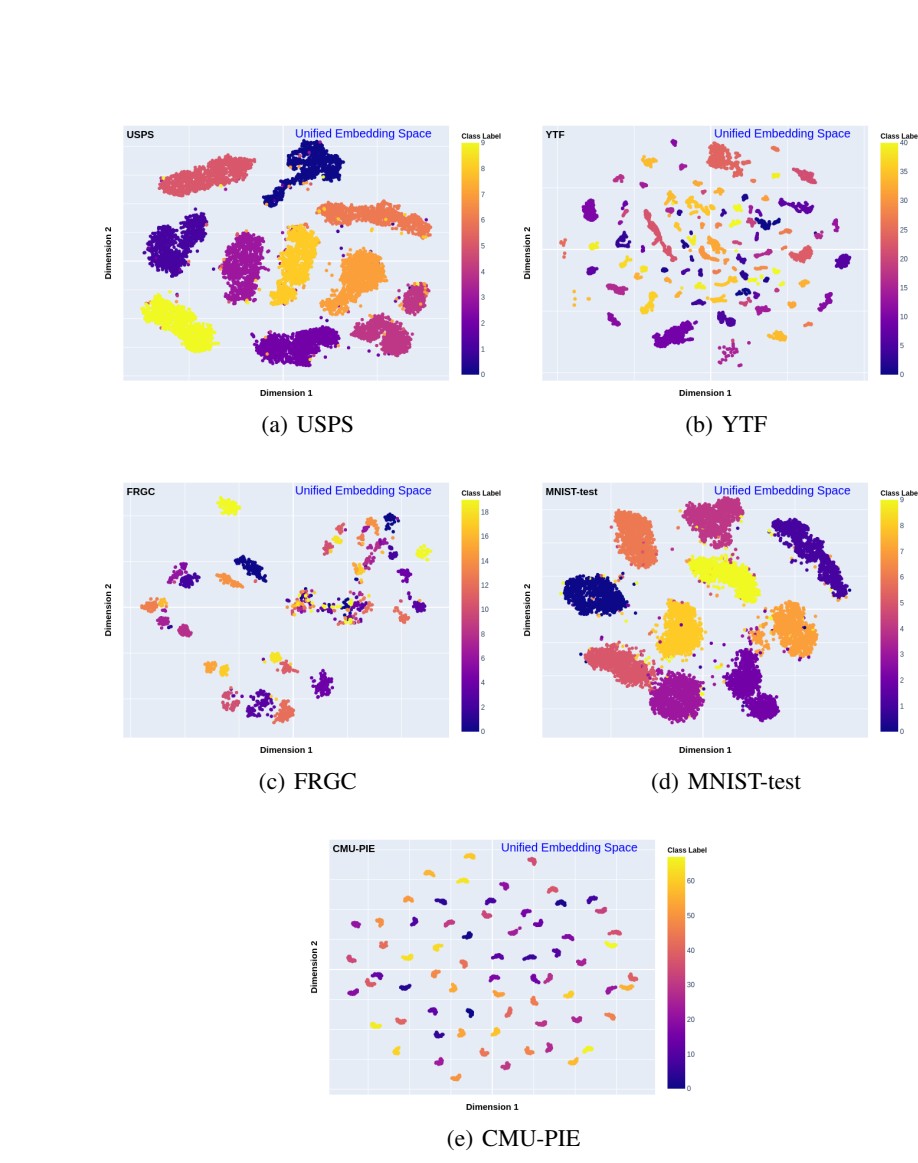

(a) USPS

(b) YTF

(c) FRGC

(d) MNIST-test

(e) CMU-PIE

Figure A6: Visualization of low-dimensional embeddings generated by the proposed approach for for cluster number determination using DEPICT.

## F.4 SENSITIVITY ANALYSIS

**Different perplexity** We explore the impact of selecting different perplexity values, which directly influence $\sigma_i$ when calculating the asymmetric similarity matrix. In addition to the commonly used perplexity value of 30, as reported in the main text, we conducted experiments with values of 5 and 50, representing the lower and upper bounds of the recommended perplexity range (Van der Maaten & Hinton, 2008). The comparative results for the hyperparameter tuning task are presented in Tables A4 and A6, while the results for the cluster number determination task are provided in Tables A5 and A7.

In most cases, we observe that using perplexity values of 30 and 50 yields similar performance, underscoring the robustness of our approach across different perplexity settings. Perplexity values of 5 also produce comparable results to 30 in the majority of instances. However, in certain cases, such as the DEPICT method (evaluated with the Davies-Bouldin index), a perplexity of 5 results in significantly lower rank correlation. This underperformance may stem from the lower perplexity being insufficient to provide each data point with an appropriate neighborhood, thereby hindering the ability to capture the local structure necessary for accurate cluster pattern identification.

| | USPS | | YTF | | FRGC | | MNIST-test | | CMU-PIE | | UMist | | COIL-20 | | COIL-100 | | Average | |
| --- | --- | --- | --- | --- | --- | --- | --- | --- | --- | --- | --- | --- | --- | --- | --- | --- | --- | --- |
| | $r_s$ | $\tau_B$ | $r_s$ | $\tau_B$ | $r_s$ | $\tau_B$ | $r_s$ | $\tau_B$ | $r_s$ | $\tau_B$ | $r_s$ | $\tau_B$ | $r_s$ | $\tau_B$ | $r_s$ | $\tau_B$ | $r_s$ | $\tau_B$ |
| JULE: Calinski-Harabasz index | | | | | | | | | | | | | | | | | | |
| Coupled space | 0.17 | 0.13 | 0.52 | 0.40 | -0.13 | -0.10 | 0.49 | 0.34 | -0.14 | -0.08 | 0.70 | 0.50 | 0.53 | 0.38 | 0.20 | 0.19 | 0.29 | 0.22 |
| Unified space ($perplexity = 5$) | 0.83 | 0.67 | 0.67 | 0.52 | 0.38 | 0.25 | 0.86 | 0.70 | 0.98 | 0.90 | 0.75 | 0.55 | 0.87 | 0.71 | 0.91 | 0.74 | 0.78 | 0.63 |
| Unified space ($perplexity = 30$) | 0.84 | 0.68 | 0.81 | 0.66 | 0.17 | 0.12 | 0.86 | 0.69 | 0.98 | 0.93 | 0.58 | 0.40 | 0.77 | 0.62 | 0.97 | 0.85 | 0.75 | 0.62 |
| Unified space ($perplexity = 50$) | 0.81 | 0.66 | 0.78 | 0.59 | 0.16 | 0.12 | 0.82 | 0.65 | 0.96 | 0.87 | 0.44 | 0.31 | 0.72 | 0.57 | 0.93 | 0.79 | 0.70 | 0.57 |
| JULE: Davies-Bouldin index | | | | | | | | | | | | | | | | | | |
| Coupled space | -0.10 | -0.03 | -0.32 | -0.21 | -0.08 | -0.05 | -0.13 | -0.06 | 0.26 | 0.19 | 0.62 | 0.44 | 0.61 | 0.43 | 0.43 | 0.35 | 0.16 | 0.13 |
| Unified space ($perplexity = 5$) | 0.31 | 0.26 | 0.18 | 0.11 | 0.21 | 0.14 | 0.81 | 0.63 | 0.94 | 0.79 | 0.34 | 0.24 | 0.80 | 0.63 | 0.85 | 0.67 | 0.55 | 0.43 |
| Unified space ($perplexity = 30$) | 0.41 | 0.35 | -0.09 | -0.08 | 0.12 | 0.10 | 0.77 | 0.57 | 0.94 | 0.82 | -0.22 | -0.16 | 0.50 | 0.39 | 0.83 | 0.62 | 0.41 | 0.33 |
| Unified space ($perplexity = 50$) | 0.19 | 0.19 | 0.06 | 0.01 | 0.22 | 0.13 | 0.77 | 0.56 | 0.95 | 0.84 | -0.00 | -0.01 | 0.38 | 0.30 | 0.71 | 0.54 | 0.41 | 0.32 |
| JULE: Silhouette score | | | | | | | | | | | | | | | | | | |
| Coupled space | 0.27 | 0.20 | 0.72 | 0.55 | 0.04 | 0.03 | 0.56 | 0.41 | 0.41 | 0.30 | 0.70 | 0.50 | 0.64 | 0.47 | 0.55 | 0.41 | 0.49 | 0.36 |
| Unified space ($perplexity = 5$) | 0.88 | 0.72 | 0.82 | 0.62 | 0.47 | 0.32 | 0.84 | 0.68 | 0.98 | 0.91 | 0.82 | 0.63 | 0.87 | 0.72 | 0.95 | 0.82 | 0.83 | 0.68 |
| Unified space ($perplexity = 30$) | 0.87 | 0.70 | 0.87 | 0.69 | 0.36 | 0.24 | 0.84 | 0.68 | 0.98 | 0.91 | 0.45 | 0.31 | 0.60 | 0.45 | 0.98 | 0.88 | 0.74 | 0.61 |
| Unified space ($perplexity = 50$) | 0.83 | 0.68 | 0.90 | 0.73 | 0.37 | 0.26 | 0.81 | 0.65 | 0.95 | 0.87 | 0.31 | 0.22 | 0.53 | 0.42 | 0.94 | 0.80 | 0.71 | 0.58 |
| DEPICT: Calinski-Harabasz index | | | | | | | | | | | | | | | | | | |
| Coupled space | 0.76 | 0.57 | 0.44 | 0.26 | 0.76 | 0.57 | 0.89 | 0.72 | 0.49 | 0.44 | | | | | | | 0.67 | 0.51 |
| Unified space ($perplexity = 5$) | 0.92 | 0.78 | 0.68 | 0.52 | 0.75 | 0.54 | 0.95 | 0.84 | -0.86 | -0.69 | | | | | | | 0.49 | 0.40 |
| Unified space ($perplexity = 30$) | 0.95 | 0.84 | 0.65 | 0.52 | 0.89 | 0.75 | 0.96 | 0.84 | 0.95 | 0.80 | | | | | | | 0.88 | 0.75 |
| Unified space ($perplexity = 50$) | 0.95 | 0.84 | 0.72 | 0.57 | 0.89 | 0.74 | 0.96 | 0.86 | 0.93 | 0.82 | | | | | | | 0.89 | 0.76 |
| DEPICT: Davies-Bouldin index | | | | | | | | | | | | | | | | | | |
| Coupled space | 0.81 | 0.59 | 0.45 | 0.31 | 0.90 | 0.74 | 0.89 | 0.72 | 0.63 | 0.59 | | | | | | | 0.73 | 0.59 |
| Unified space ($perplexity = 5$) | 0.62 | 0.48 | 0.55 | 0.41 | 0.24 | 0.19 | 0.87 | 0.72 | -0.94 | -0.83 | | | | | | | 0.27 | 0.19 |
| Unified space ($perplexity = 30$) | 0.92 | 0.78 | 0.60 | 0.42 | 0.81 | 0.66 | 0.92 | 0.80 | 0.99 | 0.92 | | | | | | | 0.85 | 0.72 |
| Unified space ($perplexity = 50$) | 0.93 | 0.79 | 0.64 | 0.48 | 0.84 | 0.72 | 0.86 | 0.74 | 0.95 | 0.84 | | | | | | | 0.85 | 0.71 |
| DEPICT: Silhouette score | | | | | | | | | | | | | | | | | | |
| Coupled space | 0.73 | 0.50 | 0.47 | 0.36 | 0.79 | 0.65 | 0.86 | 0.69 | 0.59 | 0.52 | | | | | | | 0.69 | 0.54 |
| Unified space ($perplexity = 5$) | 0.92 | 0.79 | 0.74 | 0.59 | 0.89 | 0.77 | 0.93 | 0.83 | 0.85 | 0.71 | | | | | | | 0.87 | 0.74 |
| Unified space ($perplexity = 30$) | 0.98 | 0.91 | 0.78 | 0.59 | 0.95 | 0.84 | 0.97 | 0.90 | 0.97 | 0.88 | | | | | | | 0.93 | 0.82 |
| Unified space ($perplexity = 50$) | 0.97 | 0.90 | 0.74 | 0.62 | 0.94 | 0.84 | 0.96 | 0.88 | 0.92 | 0.80 | | | | | | | 0.91 | 0.81 |

Table A4: The results of the sensitivity analysis regarding the choice of perplexity in the hyperparameter tuning experiment are presented. $r_s$ and $\tau_B$ between the generated scores and NMI scores are reported. The results obtained using coupled space are presented as a baseline for comparison.

| | USPS | | YTF | | FRGC | | MNIST-test | | CMU-PIE | | UMist | | COIL-20 | | COIL-100 | | Average | |
|---|---|---|---|---|---|---|---|---|---|---|---|---|---|---|---|---|---|---|
| | $r_s$ | $\tau_B$ | $r_s$ | $\tau_B$ | $r_s$ | $\tau_B$ | $r_s$ | $\tau_B$ | $r_s$ | $\tau_B$ | $r_s$ | $\tau_B$ | $r_s$ | $\tau_B$ | $r_s$ | $\tau_B$ | $r_s$ | $\tau_B$ |
| JULE: Calinski-Harabasz index | | | | | | | | | | | | | | | | | | |
| Coupled space | 0.65 | 0.64 | 0.1 | 0.06 | -0.93 | -0.83 | 0.64 | 0.6 | -0.03 | -0.02 | -0.13 | -0.07 | 0.76 | 0.71 | 0.74 | 0.56 | 0.22 | 0.21 |
| Unified space ($perplexity = 5$) | 0.95 | 0.87 | 0.9 | 0.72 | 0.92 | 0.83 | 0.94 | 0.82 | 0.99 | 0.96 | 0.54 | 0.38 | 0.83 | 0.71 | 0.79 | 0.64 | 0.86 | 0.74 |
| Unified space ($perplexity = 30$) | 0.98 | 0.91 | 1.0 | 1.0 | 0.83 | 0.67 | 0.96 | 0.87 | 0.95 | 0.87 | 0.43 | 0.24 | 0.83 | 0.71 | 0.61 | 0.51 | 0.82 | 0.72 |
| Unified space ($perplexity = 50$) | 0.98 | 0.91 | 0.92 | 0.78 | 0.87 | 0.72 | 0.96 | 0.87 | 0.95 | 0.87 | -0.04 | 0.02 | 0.83 | 0.71 | 0.54 | 0.38 | 0.75 | 0.66 |
| JULE: Davies-Bouldin index | | | | | | | | | | | | | | | | | | |
| Coupled space | 0.54 | 0.38 | 0.15 | 0.17 | 0.85 | 0.67 | 0.43 | 0.29 | 0.78 | 0.56 | -0.08 | 0.02 | -0.26 | -0.14 | -0.9 | -0.78 | 0.19 | 0.15 |
| Unified space ($perplexity = 5$) | 0.84 | 0.69 | -0.82 | -0.67 | 0.8 | 0.67 | 0.76 | 0.6 | 0.73 | 0.51 | 0.2 | 0.07 | 0.41 | 0.36 | 0.53 | 0.33 | 0.43 | 0.32 |
| Unified space ($perplexity = 30$) | 0.47 | 0.33 | 0.55 | 0.39 | 0.18 | 0.17 | 0.54 | 0.47 | 0.92 | 0.82 | -0.28 | -0.2 | 0.43 | 0.43 | 0.9 | 0.78 | 0.46 | 0.40 |
| Unified space ($perplexity = 50$) | 0.52 | 0.38 | -0.13 | 0.0 | -0.67 | -0.56 | 0.42 | 0.33 | 0.78 | 0.6 | -0.38 | -0.33 | 0.69 | 0.57 | 0.69 | 0.56 | 0.24 | 0.19 |
| JULE: Silhouette score | | | | | | | | | | | | | | | | | | |
| Coupled space | 0.85 | 0.73 | 0.33 | 0.28 | 0.72 | 0.61 | 0.88 | 0.69 | 0.96 | 0.87 | 0.07 | 0.16 | 0.55 | 0.43 | 0.44 | 0.29 | 0.60 | 0.51 |
| Unified space ($perplexity = 5$) | 0.82 | 0.69 | 0.78 | 0.67 | 0.7 | 0.61 | 0.88 | 0.73 | 0.99 | 0.96 | 0.61 | 0.47 | 0.81 | 0.64 | 0.9 | 0.78 | 0.81 | 0.69 |
| Unified space ($perplexity = 30$) | 0.84 | 0.69 | 0.87 | 0.72 | 0.63 | 0.5 | 0.92 | 0.78 | 0.99 | 0.96 | 0.42 | 0.29 | 0.93 | 0.86 | 0.95 | 0.87 | 0.82 | 0.71 |
| Unified space ($perplexity = 50$) | 0.89 | 0.73 | 0.98 | 0.94 | 0.68 | 0.56 | 0.93 | 0.78 | 0.99 | 0.96 | -0.12 | -0.11 | 0.93 | 0.86 | 0.99 | 0.96 | 0.78 | 0.71 |
| DEPICT: Calinski-Harabasz index | | | | | | | | | | | | | | | | | | |
| Coupled space | 0.46 | 0.6 | -0.99 | -0.96 | -0.85 | -0.72 | 0.44 | 0.56 | -0.92 | -0.82 | | | | | | | -0.37 | -0.27 |
| Unified space ($perplexity = 5$) | 0.93 | 0.87 | 0.6 | 0.47 | 0.62 | 0.44 | 1.0 | 1.0 | 0.83 | 0.69 | | | | | | | 0.80 | 0.69 |
| Unified space ($perplexity = 30$) | 0.77 | 0.64 | 0.89 | 0.73 | 0.73 | 0.61 | 0.99 | 0.96 | 0.85 | 0.69 | | | | | | | 0.85 | 0.73 |
| Unified space ($perplexity = 50$) | 0.83 | 0.69 | 0.69 | 0.51 | 0.75 | 0.61 | 0.99 | 0.96 | 0.88 | 0.73 | | | | | | | 0.83 | 0.70 |
| DEPICT: Davies-Bouldin index | | | | | | | | | | | | | | | | | | |
| Coupled space | 0.46 | 0.6 | -0.78 | -0.64 | -0.85 | -0.72 | 0.44 | 0.56 | -0.1 | 0.02 | | | | | | | -0.17 | -0.04 |
| Unified space ($perplexity = 5$) | 0.7 | 0.51 | 0.01 | -0.02 | -0.02 | 0.0 | 0.95 | 0.87 | 0.88 | 0.73 | | | | | | | 0.50 | 0.42 |
| Unified space ($perplexity = 30$) | 0.84 | 0.64 | 0.73 | 0.6 | 0.27 | 0.22 | 0.83 | 0.69 | 0.64 | 0.42 | | | | | | | 0.66 | 0.51 |
| Unified space ($perplexity = 50$) | 0.32 | 0.29 | 0.73 | 0.6 | 0.27 | 0.17 | 0.79 | 0.69 | 0.48 | 0.29 | | | | | | | 0.52 | 0.41 |
| DEPICT: Silhouette score | | | | | | | | | | | | | | | | | | |
| Coupled space | 0.44 | 0.56 | -0.61 | -0.47 | -0.85 | -0.72 | 0.44 | 0.56 | -0.12 | -0.02 | | | | | | | -0.14 | -0.02 |
| Unified space ($perplexity = 5$) | 0.77 | 0.64 | 0.66 | 0.56 | 0.43 | 0.33 | 0.98 | 0.91 | 0.95 | 0.87 | | | | | | | 0.76 | 0.66 |
| Unified space ($perplexity = 30$) | 0.93 | 0.87 | 0.95 | 0.87 | 0.55 | 0.44 | 0.99 | 0.96 | 0.99 | 0.96 | | | | | | | 0.88 | 0.82 |
| Unified space ($perplexity = 50$) | 0.74 | 0.64 | 0.99 | 0.96 | 0.68 | 0.61 | 0.99 | 0.96 | 0.98 | 0.91 | | | | | | | 0.88 | 0.82 |

Table A5: The results of the sensitivity analysis regarding the choice of perplexity in the cluster number determination experiment are presented. $r_s$ and $\tau_B$ between the generated scores and NMI scores are reported. The results obtained using coupled space are presented as a baseline for comparison.

| | USPS | | YTF | | FRGC | | MNIST-test | | CMU-PIE | | UMist | | COIL-20 | | COIL-100 | | Average | |
|---|---|---|---|---|---|---|---|---|---|---|---|---|---|---|---|---|---|---|
| | $r_s$ | $\tau_B$ | $r_s$ | $\tau_B$ | $r_s$ | $\tau_B$ | $r_s$ | $\tau_B$ | $r_s$ | $\tau_B$ | $r_s$ | $\tau_B$ | $r_s$ | $\tau_B$ | $r_s$ | $\tau_B$ | $r_s$ | $\tau_B$ |
| JULE: Calinski-Harabasz index | | | | | | | | | | | | | | | | | | |
| Coupled space | 0.04 | 0.05 | 0.39 | 0.27 | -0.26 | -0.18 | 0.31 | 0.21 | -0.20 | -0.12 | 0.64 | 0.45 | 0.57 | 0.40 | 0.09 | 0.08 | 0.20 | 0.14 |
| Unified space ($perplexity = 5$) | 0.88 | 0.71 | 0.54 | 0.43 | 0.34 | 0.21 | 0.92 | 0.78 | 0.98 | 0.91 | 0.72 | 0.50 | 0.83 | 0.65 | 0.88 | 0.72 | 0.76 | 0.61 |
| Unified space ($perplexity = 30$) | 0.88 | 0.71 | 0.58 | 0.43 | 0.21 | 0.14 | 0.94 | 0.80 | 0.98 | 0.90 | 0.62 | 0.42 | 0.73 | 0.55 | 0.92 | 0.77 | 0.73 | 0.59 |
| Unified space ($perplexity = 50$) | 0.85 | 0.69 | 0.63 | 0.46 | 0.13 | 0.07 | 0.90 | 0.73 | 0.95 | 0.83 | 0.41 | 0.27 | 0.71 | 0.54 | 0.90 | 0.74 | 0.69 | 0.54 |
| JULE: Davies-Bouldin index | | | | | | | | | | | | | | | | | | |
| Coupled space | -0.27 | -0.15 | -0.14 | -0.09 | -0.23 | -0.14 | -0.35 | -0.19 | 0.20 | 0.16 | 0.53 | 0.36 | 0.63 | 0.44 | 0.33 | 0.26 | 0.09 | 0.08 |
| Unified space ($perplexity = 5$) | 0.17 | 0.19 | 0.11 | 0.09 | 0.64 | 0.46 | 0.86 | 0.73 | 0.94 | 0.79 | 0.29 | 0.21 | 0.78 | 0.57 | 0.84 | 0.65 | 0.58 | 0.46 |
| Unified space ($perplexity = 30$) | 0.28 | 0.27 | -0.21 | -0.14 | 0.53 | 0.37 | 0.89 | 0.71 | 0.94 | 0.82 | -0.28 | -0.21 | 0.48 | 0.37 | 0.75 | 0.56 | 0.42 | 0.34 |
| Unified space ($perplexity = 50$) | 0.07 | 0.11 | 0.07 | 0.03 | 0.36 | 0.24 | 0.86 | 0.67 | 0.95 | 0.84 | -0.14 | -0.11 | 0.37 | 0.31 | 0.68 | 0.51 | 0.40 | 0.33 |
| JULE: Silhouette score | | | | | | | | | | | | | | | | | | |
| Coupled space | 0.14 | 0.12 | 0.54 | 0.39 | -0.08 | -0.02 | 0.41 | 0.27 | 0.36 | 0.27 | 0.64 | 0.46 | 0.67 | 0.48 | 0.44 | 0.31 | 0.39 | 0.28 |
| Unified space ($perplexity = 5$) | 0.94 | 0.80 | 0.83 | 0.64 | 0.71 | 0.53 | 0.91 | 0.76 | 0.98 | 0.91 | 0.79 | 0.60 | 0.84 | 0.66 | 0.96 | 0.85 | 0.87 | 0.72 |
| Unified space ($perplexity = 30$) | 0.93 | 0.79 | 0.80 | 0.63 | 0.72 | 0.53 | 0.94 | 0.80 | 0.98 | 0.90 | 0.45 | 0.29 | 0.57 | 0.39 | 0.91 | 0.76 | 0.74 | 0.64 |
| Unified space ($perplexity = 50$) | 0.87 | 0.71 | 0.81 | 0.63 | 0.70 | 0.51 | 0.91 | 0.75 | 0.95 | 0.86 | 0.23 | 0.18 | 0.52 | 0.40 | 0.90 | 0.72 | 0.74 | 0.59 |
| DEPICT: Calinski-Harabasz index | | | | | | | | | | | | | | | | | | |
| Coupled space | 0.56 | 0.40 | 0.54 | 0.35 | 0.76 | 0.57 | 0.88 | 0.69 | 0.48 | 0.43 | | | | | | | 0.64 | 0.49 |
| Unified space ($perplexity = 5$) | 0.78 | 0.61 | 0.64 | 0.48 | 0.80 | 0.59 | 0.96 | 0.88 | -0.87 | -0.71 | | | | | | | 0.46 | 0.37 |
| Unified space ($perplexity = 30$) | 0.87 | 0.70 | 0.57 | 0.42 | 0.93 | 0.80 | 0.96 | 0.88 | 0.95 | 0.81 | | | | | | | 0.86 | 0.72 |
| Unified space ($perplexity = 50$) | 0.87 | 0.70 | 0.60 | 0.48 | 0.94 | 0.84 | 0.96 | 0.87 | 0.92 | 0.79 | | | | | | | 0.86 | 0.73 |
| DEPICT: Davies-Bouldin index | | | | | | | | | | | | | | | | | | |
| Coupled space | 0.61 | 0.42 | 0.48 | 0.32 | 0.92 | 0.74 | 0.88 | 0.69 | 0.62 | 0.56 | | | | | | | 0.70 | 0.55 |
| Unified space ($perplexity = 5$) | 0.55 | 0.44 | 0.43 | 0.29 | 0.29 | 0.24 | 0.91 | 0.79 | -0.94 | -0.85 | | | | | | | 0.25 | 0.18 |
| Unified space ($perplexity = 30$) | 0.85 | 0.71 | 0.50 | 0.33 | 0.72 | 0.56 | 0.92 | 0.82 | 0.98 | 0.91 | | | | | | | 0.79 | 0.66 |
| Unified space ($perplexity = 50$) | 0.90 | 0.75 | 0.51 | 0.36 | 0.84 | 0.67 | 0.85 | 0.75 | 0.93 | 0.81 | | | | | | | 0.81 | 0.67 |
| DEPICT: Silhouette score | | | | | | | | | | | | | | | | | | |
| Coupled space | 0.52 | 0.33 | 0.57 | 0.45 | 0.80 | 0.62 | 0.85 | 0.65 | 0.59 | 0.48 | | | | | | | 0.67 | 0.51 |
| Unified space ($perplexity = 5$) | 0.81 | 0.65 | 0.63 | 0.48 | 0.91 | 0.77 | 0.96 | 0.90 | 0.86 | 0.73 | | | | | | | 0.83 | 0.70 |
| Unified space ($perplexity = 30$) | 0.88 | 0.74 | 0.69 | 0.53 | 0.95 | 0.84 | 0.96 | 0.88 | 0.96 | 0.87 | | | | | | | 0.89 | 0.77 |
| Unified space ($perplexity = 50$) | 0.89 | 0.75 | 0.60 | 0.48 | 0.94 | 0.82 | 0.96 | 0.90 | 0.91 | 0.77 | | | | | | | 0.86 | 0.74 |

Table A6: The results of the sensitivity analysis regarding the choice of perplexity in the hyperparameter tuning experiment are presented. $r_s$ and $\tau_B$ between the generated scores and ACC scores are reported. The results obtained using coupled space are presented as a baseline for comparison.

| | USPS | | YTF | | FRGC | | MNIST-test | | CMU-PIE | | UMist | | COIL-20 | | COIL-100 | | Average | |
|---|---|---|---|---|---|---|---|---|---|---|---|---|---|---|---|---|---|---|
| | $r_s$ | $\tau_B$ | $r_s$ | $\tau_B$ | $r_s$ | $\tau_B$ | $r_s$ | $\tau_B$ | $r_s$ | $\tau_B$ | $r_s$ | $\tau_B$ | $r_s$ | $\tau_B$ | $r_s$ | $\tau_B$ | $r_s$ | $\tau_B$ |
| JULE: Calinski-Harabasz index | | | | | | | | | | | | | | | | | | |
| Coupled space | 0.84 | 0.73 | 0.03 | -0.06 | -0.49 | -0.31 | 0.61 | 0.56 | -0.09 | -0.07 | -0.04 | 0.07 | 0.74 | 0.64 | 0.60 | 0.51 | 0.27 | 0.26 |
| Unified space ($perplexity = 5$) | 0.85 | 0.69 | 0.90 | 0.72 | 0.56 | 0.42 | 0.90 | 0.78 | 1.00 | 1.00 | 0.28 | 0.24 | 0.81 | 0.64 | 0.87 | 0.78 | 0.77 | 0.66 |
| Unified space ($perplexity = 30$) | 0.88 | 0.73 | 0.95 | 0.89 | 0.37 | 0.37 | 0.94 | 0.82 | 0.96 | 0.91 | 0.19 | 0.11 | 0.81 | 0.64 | 0.77 | 0.64 | 0.73 | 0.64 |
| Unified space ($perplexity = 50$) | 0.88 | 0.73 | 0.88 | 0.78 | 0.34 | 0.31 | 0.94 | 0.82 | 0.96 | 0.91 | -0.14 | -0.11 | 0.79 | 0.64 | 0.64 | 0.51 | 0.66 | 0.57 |
| JULE: Davies-Bouldin index | | | | | | | | | | | | | | | | | | |
| Coupled space | 0.39 | 0.29 | 0.10 | 0.06 | 0.37 | 0.25 | 0.49 | 0.33 | 0.83 | 0.60 | -0.28 | -0.29 | -0.29 | -0.21 | -0.87 | -0.73 | 0.09 | 0.04 |
| Unified space ($perplexity = 5$) | 0.69 | 0.51 | -0.87 | -0.78 | 0.62 | 0.48 | 0.72 | 0.56 | 0.78 | 0.56 | -0.07 | -0.07 | 0.36 | 0.29 | 0.60 | 0.38 | 0.35 | 0.24 |
| Unified space ($perplexity = 30$) | 0.53 | 0.42 | 0.43 | 0.28 | 0.49 | 0.37 | 0.53 | 0.42 | 0.93 | 0.87 | -0.58 | -0.33 | 0.41 | 0.36 | 0.85 | 0.64 | 0.45 | 0.38 |
| Unified space ($perplexity = 50$) | 0.55 | 0.47 | -0.18 | -0.11 | -0.50 | -0.37 | 0.39 | 0.29 | 0.79 | 0.64 | -0.71 | -0.56 | 0.59 | 0.50 | 0.53 | 0.42 | 0.18 | 0.16 |
| JULE: Silhouette score | | | | | | | | | | | | | | | | | | |
| Coupled space | 0.93 | 0.82 | 0.30 | 0.28 | 0.21 | 0.09 | 0.82 | 0.64 | 0.98 | 0.91 | -0.13 | -0.16 | 0.52 | 0.36 | 0.55 | 0.42 | 0.52 | 0.42 |
| Unified space ($perplexity = 5$) | 0.71 | 0.51 | 0.70 | 0.56 | 0.64 | 0.54 | 0.82 | 0.69 | 1.00 | 1.00 | 0.32 | 0.24 | 0.83 | 0.71 | 0.98 | 0.91 | 0.75 | 0.65 |
| Unified space ($perplexity = 30$) | 0.92 | 0.78 | 0.80 | 0.61 | 0.50 | 0.42 | 0.87 | 0.73 | 0.96 | 0.91 | 0.08 | 0.07 | 0.98 | 0.93 | 1.00 | 1.00 | 0.76 | 0.68 |
| Unified space ($perplexity = 50$) | 0.94 | 0.82 | 0.92 | 0.83 | 0.54 | 0.48 | 0.88 | 0.73 | 0.96 | 0.91 | -0.33 | -0.24 | 0.98 | 0.93 | 0.98 | 0.91 | 0.73 | 0.67 |
| DEPICT: Calinski-Harabasz index | | | | | | | | | | | | | | | | | | |
| Coupled space | 0.88 | 0.82 | -0.96 | -0.91 | -0.37 | -0.22 | 0.79 | 0.73 | -0.92 | -0.82 | | | | | | | -0.11 | -0.08 |
| Unified space ($perplexity = 5$) | 0.74 | 0.64 | 0.58 | 0.42 | 0.80 | 0.72 | 0.88 | 0.82 | 0.83 | 0.69 | | | | | | | 0.77 | 0.66 |
| Unified space ($perplexity = 30$) | 0.56 | 0.42 | 0.85 | 0.69 | 0.83 | 0.67 | 0.87 | 0.78 | 0.85 | 0.69 | | | | | | | 0.79 | 0.65 |
| Unified space ($perplexity = 50$) | 0.71 | 0.56 | 0.70 | 0.56 | 0.82 | 0.67 | 0.87 | 0.78 | 0.88 | 0.73 | | | | | | | 0.79 | 0.66 |
| DEPICT: Davies-Bouldin index | | | | | | | | | | | | | | | | | | |
| Coupled space | 0.88 | 0.82 | -0.77 | -0.60 | -0.37 | -0.22 | 0.79 | 0.73 | -0.10 | 0.02 | | | | | | | 0.09 | 0.15 |
| Unified space ($perplexity = 5$) | 0.56 | 0.38 | -0.03 | -0.07 | 0.25 | 0.17 | 0.82 | 0.69 | 0.88 | 0.73 | | | | | | | 0.50 | 0.38 |
| Unified space ($perplexity = 30$) | 0.81 | 0.60 | 0.71 | 0.56 | 0.82 | 0.72 | 0.70 | 0.51 | 0.64 | 0.42 | | | | | | | 0.73 | 0.56 |
| Unified space ($perplexity = 50$) | 0.74 | 0.51 | 0.72 | 0.56 | 0.68 | 0.56 | 0.72 | 0.60 | 0.48 | 0.29 | | | | | | | 0.67 | 0.50 |
| DEPICT: Silhouette score | | | | | | | | | | | | | | | | | | |
| Coupled space | 0.87 | 0.78 | -0.64 | -0.51 | -0.37 | -0.22 | 0.79 | 0.73 | -0.12 | -0.02 | | | | | | | 0.11 | 0.15 |
| Unified space ($perplexity = 5$) | 0.56 | 0.42 | 0.65 | 0.51 | 0.73 | 0.61 | 0.84 | 0.73 | 0.95 | 0.87 | | | | | | | 0.75 | 0.63 |
| Unified space ($perplexity = 30$) | 0.74 | 0.64 | 0.94 | 0.82 | 0.93 | 0.83 | 0.85 | 0.78 | 0.99 | 0.96 | | | | | | | 0.89 | 0.81 |
| Unified space ($perplexity = 50$) | 0.93 | 0.87 | 0.98 | 0.91 | 0.90 | 0.78 | 0.85 | 0.78 | 0.98 | 0.91 | | | | | | | 0.93 | 0.85 |

Table A7: The results of the sensitivity analysis regarding the choice of perplexity in the cluster number determination experiment are presented. $r_s$ and $\tau_B$ between the generated scores and ACC scores are reported. The results obtained from using coupled space are presented as a baseline for comparison.

**Different dimension**   In our main experiments, we set the dimensionality of the low-dimensional space to two, consistent with typical implementations of t-SNE. We chose this value because increasing the dimensionality can distort the local structure between data points. To assess the effects of higher dimensionality, we conducted additional experiments with dimensions of $4, 8, 16, 32,$ and $128$, alongside the original two-dimensional setting. The comparative results for hyperparameter tuning are presented in Tables A8 and A10, while the results for determining the number of clusters are reported in Tables A9 and A11.

Across the experiments, we found that dimensionalities between four and eight produced very similar performance, indicating that as long as the dimensionality remains low, its exact value has minimal impact on validation. However, when the dimensionality increased to 16, the rank correlation dropped significantly in some cases, confirming our hypothesis that a higher number of dimensions can distort the local structure of the data.

| | USPS $r_s$ | USPS $\tau_B$ | YTF $r_s$ | YTF $\tau_B$ | FRGC $r_s$ | FRGC $\tau_B$ | MNIST-test $r_s$ | MNIST-test $\tau_B$ | CMU-PIE $r_s$ | CMU-PIE $\tau_B$ | UMist $r_s$ | UMist $\tau_B$ | COIL-20 $r_s$ | COIL-20 $\tau_B$ | COIL-100 $r_s$ | COIL-100 $\tau_B$ | Average $r_s$ | Average $\tau_B$ |
|---|---|---|---|---|---|---|---|---|---|---|---|---|---|---|---|---|---|---|
| **JULE: Calinski-Harabasz index** | | | | | | | | | | | | | | | | | | |
| Coupled space | 0.17 | 0.13 | 0.52 | 0.40 | -0.13 | -0.10 | 0.49 | 0.34 | -0.14 | -0.08 | 0.70 | 0.50 | 0.53 | 0.38 | 0.20 | 0.19 | 0.29 | 0.22 |
| Unified space ($dim = 128$) | 0.61 | 0.48 | 0.93 | 0.79 | 0.30 | 0.22 | 0.87 | 0.73 | 0.25 | 0.22 | 0.59 | 0.42 | 0.57 | 0.40 | 0.90 | 0.74 | 0.63 | 0.50 |
| Unified space ($dim = 32$) | 0.62 | 0.50 | 0.93 | 0.79 | 0.31 | 0.23 | 0.86 | 0.73 | 0.43 | 0.37 | 0.54 | 0.37 | 0.47 | 0.34 | 0.90 | 0.73 | 0.63 | 0.51 |
| Unified space ($dim = 16$) | 0.63 | 0.50 | 0.95 | 0.83 | 0.43 | 0.33 | 0.87 | 0.73 | 0.93 | 0.79 | 0.56 | 0.40 | 0.46 | 0.34 | 0.91 | 0.75 | 0.72 | 0.58 |
| Unified space ($dim = 8$) | 0.64 | 0.53 | 0.96 | 0.86 | 0.44 | 0.32 | 0.87 | 0.74 | 0.97 | 0.86 | 0.60 | 0.42 | 0.33 | 0.23 | 0.92 | 0.76 | 0.71 | 0.59 |
| Unified space ($dim = 4$) | 0.76 | 0.60 | 0.94 | 0.79 | 0.45 | 0.34 | 0.86 | 0.71 | 0.98 | 0.93 | 0.53 | 0.37 | 0.28 | 0.22 | 0.97 | 0.86 | 0.72 | 0.60 |
| Unified space ($dim = 2$) | 0.84 | 0.68 | 0.81 | 0.66 | 0.17 | 0.12 | 0.86 | 0.69 | 0.98 | 0.93 | 0.58 | 0.40 | 0.77 | 0.62 | 0.97 | 0.85 | 0.75 | 0.62 |
| **JULE: Davies-Bouldin index** | | | | | | | | | | | | | | | | | | |
| Coupled space | -0.10 | -0.03 | -0.32 | -0.21 | -0.08 | -0.05 | -0.13 | -0.06 | 0.26 | 0.19 | 0.62 | 0.44 | 0.61 | 0.43 | 0.43 | 0.35 | 0.16 | 0.13 |
| Unified space ($dim = 128$) | -0.11 | -0.08 | -0.50 | -0.37 | 0.03 | 0.02 | 0.40 | 0.27 | -0.15 | -0.17 | -0.41 | -0.29 | -0.13 | -0.04 | -0.64 | -0.48 | -0.19 | -0.14 |
| Unified space ($dim = 32$) | -0.08 | -0.04 | -0.69 | -0.53 | -0.01 | 0.00 | 0.41 | 0.28 | 0.19 | 0.13 | -0.39 | -0.26 | -0.30 | -0.21 | -0.66 | -0.50 | -0.19 | -0.14 |
| Unified space ($dim = 16$) | -0.10 | -0.06 | -0.56 | -0.42 | 0.06 | 0.06 | 0.48 | 0.34 | 0.79 | 0.65 | -0.21 | -0.15 | -0.19 | -0.15 | -0.41 | -0.32 | -0.02 | -0.01 |
| Unified space ($dim = 8$) | -0.18 | -0.09 | -0.65 | -0.47 | 0.08 | 0.04 | 0.69 | 0.52 | 0.89 | 0.73 | -0.12 | -0.06 | -0.24 | -0.20 | -0.18 | -0.15 | 0.04 | 0.04 |
| Unified space ($dim = 4$) | 0.18 | 0.10 | -0.17 | -0.13 | 0.16 | 0.12 | 0.79 | 0.63 | 0.93 | 0.82 | 0.10 | 0.06 | -0.23 | -0.18 | 0.67 | 0.48 | 0.30 | 0.24 |
| Unified space ($dim = 2$) | 0.41 | 0.35 | -0.09 | -0.08 | 0.12 | 0.10 | 0.77 | 0.57 | 0.94 | 0.82 | -0.22 | -0.16 | 0.50 | 0.39 | 0.83 | 0.62 | 0.41 | 0.33 |
| **JULE: Silhouette score** | | | | | | | | | | | | | | | | | | |
| Coupled space | 0.27 | 0.20 | 0.72 | 0.55 | 0.04 | 0.03 | 0.56 | 0.41 | 0.41 | 0.30 | 0.70 | 0.50 | 0.64 | 0.47 | 0.55 | 0.41 | 0.49 | 0.36 |
| Unified space ($dim = 128$) | 0.80 | 0.62 | 0.78 | 0.56 | 0.41 | 0.29 | 0.79 | 0.65 | 0.94 | 0.82 | 0.67 | 0.48 | 0.69 | 0.52 | 0.84 | 0.66 | 0.74 | 0.58 |
| Unified space ($dim = 32$) | 0.76 | 0.63 | 0.83 | 0.64 | 0.35 | 0.24 | 0.78 | 0.64 | 0.94 | 0.81 | 0.53 | 0.39 | 0.55 | 0.40 | 0.88 | 0.70 | 0.70 | 0.56 |
| Unified space ($dim = 16$) | 0.79 | 0.64 | 0.85 | 0.69 | 0.52 | 0.38 | 0.80 | 0.64 | 0.95 | 0.84 | 0.49 | 0.35 | 0.49 | 0.36 | 0.93 | 0.79 | 0.73 | 0.59 |
| Unified space ($dim = 8$) | 0.71 | 0.57 | 0.90 | 0.73 | 0.53 | 0.38 | 0.78 | 0.63 | 0.97 | 0.87 | 0.65 | 0.45 | 0.22 | 0.10 | 0.85 | 0.66 | 0.70 | 0.55 |
| Unified space ($dim = 4$) | 0.81 | 0.65 | 0.94 | 0.81 | 0.53 | 0.38 | 0.83 | 0.68 | 0.98 | 0.91 | 0.51 | 0.32 | 0.24 | 0.15 | 0.97 | 0.86 | 0.73 | 0.60 |
| Unified space ($dim = 2$) | 0.87 | 0.70 | 0.87 | 0.69 | 0.36 | 0.24 | 0.84 | 0.68 | 0.98 | 0.91 | 0.45 | 0.31 | 0.60 | 0.45 | 0.98 | 0.88 | 0.74 | 0.61 |
| **DEPICT: Calinski-Harabasz index** | | | | | | | | | | | | | | | | | | |
| Coupled space | 0.76 | 0.57 | 0.44 | 0.26 | 0.76 | 0.57 | 0.89 | 0.72 | 0.49 | 0.44 | | | | | | | 0.67 | 0.51 |
| Unified space ($dim = 128$) | 0.59 | 0.46 | 0.36 | 0.26 | 0.82 | 0.62 | 0.94 | 0.82 | 0.97 | 0.90 | | | | | | | 0.74 | 0.61 |
| Unified space ($dim = 32$) | 0.75 | 0.63 | 0.49 | 0.36 | 0.81 | 0.62 | 0.94 | 0.82 | 0.98 | 0.91 | | | | | | | 0.79 | 0.67 |
| Unified space ($dim = 16$) | 0.75 | 0.65 | 0.70 | 0.57 | 0.81 | 0.62 | 0.92 | 0.79 | 0.98 | 0.91 | | | | | | | 0.83 | 0.71 |
| Unified space ($dim = 8$) | 0.91 | 0.77 | 0.75 | 0.58 | 0.87 | 0.71 | 0.94 | 0.84 | 0.97 | 0.88 | | | | | | | 0.89 | 0.76 |
| Unified space ($dim = 4$) | 0.95 | 0.83 | 0.61 | 0.46 | 0.91 | 0.79 | 0.94 | 0.82 | 0.97 | 0.91 | | | | | | | 0.88 | 0.76 |
| Unified space ($dim = 2$) | 0.95 | 0.84 | 0.65 | 0.52 | 0.89 | 0.75 | 0.96 | 0.84 | 0.95 | 0.80 | | | | | | | 0.88 | 0.75 |
| **DEPICT: Davies-Bouldin index** | | | | | | | | | | | | | | | | | | |
| Coupled space | 0.81 | 0.59 | 0.45 | 0.31 | 0.90 | 0.74 | 0.89 | 0.72 | 0.63 | 0.59 | | | | | | | 0.73 | 0.59 |
| Unified space ($dim = 128$) | 0.42 | 0.28 | 0.53 | 0.37 | 0.71 | 0.57 | 0.74 | 0.56 | 0.96 | 0.86 | | | | | | | 0.67 | 0.53 |
| Unified space ($dim = 32$) | 0.55 | 0.42 | 0.49 | 0.35 | 0.70 | 0.54 | 0.73 | 0.53 | 0.96 | 0.88 | | | | | | | 0.69 | 0.54 |
| Unified space ($dim = 16$) | 0.58 | 0.48 | 0.67 | 0.52 | 0.74 | 0.58 | 0.67 | 0.50 | 0.97 | 0.90 | | | | | | | 0.73 | 0.59 |
| Unified space ($dim = 8$) | 0.71 | 0.61 | 0.65 | 0.49 | 0.75 | 0.59 | 0.82 | 0.65 | 0.99 | 0.92 | | | | | | | 0.78 | 0.65 |
| Unified space ($dim = 4$) | 0.73 | 0.65 | 0.53 | 0.42 | 0.79 | 0.62 | 0.90 | 0.78 | 0.96 | 0.84 | | | | | | | 0.78 | 0.66 |
| Unified space ($dim = 2$) | 0.92 | 0.78 | 0.60 | 0.42 | 0.81 | 0.66 | 0.92 | 0.80 | 0.99 | 0.92 | | | | | | | 0.85 | 0.72 |
| **DEPICT: Silhouette score** | | | | | | | | | | | | | | | | | | |
| Coupled space | 0.73 | 0.50 | 0.47 | 0.36 | 0.79 | 0.65 | 0.86 | 0.69 | 0.59 | 0.52 | | | | | | | 0.69 | 0.54 |
| Unified space ($dim = 128$) | 0.55 | 0.44 | 0.18 | 0.12 | 0.82 | 0.66 | 0.88 | 0.72 | 0.91 | 0.78 | | | | | | | 0.67 | 0.55 |
| Unified space ($dim = 32$) | 0.75 | 0.66 | 0.31 | 0.23 | 0.86 | 0.70 | 0.88 | 0.74 | 0.93 | 0.80 | | | | | | | 0.75 | 0.63 |
| Unified space ($dim = 16$) | 0.80 | 0.70 | 0.75 | 0.59 | 0.83 | 0.66 | 0.89 | 0.75 | 0.96 | 0.87 | | | | | | | 0.85 | 0.72 |
| Unified space ($dim = 8$) | 0.93 | 0.83 | 0.80 | 0.66 | 0.87 | 0.70 | 0.92 | 0.80 | 0.96 | 0.86 | | | | | | | 0.90 | 0.77 |
| Unified space ($dim = 4$) | 0.97 | 0.88 | 0.75 | 0.57 | 0.92 | 0.78 | 0.92 | 0.82 | 0.95 | 0.84 | | | | | | | 0.90 | 0.78 |
| Unified space ($dim = 2$) | 0.98 | 0.91 | 0.78 | 0.59 | 0.95 | 0.84 | 0.97 | 0.90 | 0.97 | 0.88 | | | | | | | 0.93 | 0.82 |

Table A8: The results of using various dimensions in the low-dimensional space in the hyperparameter tuning experiment are presented. $r_s$ and $\tau_B$ between the generated scores and NMI scores are reported. The results obtained using coupled space are presented as a baseline for comparison.

| | USPS | | YTF | | FRGC | | MNIST-test | | CMU-PIE | | UMist | | COIL-20 | | COIL-100 | | Average | |
|---|---|---|---|---|---|---|---|---|---|---|---|---|---|---|---|---|---|---|
| | $r_s$ | $\tau_B$ | $r_s$ | $\tau_B$ | $r_s$ | $\tau_B$ | $r_s$ | $\tau_B$ | $r_s$ | $\tau_B$ | $r_s$ | $\tau_B$ | $r_s$ | $\tau_B$ | $r_s$ | $\tau_B$ | $r_s$ | $\tau_B$ |
| JULE: Calinski-Harabasz index | | | | | | | | | | | | | | | | | | |
| Coupled space | 0.65 | 0.64 | 0.1 | 0.06 | -0.93 | -0.83 | 0.64 | 0.6 | -0.03 | -0.02 | -0.13 | -0.07 | 0.76 | 0.71 | 0.74 | 0.56 | 0.22 | 0.21 |
| Unified space ($dim = 128$) | 0.73 | 0.6 | 0.53 | 0.44 | -0.82 | -0.61 | 0.84 | 0.73 | 0.33 | 0.29 | -0.19 | -0.16 | 0.74 | 0.64 | 0.46 | 0.42 | 0.33 | 0.29 |
| Unified space ($dim = 32$) | 0.77 | 0.64 | 0.85 | 0.72 | -0.82 | -0.61 | 0.84 | 0.73 | 0.78 | 0.64 | -0.19 | -0.16 | 0.74 | 0.64 | 0.54 | 0.47 | 0.44 | 0.38 |
| Unified space ($dim = 16$) | 0.77 | 0.64 | 0.82 | 0.72 | -0.4 | -0.33 | 0.84 | 0.73 | 0.96 | 0.87 | -0.27 | -0.2 | 0.74 | 0.64 | 0.79 | 0.69 | 0.53 | 0.47 |
| Unified space ($dim = 8$) | 0.77 | 0.64 | 0.92 | 0.78 | 0.47 | 0.28 | 0.84 | 0.73 | 0.99 | 0.96 | -0.42 | -0.33 | 0.83 | 0.71 | 0.99 | 0.96 | 0.67 | 0.59 |
| Unified space ($dim = 4$) | 0.93 | 0.82 | 0.92 | 0.78 | 0.82 | 0.61 | 0.98 | 0.91 | 0.95 | 0.87 | 0.15 | 0.02 | 0.88 | 0.79 | 0.98 | 0.91 | 0.83 | 0.71 |
| Unified space ($dim = 2$) | 0.98 | 0.91 | 1.0 | 1.0 | 0.83 | 0.67 | 0.96 | 0.87 | 0.95 | 0.87 | 0.43 | 0.24 | 0.83 | 0.71 | 0.61 | 0.51 | 0.82 | 0.72 |
| JULE: Davies-Bouldin index | | | | | | | | | | | | | | | | | | |
| Coupled space | 0.54 | 0.38 | 0.15 | 0.17 | 0.85 | 0.67 | 0.43 | 0.29 | 0.78 | 0.56 | -0.08 | 0.02 | -0.26 | -0.14 | -0.9 | -0.78 | 0.19 | 0.15 |
| Unified space ($dim = 128$) | -0.14 | -0.16 | 0.88 | 0.78 | 0.78 | 0.67 | 0.18 | 0.11 | 0.79 | 0.6 | -0.66 | -0.47 | -0.69 | -0.57 | 0.33 | 0.11 | 0.18 | 0.13 |
| Unified space ($dim = 32$) | -0.09 | -0.07 | 0.88 | 0.78 | 0.93 | 0.83 | 0.41 | 0.29 | 0.79 | 0.6 | -0.52 | -0.29 | -0.67 | -0.5 | 0.07 | -0.07 | 0.22 | 0.20 |
| Unified space ($dim = 16$) | -0.1 | -0.07 | 0.98 | 0.94 | 0.88 | 0.78 | 0.49 | 0.33 | 0.98 | 0.91 | -0.37 | -0.16 | -0.86 | -0.79 | 0.25 | 0.07 | 0.28 | 0.25 |
| Unified space ($dim = 8$) | 0.36 | 0.33 | 0.6 | 0.44 | 0.88 | 0.78 | 0.89 | 0.78 | 0.98 | 0.91 | 0.02 | 0.02 | -0.55 | -0.43 | 0.41 | 0.24 | 0.45 | 0.38 |
| Unified space ($dim = 4$) | 0.94 | 0.82 | 0.6 | 0.5 | 0.72 | 0.56 | 0.81 | 0.6 | 0.95 | 0.87 | 0.18 | 0.16 | 0.02 | 0.0 | 0.66 | 0.51 | 0.61 | 0.50 |
| Unified space ($dim = 2$) | 0.47 | 0.33 | 0.55 | 0.39 | 0.18 | 0.17 | 0.54 | 0.47 | 0.92 | 0.82 | -0.28 | -0.2 | 0.43 | 0.43 | 0.9 | 0.78 | 0.46 | 0.40 |
| JULE: Silhouette score | | | | | | | | | | | | | | | | | | |
| Coupled space | 0.85 | 0.73 | 0.33 | 0.28 | 0.72 | 0.61 | 0.88 | 0.69 | 0.96 | 0.87 | 0.07 | 0.16 | 0.55 | 0.43 | 0.44 | 0.29 | 0.60 | 0.51 |
| Unified space ($dim = 128$) | 0.5 | 0.29 | 0.92 | 0.78 | 0.55 | 0.39 | 0.76 | 0.6 | 0.88 | 0.82 | -0.14 | -0.11 | 0.59 | 0.43 | 0.76 | 0.56 | 0.61 | 0.47 |
| Unified space ($dim = 32$) | 0.64 | 0.42 | 0.78 | 0.67 | 0.48 | 0.33 | 0.78 | 0.64 | 0.99 | 0.96 | -0.14 | -0.11 | 0.79 | 0.64 | 0.7 | 0.56 | 0.63 | 0.51 |
| Unified space ($dim = 16$) | 0.76 | 0.6 | 0.95 | 0.89 | 0.67 | 0.5 | 0.79 | 0.64 | 0.99 | 0.96 | -0.21 | -0.16 | 0.12 | 0.14 | 0.58 | 0.42 | 0.58 | 0.50 |
| Unified space ($dim = 8$) | 0.89 | 0.73 | 0.98 | 0.94 | 0.62 | 0.5 | 0.9 | 0.78 | 0.99 | 0.96 | 0.18 | 0.11 | 0.74 | 0.5 | 0.84 | 0.69 | 0.77 | 0.65 |
| Unified space ($dim = 4$) | 0.87 | 0.69 | 0.98 | 0.94 | 0.58 | 0.39 | 0.95 | 0.82 | 0.99 | 0.96 | 0.44 | 0.33 | 0.91 | 0.79 | 0.99 | 0.96 | 0.84 | 0.73 |
| Unified space ($dim = 2$) | 0.84 | 0.69 | 0.87 | 0.72 | 0.63 | 0.5 | 0.92 | 0.78 | 0.99 | 0.96 | 0.42 | 0.29 | 0.93 | 0.86 | 0.95 | 0.87 | 0.82 | 0.71 |
| DEPICT: Calinski-Harabasz index | | | | | | | | | | | | | | | | | | |
| Coupled space | 0.46 | 0.6 | -0.99 | -0.96 | -0.85 | -0.72 | 0.44 | 0.56 | -0.92 | -0.82 | | | | | | | -0.37 | -0.27 |
| Unified space ($dim = 128$) | 0.73 | 0.6 | -1.0 | -1.0 | -0.85 | -0.72 | 0.81 | 0.73 | -0.88 | -0.73 | | | | | | | -0.24 | -0.22 |
| Unified space ($dim = 32$) | 0.73 | 0.6 | -1.0 | -1.0 | -0.83 | -0.67 | 0.81 | 0.73 | 0.95 | 0.87 | | | | | | | 0.13 | 0.11 |
| Unified space ($dim = 16$) | 0.73 | 0.6 | -0.1 | -0.02 | -0.75 | -0.56 | 0.92 | 0.82 | 0.95 | 0.87 | | | | | | | 0.35 | 0.34 |
| Unified space ($dim = 8$) | 0.76 | 0.6 | 0.25 | 0.24 | 0.1 | 0.11 | 0.95 | 0.87 | 0.95 | 0.87 | | | | | | | 0.60 | 0.54 |
| Unified space ($dim = 4$) | 0.69 | 0.56 | 0.73 | 0.69 | 0.6 | 0.5 | 1.0 | 1.0 | 0.92 | 0.78 | | | | | | | 0.79 | 0.71 |
| Unified space ($dim = 2$) | 0.77 | 0.64 | 0.89 | 0.73 | 0.73 | 0.61 | 0.99 | 0.96 | 0.85 | 0.69 | | | | | | | 0.85 | 0.73 |
| DEPICT: Davies-Bouldin index | | | | | | | | | | | | | | | | | | |
| Coupled space | 0.46 | 0.6 | -0.78 | -0.64 | -0.85 | -0.72 | 0.44 | 0.56 | -0.1 | 0.02 | | | | | | | -0.17 | -0.04 |
| Unified space ($dim = 128$) | 0.19 | 0.16 | 0.96 | 0.91 | 0.62 | 0.5 | 0.82 | 0.69 | 0.84 | 0.73 | | | | | | | 0.69 | 0.60 |
| Unified space ($dim = 32$) | 0.22 | 0.2 | 0.99 | 0.96 | 0.83 | 0.67 | 0.88 | 0.73 | 0.99 | 0.96 | | | | | | | 0.78 | 0.70 |
| Unified space ($dim = 16$) | 0.28 | 0.24 | 0.99 | 0.96 | 0.35 | 0.28 | 0.88 | 0.73 | 0.93 | 0.87 | | | | | | | 0.69 | 0.62 |
| Unified space ($dim = 8$) | 0.46 | 0.42 | 0.96 | 0.87 | 0.43 | 0.33 | 0.9 | 0.78 | 0.92 | 0.78 | | | | | | | 0.73 | 0.64 |
| Unified space ($dim = 4$) | 0.6 | 0.56 | 1.0 | 1.0 | 0.3 | 0.22 | 0.96 | 0.87 | 0.82 | 0.6 | | | | | | | 0.74 | 0.65 |
| Unified space ($dim = 2$) | 0.84 | 0.64 | 0.73 | 0.6 | 0.27 | 0.22 | 0.83 | 0.69 | 0.64 | 0.42 | | | | | | | 0.66 | 0.51 |
| DEPICT: Silhouette score | | | | | | | | | | | | | | | | | | |
| Coupled space | 0.44 | 0.56 | -0.61 | -0.47 | -0.85 | -0.72 | 0.44 | 0.56 | -0.12 | -0.02 | | | | | | | -0.14 | -0.02 |
| Unified space ($dim = 128$) | 0.25 | 0.24 | -0.08 | 0.02 | 0.45 | 0.33 | 0.99 | 0.96 | 0.96 | 0.87 | | | | | | | 0.51 | 0.48 |
| Unified space ($dim = 32$) | 0.43 | 0.42 | 0.81 | 0.64 | 0.68 | 0.56 | 0.99 | 0.96 | 0.98 | 0.91 | | | | | | | 0.78 | 0.70 |
| Unified space ($dim = 16$) | 0.53 | 0.47 | 1.0 | 1.0 | 0.68 | 0.56 | 0.99 | 0.96 | 0.98 | 0.91 | | | | | | | 0.84 | 0.78 |
| Unified space ($dim = 8$) | 0.79 | 0.69 | 0.99 | 0.96 | 0.5 | 0.39 | 0.98 | 0.91 | 0.99 | 0.96 | | | | | | | 0.85 | 0.78 |
| Unified space ($dim = 4$) | 0.92 | 0.82 | 0.96 | 0.91 | 0.62 | 0.5 | 0.98 | 0.91 | 0.99 | 0.96 | | | | | | | 0.89 | 0.82 |
| Unified space ($dim = 2$) | 0.93 | 0.87 | 0.95 | 0.87 | 0.55 | 0.44 | 0.99 | 0.96 | 0.99 | 0.96 | | | | | | | 0.88 | 0.82 |

Table A9: The results of using various dimensions in the low-dimensional space in the cluster number determination experiment are presented. $r_s$ and $\tau_B$ between the generated scores and NMI scores are reported. The results obtained using coupled space are presented as a baseline for comparison.

| | USPS $r_s$ | USPS $\tau_B$ | YTF $r_s$ | YTF $\tau_B$ | FRGC $r_s$ | FRGC $\tau_B$ | MNIST-test $r_s$ | MNIST-test $\tau_B$ | CMU-PIE $r_s$ | CMU-PIE $\tau_B$ | UMist $r_s$ | UMist $\tau_B$ | COIL-20 $r_s$ | COIL-20 $\tau_B$ | COIL-100 $r_s$ | COIL-100 $\tau_B$ | Average $r_s$ | Average $\tau_B$ |
|---|---|---|---|---|---|---|---|---|---|---|---|---|---|---|---|---|---|---|
| **JULE: Calinski-Harabasz index** | | | | | | | | | | | | | | | | | | |
| Coupled space | 0.04 | 0.05 | 0.39 | 0.27 | -0.26 | -0.18 | 0.31 | 0.21 | -0.20 | -0.12 | 0.64 | 0.45 | 0.57 | 0.40 | 0.09 | 0.08 | 0.20 | 0.14 |
| Unified space ($dim = 128$) | 0.75 | 0.58 | 0.77 | 0.58 | 0.03 | 0.03 | 0.93 | 0.81 | 0.28 | 0.23 | 0.60 | 0.42 | 0.56 | 0.41 | 0.95 | 0.83 | 0.61 | 0.49 |
| Unified space ($dim = 32$) | 0.75 | 0.58 | 0.78 | 0.61 | 0.07 | 0.06 | 0.93 | 0.82 | 0.45 | 0.39 | 0.56 | 0.37 | 0.44 | 0.32 | 0.95 | 0.82 | 0.62 | 0.50 |
| Unified space ($dim = 16$) | 0.75 | 0.57 | 0.77 | 0.58 | 0.21 | 0.15 | 0.93 | 0.82 | 0.94 | 0.81 | 0.58 | 0.41 | 0.43 | 0.34 | 0.94 | 0.81 | 0.69 | 0.56 |
| Unified space ($dim = 8$) | 0.73 | 0.58 | 0.80 | 0.62 | 0.29 | 0.20 | 0.93 | 0.82 | 0.97 | 0.87 | 0.60 | 0.42 | 0.30 | 0.20 | 0.87 | 0.70 | 0.69 | 0.55 |
| Unified space ($dim = 4$) | 0.83 | 0.66 | 0.71 | 0.54 | 0.32 | 0.24 | 0.92 | 0.79 | 0.97 | 0.88 | 0.59 | 0.40 | 0.23 | 0.16 | 0.93 | 0.79 | 0.69 | 0.56 |
| Unified space ($dim = 2$) | 0.88 | 0.71 | 0.58 | 0.43 | 0.21 | 0.14 | 0.94 | 0.80 | 0.98 | 0.90 | 0.62 | 0.42 | 0.73 | 0.55 | 0.92 | 0.77 | 0.73 | 0.59 |
| **JULE: Davies-Bouldin index** | | | | | | | | | | | | | | | | | | |
| Coupled space | -0.27 | -0.15 | -0.14 | -0.09 | -0.23 | -0.14 | -0.35 | -0.19 | 0.20 | 0.16 | 0.53 | 0.36 | 0.63 | 0.44 | 0.33 | 0.26 | 0.09 | 0.08 |
| Unified space ($dim = 128$) | -0.27 | -0.15 | -0.33 | -0.24 | 0.40 | 0.26 | 0.45 | 0.33 | -0.16 | -0.20 | -0.45 | -0.34 | -0.15 | -0.07 | -0.56 | -0.40 | -0.14 | -0.10 |
| Unified space ($dim = 32$) | -0.28 | -0.20 | -0.54 | -0.40 | 0.47 | 0.32 | 0.39 | 0.28 | 0.17 | 0.11 | -0.46 | -0.32 | -0.32 | -0.24 | -0.59 | -0.43 | -0.15 | -0.11 |
| Unified space ($dim = 16$) | -0.29 | -0.21 | -0.37 | -0.26 | 0.46 | 0.31 | 0.50 | 0.39 | 0.76 | 0.63 | -0.25 | -0.19 | -0.23 | -0.17 | -0.41 | -0.32 | 0.02 | 0.02 |
| Unified space ($dim = 8$) | -0.38 | -0.23 | -0.52 | -0.39 | 0.47 | 0.34 | 0.75 | 0.58 | 0.87 | 0.71 | -0.16 | -0.10 | -0.29 | -0.23 | -0.22 | -0.16 | 0.06 | 0.06 |
| Unified space ($dim = 4$) | 0.04 | 0.08 | -0.24 | -0.19 | 0.52 | 0.35 | 0.87 | 0.71 | 0.90 | 0.78 | 0.08 | 0.03 | -0.29 | -0.21 | 0.62 | 0.43 | 0.31 | 0.25 |
| Unified space ($dim = 2$) | 0.28 | 0.27 | -0.21 | -0.14 | 0.53 | 0.37 | 0.89 | 0.71 | 0.94 | 0.82 | -0.28 | -0.21 | 0.48 | 0.37 | 0.75 | 0.56 | 0.42 | 0.34 |
| **JULE: Silhouette score** | | | | | | | | | | | | | | | | | | |
| Coupled space | 0.14 | 0.12 | 0.54 | 0.39 | -0.08 | -0.02 | 0.41 | 0.27 | 0.36 | 0.27 | 0.64 | 0.46 | 0.67 | 0.48 | 0.44 | 0.31 | 0.39 | 0.28 |
| Unified space ($dim = 128$) | 0.90 | 0.72 | 0.71 | 0.50 | 0.39 | 0.27 | 0.89 | 0.74 | 0.96 | 0.85 | 0.69 | 0.50 | 0.70 | 0.54 | 0.85 | 0.67 | 0.76 | 0.60 |
| Unified space ($dim = 32$) | 0.86 | 0.70 | 0.63 | 0.48 | 0.34 | 0.24 | 0.89 | 0.74 | 0.96 | 0.85 | 0.56 | 0.40 | 0.55 | 0.41 | 0.89 | 0.70 | 0.71 | 0.56 |
| Unified space ($dim = 16$) | 0.88 | 0.69 | 0.64 | 0.47 | 0.52 | 0.37 | 0.90 | 0.74 | 0.97 | 0.88 | 0.56 | 0.40 | 0.50 | 0.40 | 0.92 | 0.78 | 0.74 | 0.59 |
| Unified space ($dim = 8$) | 0.76 | 0.62 | 0.71 | 0.52 | 0.67 | 0.47 | 0.89 | 0.73 | 0.98 | 0.90 | 0.67 | 0.47 | 0.18 | 0.06 | 0.76 | 0.59 | 0.70 | 0.54 |
| Unified space ($dim = 4$) | 0.87 | 0.70 | 0.78 | 0.60 | 0.72 | 0.53 | 0.92 | 0.77 | 0.98 | 0.91 | 0.58 | 0.38 | 0.20 | 0.11 | 0.91 | 0.76 | 0.75 | 0.60 |
| Unified space ($dim = 2$) | 0.93 | 0.79 | 0.80 | 0.63 | 0.72 | 0.53 | 0.94 | 0.80 | 0.98 | 0.90 | 0.45 | 0.29 | 0.57 | 0.39 | 0.91 | 0.76 | 0.79 | 0.64 |
| **DEPICT: Calinski-Harabasz index** | | | | | | | | | | | | | | | | | | |
| Coupled space | 0.56 | 0.40 | 0.54 | 0.35 | 0.76 | 0.57 | 0.88 | 0.69 | 0.48 | 0.43 | | | | | | | 0.64 | 0.49 |
| Unified space ($dim = 128$) | 0.52 | 0.37 | 0.27 | 0.16 | 0.88 | 0.72 | 0.96 | 0.88 | 0.96 | 0.87 | | | | | | | 0.72 | 0.60 |
| Unified space ($dim = 32$) | 0.60 | 0.46 | 0.40 | 0.27 | 0.86 | 0.70 | 0.96 | 0.88 | 0.97 | 0.88 | | | | | | | 0.76 | 0.64 |
| Unified space ($dim = 16$) | 0.60 | 0.48 | 0.57 | 0.45 | 0.83 | 0.67 | 0.95 | 0.86 | 0.98 | 0.91 | | | | | | | 0.79 | 0.67 |
| Unified space ($dim = 8$) | 0.79 | 0.65 | 0.62 | 0.46 | 0.91 | 0.77 | 0.96 | 0.91 | 0.98 | 0.92 | | | | | | | 0.85 | 0.74 |
| Unified space ($dim = 4$) | 0.91 | 0.79 | 0.49 | 0.35 | 0.91 | 0.79 | 0.96 | 0.88 | 0.94 | 0.87 | | | | | | | 0.85 | 0.74 |
| Unified space ($dim = 2$) | 0.87 | 0.70 | 0.57 | 0.42 | 0.93 | 0.80 | 0.96 | 0.88 | 0.95 | 0.81 | | | | | | | 0.86 | 0.72 |
| **DEPICT: Davies-Bouldin index** | | | | | | | | | | | | | | | | | | |
| Coupled space | 0.61 | 0.42 | 0.48 | 0.32 | 0.92 | 0.74 | 0.88 | 0.69 | 0.62 | 0.56 | | | | | | | 0.70 | 0.55 |
| Unified space ($dim = 128$) | 0.39 | 0.24 | 0.39 | 0.23 | 0.68 | 0.54 | 0.84 | 0.65 | 0.97 | 0.89 | | | | | | | 0.65 | 0.51 |
| Unified space ($dim = 32$) | 0.43 | 0.28 | 0.38 | 0.23 | 0.66 | 0.52 | 0.81 | 0.62 | 0.97 | 0.89 | | | | | | | 0.65 | 0.51 |
| Unified space ($dim = 16$) | 0.45 | 0.33 | 0.50 | 0.35 | 0.72 | 0.58 | 0.77 | 0.59 | 0.97 | 0.91 | | | | | | | 0.68 | 0.55 |
| Unified space ($dim = 8$) | 0.62 | 0.52 | 0.50 | 0.29 | 0.72 | 0.59 | 0.93 | 0.77 | 0.99 | 0.96 | | | | | | | 0.75 | 0.63 |
| Unified space ($dim = 4$) | 0.74 | 0.69 | 0.44 | 0.33 | 0.78 | 0.65 | 0.93 | 0.82 | 0.95 | 0.83 | | | | | | | 0.77 | 0.66 |
| Unified space ($dim = 2$) | 0.85 | 0.71 | 0.50 | 0.33 | 0.72 | 0.56 | 0.92 | 0.82 | 0.98 | 0.91 | | | | | | | 0.79 | 0.66 |
| **DEPICT: Silhouette score** | | | | | | | | | | | | | | | | | | |
| Coupled space | 0.52 | 0.33 | 0.57 | 0.45 | 0.80 | 0.62 | 0.85 | 0.65 | 0.59 | 0.48 | | | | | | | 0.67 | 0.51 |
| Unified space ($dim = 128$) | 0.67 | 0.56 | 0.18 | 0.14 | 0.88 | 0.74 | 0.93 | 0.79 | 0.92 | 0.83 | | | | | | | 0.72 | 0.61 |
| Unified space ($dim = 32$) | 0.63 | 0.52 | 0.29 | 0.22 | 0.91 | 0.78 | 0.94 | 0.80 | 0.94 | 0.85 | | | | | | | 0.74 | 0.63 |
| Unified space ($dim = 16$) | 0.66 | 0.53 | 0.66 | 0.50 | 0.88 | 0.74 | 0.93 | 0.79 | 0.98 | 0.93 | | | | | | | 0.82 | 0.70 |
| Unified space ($dim = 8$) | 0.80 | 0.66 | 0.73 | 0.57 | 0.92 | 0.75 | 0.96 | 0.87 | 0.98 | 0.92 | | | | | | | 0.88 | 0.75 |
| Unified space ($dim = 4$) | 0.90 | 0.79 | 0.67 | 0.53 | 0.97 | 0.88 | 0.98 | 0.94 | 0.96 | 0.88 | | | | | | | 0.90 | 0.80 |
| Unified space ($dim = 2$) | 0.88 | 0.74 | 0.69 | 0.53 | 0.95 | 0.84 | 0.96 | 0.88 | 0.96 | 0.87 | | | | | | | 0.89 | 0.77 |

Table A10: The results of using various dimensions in the low-dimensional space in the hyperparameter tuning experiment are presented. $r_s$ and $\tau_B$ between the generated scores and ACC scores are reported. The results obtained using coupled space are presented as a baseline for comparison.

| | USPS | | YTF | | FRGC | | MNIST-test | | CMU-PIE | | UMist | | COIL-20 | | COIL-100 | | Average | |
|---|---|---|---|---|---|---|---|---|---|---|---|---|---|---|---|---|---|---|
| | $r_s$ | $\tau_B$ | $r_s$ | $\tau_B$ | $r_s$ | $\tau_B$ | $r_s$ | $\tau_B$ | $r_s$ | $\tau_B$ | $r_s$ | $\tau_B$ | $r_s$ | $\tau_B$ | $r_s$ | $\tau_B$ | $r_s$ | $\tau_B$ |
| JULE: Calinski-Harabasz index | | | | | | | | | | | | | | | | | | |
| Coupled space | 0.84 | 0.73 | 0.03 | -0.06 | -0.49 | -0.31 | 0.61 | 0.56 | -0.09 | -0.07 | -0.04 | 0.07 | 0.74 | 0.64 | 0.60 | 0.51 | 0.27 | 0.26 |
| Unified space ($dim = 128$) | 0.81 | 0.69 | 0.55 | 0.44 | -0.25 | -0.09 | 0.81 | 0.69 | 0.24 | 0.24 | -0.02 | 0.16 | 0.76 | 0.71 | 0.28 | 0.29 | 0.40 | 0.39 |
| Unified space ($dim = 32$) | 0.85 | 0.73 | 0.77 | 0.61 | -0.25 | -0.09 | 0.81 | 0.69 | 0.72 | 0.60 | -0.02 | 0.16 | 0.76 | 0.71 | 0.36 | 0.33 | 0.50 | 0.47 |
| Unified space ($dim = 16$) | 0.85 | 0.73 | 0.75 | 0.61 | 0.22 | 0.20 | 0.81 | 0.69 | 0.93 | 0.82 | -0.04 | 0.11 | 0.76 | 0.71 | 0.65 | 0.56 | 0.62 | 0.55 |
| Unified space ($dim = 8$) | 0.85 | 0.73 | 0.90 | 0.78 | 0.39 | 0.31 | 0.81 | 0.69 | 0.98 | 0.91 | -0.27 | -0.20 | 0.86 | 0.79 | 0.92 | 0.82 | 0.68 | 0.60 |
| Unified space ($dim = 4$) | 0.79 | 0.64 | 0.90 | 0.78 | 0.19 | 0.20 | 0.95 | 0.87 | 0.96 | 0.91 | -0.13 | -0.11 | 0.93 | 0.86 | 0.94 | 0.87 | 0.69 | 0.63 |
| Unified space ($dim = 2$) | 0.88 | 0.73 | 0.95 | 0.89 | 0.37 | 0.37 | 0.94 | 0.82 | 0.96 | 0.91 | 0.19 | 0.11 | 0.81 | 0.64 | 0.77 | 0.64 | 0.73 | 0.64 |
| JULE: Davies-Bouldin index | | | | | | | | | | | | | | | | | | |
| Coupled space | 0.39 | 0.29 | 0.10 | 0.06 | 0.37 | 0.25 | 0.49 | 0.33 | 0.83 | 0.60 | -0.28 | -0.29 | -0.29 | -0.21 | -0.87 | -0.73 | 0.09 | 0.04 |
| Unified space ($dim = 128$) | -0.33 | -0.24 | 0.87 | 0.78 | 0.38 | 0.25 | 0.22 | 0.16 | 0.84 | 0.64 | -0.78 | -0.69 | -0.67 | -0.50 | 0.48 | 0.24 | 0.13 | 0.08 |
| Unified space ($dim = 32$) | -0.26 | -0.16 | 0.87 | 0.78 | 0.59 | 0.42 | 0.46 | 0.33 | 0.84 | 0.64 | -0.69 | -0.51 | -0.69 | -0.57 | 0.27 | 0.07 | 0.17 | 0.13 |
| Unified space ($dim = 16$) | -0.27 | -0.16 | 0.93 | 0.83 | 0.68 | 0.48 | 0.54 | 0.38 | 0.99 | 0.96 | -0.55 | -0.47 | -0.83 | -0.71 | 0.43 | 0.20 | 0.24 | 0.19 |
| Unified space ($dim = 8$) | 0.26 | 0.24 | 0.53 | 0.33 | 0.68 | 0.48 | 0.94 | 0.82 | 0.99 | 0.96 | -0.13 | -0.20 | -0.52 | -0.36 | 0.55 | 0.38 | 0.41 | 0.33 |
| Unified space ($dim = 4$) | 0.81 | 0.64 | 0.70 | 0.61 | 0.21 | 0.09 | 0.84 | 0.64 | 0.96 | 0.91 | -0.28 | -0.16 | 0.12 | 0.07 | 0.74 | 0.56 | 0.51 | 0.42 |
| Unified space ($dim = 2$) | 0.53 | 0.42 | 0.43 | 0.28 | 0.49 | 0.37 | 0.53 | 0.42 | 0.93 | 0.87 | -0.58 | -0.33 | 0.41 | 0.36 | 0.85 | 0.64 | 0.45 | 0.38 |
| JULE: Silhouette score | | | | | | | | | | | | | | | | | | |
| Coupled space | 0.93 | 0.82 | 0.30 | 0.28 | 0.21 | 0.09 | 0.82 | 0.64 | 0.98 | 0.91 | -0.13 | -0.16 | 0.52 | 0.36 | 0.55 | 0.42 | 0.52 | 0.42 |
| Unified space ($dim = 128$) | 0.31 | 0.11 | 0.87 | 0.67 | 0.34 | 0.20 | 0.69 | 0.56 | 0.93 | 0.87 | -0.38 | -0.33 | 0.62 | 0.50 | 0.83 | 0.69 | 0.52 | 0.41 |
| Unified space ($dim = 32$) | 0.42 | 0.24 | 0.73 | 0.56 | 0.28 | 0.14 | 0.71 | 0.60 | 1.00 | 1.00 | -0.36 | -0.33 | 0.83 | 0.71 | 0.78 | 0.69 | 0.55 | 0.45 |
| Unified space ($dim = 16$) | 0.62 | 0.42 | 0.90 | 0.78 | 0.29 | 0.20 | 0.73 | 0.60 | 0.96 | 0.91 | -0.33 | -0.29 | 0.10 | 0.07 | 0.71 | 0.56 | 0.50 | 0.41 |
| Unified space ($dim = 8$) | 0.88 | 0.73 | 0.93 | 0.83 | 0.41 | 0.31 | 0.84 | 0.73 | 0.96 | 0.91 | 0.03 | -0.11 | 0.76 | 0.57 | 0.94 | 0.82 | 0.72 | 0.60 |
| Unified space ($dim = 4$) | 0.93 | 0.78 | 0.93 | 0.83 | 0.40 | 0.31 | 0.92 | 0.78 | 0.96 | 0.91 | 0.10 | 0.02 | 0.93 | 0.86 | 0.96 | 0.91 | 0.77 | 0.68 |
| Unified space ($dim = 2$) | 0.92 | 0.78 | 0.80 | 0.61 | 0.50 | 0.42 | 0.87 | 0.73 | 0.96 | 0.91 | 0.08 | 0.07 | 0.98 | 0.93 | 1.00 | 1.00 | 0.76 | 0.68 |
| DEPICT: Calinski-Harabasz index | | | | | | | | | | | | | | | | | | |
| Coupled space | 0.88 | 0.82 | -0.96 | -0.91 | -0.37 | -0.22 | 0.79 | 0.73 | -0.92 | -0.82 | | | | | | | -0.11 | -0.08 |
| Unified space ($dim = 128$) | 0.52 | 0.38 | -0.99 | -0.96 | -0.37 | -0.22 | 0.96 | 0.91 | -0.88 | -0.73 | | | | | | | -0.15 | -0.12 |
| Unified space ($dim = 32$) | 0.52 | 0.38 | -0.99 | -0.96 | -0.35 | -0.17 | 0.96 | 0.91 | 0.95 | 0.87 | | | | | | | 0.22 | 0.21 |
| Unified space ($dim = 16$) | 0.52 | 0.38 | -0.08 | 0.02 | -0.20 | -0.06 | 0.96 | 0.91 | 0.95 | 0.87 | | | | | | | 0.43 | 0.42 |
| Unified space ($dim = 8$) | 0.62 | 0.47 | 0.22 | 0.20 | 0.77 | 0.61 | 0.94 | 0.87 | 0.95 | 0.87 | | | | | | | 0.70 | 0.60 |
| Unified space ($dim = 4$) | 0.47 | 0.33 | 0.72 | 0.64 | 0.97 | 0.89 | 0.88 | 0.82 | 0.92 | 0.78 | | | | | | | 0.79 | 0.69 |
| Unified space ($dim = 2$) | 0.56 | 0.42 | 0.85 | 0.69 | 0.83 | 0.67 | 0.87 | 0.78 | 0.85 | 0.69 | | | | | | | 0.79 | 0.65 |
| DEPICT: Davies-Bouldin index | | | | | | | | | | | | | | | | | | |
| Coupled space | 0.88 | 0.82 | -0.77 | -0.60 | -0.37 | -0.22 | 0.79 | 0.73 | -0.10 | 0.02 | | | | | | | 0.09 | 0.15 |
| Unified space ($dim = 128$) | -0.13 | -0.07 | 0.99 | 0.96 | 0.57 | 0.44 | 0.64 | 0.51 | 0.84 | 0.73 | | | | | | | 0.58 | 0.52 |
| Unified space ($dim = 32$) | -0.08 | -0.02 | 1.00 | 1.00 | 0.73 | 0.61 | 0.71 | 0.56 | 0.99 | 0.96 | | | | | | | 0.67 | 0.62 |
| Unified space ($dim = 16$) | -0.01 | 0.02 | 1.00 | 1.00 | 0.80 | 0.67 | 0.71 | 0.56 | 0.93 | 0.87 | | | | | | | 0.69 | 0.62 |
| Unified space ($dim = 8$) | 0.20 | 0.20 | 0.94 | 0.82 | 0.82 | 0.72 | 0.81 | 0.69 | 0.92 | 0.78 | | | | | | | 0.74 | 0.64 |
| Unified space ($dim = 4$) | 0.37 | 0.33 | 0.99 | 0.96 | 0.77 | 0.61 | 0.83 | 0.69 | 0.82 | 0.60 | | | | | | | 0.75 | 0.64 |
| Unified space ($dim = 2$) | 0.81 | 0.60 | 0.71 | 0.56 | 0.82 | 0.72 | 0.70 | 0.51 | 0.64 | 0.42 | | | | | | | 0.73 | 0.56 |
| DEPICT: Silhouette score | | | | | | | | | | | | | | | | | | |
| Coupled space | 0.87 | 0.78 | -0.64 | -0.51 | -0.37 | -0.22 | 0.79 | 0.73 | -0.12 | -0.02 | | | | | | | 0.11 | 0.15 |
| Unified space ($dim = 128$) | -0.06 | 0.02 | -0.03 | 0.07 | 0.40 | 0.28 | 0.85 | 0.78 | 0.96 | 0.87 | | | | | | | 0.43 | 0.40 |
| Unified space ($dim = 32$) | 0.16 | 0.20 | 0.78 | 0.60 | 0.37 | 0.28 | 0.85 | 0.78 | 0.98 | 0.91 | | | | | | | 0.63 | 0.55 |
| Unified space ($dim = 16$) | 0.27 | 0.24 | 0.99 | 0.96 | 0.82 | 0.61 | 0.85 | 0.78 | 0.98 | 0.91 | | | | | | | 0.78 | 0.70 |
| Unified space ($dim = 8$) | 0.59 | 0.47 | 0.98 | 0.91 | 0.83 | 0.67 | 0.90 | 0.82 | 0.99 | 0.96 | | | | | | | 0.86 | 0.76 |
| Unified space ($dim = 4$) | 0.73 | 0.60 | 0.95 | 0.87 | 0.92 | 0.78 | 0.90 | 0.82 | 0.99 | 0.96 | | | | | | | 0.90 | 0.80 |
| Unified space ($dim = 2$) | 0.74 | 0.64 | 0.94 | 0.82 | 0.93 | 0.83 | 0.85 | 0.78 | 0.99 | 0.96 | | | | | | | 0.89 | 0.81 |

Table A11: The results of using various dimensions in the low-dimensional space in the cluster number determination experiment are presented. $r_s$ and $\tau_B$ between the generated scores and ACC scores are reported. The results obtained using coupled space are presented as a baseline for comparison.

