# OpenReview forum: "Towards Accurate Validation in Deep Clustering through Unified Embedding Learning"
_ICLR.cc/2025/Conference — Submitted to ICLR 2025_

### Official Review · Reviewer_8u7M · 2024-10-29

**Soundness:** 3
**Presentation:** 3
**Contribution:** 2
**Rating:** 5
**Confidence:** 4

**Summary:**

This paper addresses challenges in evaluating deep clustering methods, particularly discrepancies in comparing clustering results across different models due to varying learned embedding spaces. The authors propose a novel evaluation framework that introduces a unified embedding space for more accurate comparisons. This unified space aligns embeddings from multiple clustering results into a consistent representation, making internal validation measures more reliable and reducing inconsistencies. The proposed approach is empirically validated across several datasets, demonstrating improved accuracy in ranking clustering results.

**Strengths:**

- The authors provide thorough theoretical analyses of the limitations of traditional clustering evaluation approaches. They highlight the pitfalls of using internal validation measures in high-dimensional input spaces due to the curse of dimensionality and demonstrate the inconsistencies that arise when using these measures on coupled embedding spaces generated by different clustering models.
- The proposed method is evaluated extensively across several benchmark datasets, including MNIST, COIL, UMist, and others. The empirical results consistently show that the unified embedding framework outperforms traditional approaches (i.e., raw space, coupled space, and averaging across all spaces) in terms of rank correlation with external validation metrics.

**Weaknesses:**

- The proposed approach resembles multi-view learning methods, particularly in S1 where a fusion weight and unified similarity matrix are learned, and S2 where a low-dimensional multi-view fused embedding is developed. This raises the question: Could most multi-view learning methods achieve similar unified spaces? If so, what differentiates the proposed method from existing multi-view techniques?
- The quality of the unified embedding space may directly impact the framework’s ability to compare clustering models. If the unified space is not well-learned, how would this influence the reliability of the evaluations?
- The framework requires several optimization steps, such as learning the unified similarity matrix and the unified embedding space, which may be challenging for large datasets. S1, in particular, might not scale well for massive datasets. How does the proposed approach address these scalability concerns? The authors’ claim that datasets of more than 10,000 samples represent a sufficiently large scale is not convincing—evaluation on larger datasets (e.g., the complete MNIST dataset) is strongly recommended.
- Comparisons are limited to only two clustering methods. To fully demonstrate the robustness of the evaluation approach, at least three different clustering models should be included.

**Questions:**

- How does the proposed method differ from standard multi-view learning methods, particularly those that also learn a unified embedding space by combining multiple views? Would it be possible to benchmark against a few of these existing multi-view learning methods (e.g., Completer, cvpr'21) to clarify the distinctions?
- Are there any existing clustering evaluation frameworks that could be used as baselines for comparison to better highlight the strengths of the proposed approach?

---

> ### Author Response · Authors · 2024-11-24
>
> > Weakness1: The proposed approach resembles multi-view learning methods, particularly in S1 where a fusion weight and unified similarity matrix are learned, and S2 where a low-dimensional multi-view fused embedding is developed. This raises the question: Could most multi-view learning methods achieve similar unified spaces? If so, what differentiates the proposed method from existing multi-view techniques?
>
> R: While the optimization procedure for unifying similarity matrices is inspired by multi-view methods, the foundation of our approach differs fundamentally from these methods. Our algorithm is grounded in Theorem 3.7, which guarantees the consistency of our unified approach in estimating an informative space for reliable evaluation, given a reasonable collection of embedding spaces and a reliable weight. This is different from multi-view learning methods that aim to learn a better unified similarity matrix. In this regard, although applying other multi-view techniques will also achieve unified similarity matrices, we do not anticipate the same theoretical guarantee.
> More importantly, the originality of this work lies in identifying two critical pitfalls often overlooked in the current practices of model validation and results evaluation for deep clustering algorithms. Additionally, the novel evaluation strategy we proposed ensures more consistent and reliable assessments of these models. While the optimization algorithm employed in our evaluation strategy is inspired by existing work from other domains, the strategy itself is entirely new for deep clustering models. It addresses key challenges associated with existing evaluation routines (specifically the use of "raw space" and "coupled space"), as outlined in our paper. Many studies utilizing deep clustering techniques rely on internal validation methods to compare clustering approaches, optimize model configurations, and choose the optimal number of clusters. Proper implementation of internal validation is crucial for drawing reliable conclusions and ensuring accurate findings. As such, both the proposed evaluation strategy and the identified challenges represent significant and original contributions. It is important to note that no formally published work has previously addressed these challenges or provided solutions. Thus, we do not believe the novelty of our work lies in differentiating the proposed unification approach from other multi-view learning methods.

---

> ### Author Response · Authors · 2024-11-24
>
> > Weakness2: The quality of the unified embedding space may directly impact the framework’s ability to compare clustering models. If the unified space is not well-learned, how would this influence the reliability of the evaluations?
>
> R: If the unified space is not well-learned, the evaluation can become unreliable. However, based on our experiments, we find that this rarely occurs. We have discussed this in Section 5 (L502–507 in the initial submission, L506-510 in the revised submission), where we note that if most candidate embedding spaces fail to accurately preserve the similarity information and clustering structure in the data, the unified space is likely to inherit these deficiencies, resulting in a poorly learned unified space. In such cases, the clustering results derived from these spaces are typically untrustworthy. We argue that comparing subpar results to determine which is “less bad” does not constitute a meaningful evaluation strategy. To address this, we have included a discussion on this limitation and propose developing a testing procedure in future work to assess the viability of evaluations conducted on the obtained embedding spaces.

---

> ### Author Response · Authors · 2024-11-24
>
> > Weakness3: The framework requires several optimization steps, such as learning the unified similarity matrix and the unified embedding space, which may be challenging for large datasets. S1, in particular, might not scale well for massive datasets. How does the proposed approach address these scalability concerns? The authors’ claim that datasets of more than 10,000 samples represent a sufficiently large scale is not convincing—evaluation on larger datasets (e.g., the complete MNIST dataset) is strongly recommended.
>
> R: As discussed in Section 5, the proposed approach has a computational and memory complexity of $O(Mn^2)$, with the majority of the cost stemming from the calculation of pairwise distances between observations across all obtained embedding spaces. Since the complexity scales with $n^2$, it increases as $n$ grows. However, it is important to note that all the pairwise distances between observations from different spaces can be computed independently. This allows for easy parallelization, making the computation highly scalable even when $n$ is large. As a result, we do not anticipate any significant scalability issues for our method. Additionally, given the running time for deep clustering algorithms, the evaluation time is comparatively much shorter. The decision to use only the MNIST-test dataset was not related to the running time of our evaluation method. The primary reason for selecting MNIST-test is that our experiments involved two tasks requiring extensive tuning of configurations for each tested method. Specifically, we ran 42+10 configurations with JULE and 18+10 configurations with DEPICT, resulting in 52 runs of JULE and 28 runs of DEPICT per dataset. These experiments required substantial computational resources and took a considerable amount of time to complete. Running these experiments on the full MNIST dataset would have considerably extended the overall completion time—an increase that is not due to our evaluation approach, but rather the deep clustering methods themselves. Since MNIST-test produces clustering outcomes comparable to the full MNIST dataset, we opted for MNIST-test to reduce the computational time required for running these methods.

---

> ### Author Response · Authors · 2024-11-24
>
> > Weakness4: Comparisons are limited to only two clustering methods. To fully demonstrate the robustness of the evaluation approach, at least three different clustering models should be included.
>
> R: We follow the conventional categorization of deep clustering methods, which broadly classifies these methods into two branches: autoencoder-based and deep neural network-based (CDNN). From each branch, we selected one of the most widely cited and forked works—DEPICT and JULE—for our experiments. We chose DEPICT and JULE not only for their popularity but also for their strong reproducibility and comprehensive experiments conducted on well-known benchmark datasets. Since our focus is on model validation, we designed two tasks that require tuning configurations for each tested method. Specifically, we executed 42+10 configurations with JULE and 18+10 configurations with DEPICT, resulting in 52 runs of JULE and 28 runs of DEPICT per dataset. These experiments required significant computational time, taking days to weeks to complete. Including additional deep clustering models for testing would drastically increase the computational time in our experiment, even though this burden is not introduced by our evaluation approach. We believe including a large number of deep clustering methods will not change the conclusion in our paper, as the primary focus of our work is addressing the challenge of comparing clustering results generated from different embedding spaces. By running JULE and DEPICT, we generate a sufficient number of diverse embedding spaces, which are visualized in the supplementary files we submitted. These visualizations reveal substantial variations: some embedding spaces exhibit good clustering separation, while others do not; some show convex-shaped clusters, while others do not. This diversity assures us that the experimental findings drawn from these embedding spaces are adequate to support our conclusions.

---

> ### Author Response · Authors · 2024-11-24
>
> > Question 1: How does the proposed method differ from standard multi-view learning methods, particularly those that also learn a unified embedding space by combining multiple views? Would it be possible to benchmark against a few of these existing multi-view learning methods (e.g., Completer, cvpr'21) to clarify the distinctions?
>
> R: While the optimization procedure for unifying similarity matrices is inspired by multi-view methods, our approach is fundamentally different in its foundation. Our algorithm is built on Theorem 3.7, which ensures the consistency of our unified approach in estimating an informative space for reliable evaluation, given a reasonable collection of embedding spaces and a reliable weighting scheme. This differs from conventional multi-view learning methods, which primarily focus on constructing a unified similarity matrix optimized for the specific objectives of a downstream task (e.g., clustering). In this context, while applying other multi-view techniques can also produce unified similarity matrices, they lack the same theoretical consistency guarantees that underpin our approach. Although existing multi-view clustering methods may lack the same theoretical guarantees as our approach, their algorithms could potentially be adapted to enhance our framework. Such adaptations would represent a more detailed and granular extension of our work. As the first effort in this area, our primary objective is to identify key challenges in evaluating deep clustering methods and propose an effective initial solution. We view these adaptations and refinements as promising directions for future work. A discussion of this point, along with references to the suggested work, has been included in the revised version.

---

> ### Author Response · Authors · 2024-11-24
>
> > Question 2: Are there any existing clustering evaluation frameworks that could be used as baselines for comparison to better highlight the strengths of the proposed approach?
>
> R: As discussed in Section 1, evaluations based on “raw space” and “coupled space” represent the two existing approaches commonly used for comparing clustering results in deep clustering tasks. We have cited numerous works that follow these methods. However, prior to our work, there has been no formally published research that directly discusses or addresses the challenges associated with these evaluation methods. To better highlight the strengths of our proposed approach, we introduced a baseline method called “all space”. This baseline represents an intuitive way to address the identified pitfalls in existing methods. We compared the performance of our proposed unified space approach against this baseline to demonstrate its effectiveness.

---

> > ### Comment · Reviewer_8u7M · 2024-11-26
> >
> > Thank you for the additional clarifications. However, I will maintain my initial score as the authors’ explanation of the differences in multi-view learning remains unclear. Moreover, as Reviewer QSGy highlighted, the lack of comparisons with key representative deep clustering algorithms, also noted in Weakness 4, has not been adequately addressed.

---

### Official Review · Reviewer_QSGy · 2024-11-01

**Soundness:** 2
**Presentation:** 3
**Contribution:** 3
**Rating:** 5
**Confidence:** 5

**Summary:**

The authors propose a novel internal cluster quality measure for deep clustering algorithms. The key idea is to learn a unified embedding space that algins different embedding spaces learned by deep clustering models into a common space. The unified embedding is then used to compare the different clusterings with commonly used internal cluster evaluation methods, like the silhouette score.

**Strengths:**

**Originality**
- The idea of combining multiple embeddings learned from deep clustering methods to achieve a unified embedding to compare different clustering solutions is interesting.

**Quality**
- Evaluation across a wide range of diverse data sets and three different internal cluster evaluation methods provides good evidence for their proposed evaluation procedure.

**Clarity**
- The method description and Figure 1 illustrate the method clearly

**Significance**
- Internal cluster quality measures are of high significance for the deep clustering community. I would even say that it is one of the most pressing issues that holds back the application of deep clustering algorithms in practice. Currently, almost all deep clustering methods need to be tuned with access to ground truth labels, which is fine for method development, but is not a realistic use case for clustering in practice. Therefore, the presented  work is of high significance to the deep clustering community.

**Weaknesses:**

**Originality**
- Existing work (Figure 4 in Lowe et al, 2024) provides already a large-scale analysis of internal cluster measures (silhouette score) for clustering methods in embedding spaces. Their work shows that there is a strong correlation between the AMI (Adjusted Mutual Information) and the silhouette score computed in the UMAP reduced embedding space. This work should be discussed in the related work section so that it is clear, why the proposed method is necessary and a simple UMAP reduction for each embedding would not work.

**Quality**
- The selection of DEPICT and JULE for evaluation experiments is not well motivated. There are many more “foundational” deep clustering methods that are widely used and have inspired many follow-ups, e.g., DEC (Xie et al, 2016), IDEC (Guo et al, 2017), DCN (Yang et al, 2017). Further, only autoencoder-based methods are compared and no recent contrastive methods, like Contrastive Clustering (Li et al, 2021), SCAN (Van Gansbeke et al, 2020) or SeCu (Qi 2023). I understand that it is not feasible to compare with every deep clustering method there is, but the selection of methods in your experiment section should be clearly motivated. For example, take one or two methods from each deep clustering family, like k-means based, hierarchical clustering based, density based… and with different representation learning objectives, like autoencoder and contrastive learning.

**Significance**
- My concern with the proposed method is that it might not be very useful in practice, as it requires multiple embedded spaces that need to be learned first with deep clustering methods. This makes it quite expensive to compare clustering solutions.


**References**

Xie, J., Girshick, R. and Farhadi, A., 2016, June. Unsupervised deep embedding for clustering analysis. In International conference on machine learning (pp. 478-487). PMLR.

Yang, B., Fu, X., Sidiropoulos, N.D. and Hong, M., 2017, July. Towards k-means-friendly spaces: Simultaneous deep learning and clustering. In international conference on machine learning (pp. 3861-3870). PMLR.

Guo, X., Gao, L., Liu, X. and Yin, J., 2017, August. Improved deep embedded clustering with local structure preservation. In Ijcai (Vol. 17, pp. 1753-1759).

Li, Y., Hu, P., Liu, Z., Peng, D., Zhou, J.T. and Peng, X., 2021, May. Contrastive clustering. In Proceedings of the AAAI conference on artificial intelligence (Vol. 35, No. 10, pp. 8547-8555).

Van Gansbeke, W., Vandenhende, S., Georgoulis, S., Proesmans, M. and Van Gool, L., 2020, August. Scan: Learning to classify images without labels. In European conference on computer vision (pp. 268-285). Cham: Springer International Publishing.

Qian, Q., 2023. Stable cluster discrimination for deep clustering. In Proceedings of the IEEE/CVF International Conference on Computer Vision (pp. 16645-16654).

Lowe, S. C., Haurum, J. B., Oore, S., Moeslund, T. B., & Taylor, G. W. (2024). An Empirical Study into Clustering of Unseen Datasets with Self-Supervised Encoders. arXiv preprint arXiv:2406.02465.

**Questions:**

- Are the internal cluster measures in the comparison representations (“all spaces”, “coupled spaces” and “raw space”) computed in a t-SNE reduced space or in the higher dimensional representation space?

- How many embedding spaces are needed to learn a sufficiently representative “unified embedding”?

- Please justify the selection of JULE and DEPICT for your main experiments. If possible, add further deep clustering methods to your evaluation. See discussed weakness.

- Please explain how your approach relates to the results in Lowe et al. (2024). I would like to see a clear motivation of why your method is needed and a simpler baseline like UMAP reduced embeddings does not work. See also the corresponding discussed weakness.

---

> ### Author Response · Authors · 2024-11-24
>
> > Originality: Existing work (Figure 4 in Lowe et al, 2024) provides already a large-scale analysis of internal cluster measures (silhouette score) for clustering methods in embedding spaces. Their work shows that there is a strong correlation between the AMI (Adjusted Mutual Information) and the silhouette score computed in the UMAP reduced embedding space. This work should be discussed in the related work section so that it is clear, why the proposed method is necessary and a simple UMAP reduction for each embedding would not work.
>
> R: We sincerely thank the reviewer for highlighting this related work. While our paper focuses on comparing the performance of different deep clustering methods on the same dataset, the suggested work takes a broader perspective by evaluating the utility of self-supervised learned representations based on the consistency of clustering results with external measures such as AMI. In the suggested work's Figure 4, the correlations between AMI and Silhouette scores are computed across different datasets and encoders, providing an exploratory analysis. In contrast, our work aims to establish a more rigorous foundation and framework for evaluating the clustering results of multiple deep clustering methods on the same dataset. This distinction in goals is crucial, as comparing evaluation metrics across results from different datasets generally lacks statistical rigor, though it may be acceptable in the context of an exploratory analysis.
>
> We also note that a simple UMAP reduction for each embedding may not suffice because methods would still be evaluated using a "paired score" approach, which still violates the assumptions of internal validation measures. Following the reviewer’s suggestion, we implemented the UMAP reduction method with the same parameter settings as in the suggested work and compared the average Spearman rank correlation of different evaluation regimes with NMI scores. Our results corroborate the findings of the suggested work, showing that UMAP-reduced space significantly increases correlation. However, the correlation remains below that achieved by our proposed unified embedding approach. Additionally, the UMAP-reduced space method requires repeated optimization to estimate the low-dimensional space for every coupled embedding space, whereas our proposed approach requires only a single optimization to obtain the unified space, making it significantly more efficient. For these reasons, we believe the proposed method is still necessary. We will include the results of this comparison and incorporate the insights from the suggested work, as recommended by the reviewer, in our camera-ready version, should it be accepted.
>
> ### Hyperparameter Tuning
>
> |                       | Unified Space | Coupled Space | UMAP Reduced Coupled Space |
> |-------------------------------|---------------|---------------|----------------------------|
> | **JULE: Calinski-Harabasz index** | 0.75          | 0.29          | 0.66                       |
> | **JULE: Davies-Bouldin index**  | 0.41          | 0.16          | 0.54                       |
> | **JULE: Silhouette score**      | 0.74          | 0.49          | 0.72                       |
> | **DEPICT: Calinski-Harabasz index** | 0.88          | 0.67          | 0.69                       |
> | **DEPICT: Davies-Bouldin index** | 0.85          | 0.73          | 0.74                       |
> | **DEPICT: Silhouette score**    | 0.93          | 0.69          | 0.76                       |
>
> ### Cluster Number Determination
>
> |                         | Unified Space | Coupled Space | UMAP Reduced Coupled Space |
> |-------------------------------|---------------|---------------|----------------------------|
> | **JULE: Calinski-Harabasz index** | 0.82          | 0.22          | 0.73                       |
> | **JULE: Davies-Bouldin index**  | 0.46          | 0.19          | 0.40                       |
> | **JULE: Silhouette score**      | 0.82          | 0.60          | 0.71                       |
> | **DEPICT: Calinski-Harabasz index** | 0.85          | -0.37         | 0.44                       |
> | **DEPICT: Davies-Bouldin index** | 0.66          | -0.17         | 0.52                       |
> | **DEPICT: Silhouette score**    | 0.88          | -0.14         | 0.79                       |

---

> ### Author Response · Authors · 2024-11-24
>
> > Quality: The selection of DEPICT and JULE for evaluation experiments is not well motivated. There are many more “foundational” deep clustering methods that are widely used and have inspired many follow-ups, e.g., DEC (Xie et al, 2016), IDEC (Guo et al, 2017), DCN (Yang et al, 2017). Further, only autoencoder-based methods are compared and no recent contrastive methods, like Contrastive Clustering (Li et al, 2021), SCAN (Van Gansbeke et al, 2020) or SeCu (Qi 2023). I understand that it is not feasible to compare with every deep clustering method there is, but the selection of methods in your experiment section should be clearly motivated. For example, take one or two methods from each deep clustering family, like k-means based, hierarchical clustering based, density based… and with different representation learning objectives, like autoencoder and contrastive learning.
>
> R: We would like to first thank the reviewer for bringing up these deep clustering works. We follow the conventional categorization of deep clustering methods, which broadly classifies these methods into two branches: autoencoder-based and deep neural network-based (CDNN). From each branch, we selected one of the most widely cited and forked works—DEPICT and JULE—for our experiments. The majority of the works mentioned by the reviewer can still be classified within these two branches. We chose DEPICT and JULE not only for their popularity but also for their strong reproducibility and comprehensive experiments conducted on well-known benchmark datasets.
> Since our focus is on model validation, we designed two tasks that require tuning configurations for each tested method. Specifically, we executed 42+10 configurations with JULE and 18+10 configurations with DEPICT, resulting in 52 runs of JULE and 28 runs of DEPICT per dataset. These experiments required significant computational time, taking days to weeks to complete. Including additional deep clustering models for testing would drastically increase the computational time in our experiment, even though this burden is not introduced by our evaluation approach.
> We believe including a large number of deep clustering methods will not change the conclusion in our paper, as the primary focus of our work is addressing the challenge of comparing clustering results generated from different embedding spaces. By running JULE and DEPICT, we generate a sufficient number of diverse embedding spaces, which are visualized in the supplementary files we submitted. These visualizations reveal substantial variations: some embedding spaces exhibit good clustering separation, while others do not; some show convex-shaped clusters, while others do not. This diversity assures us that the experimental findings drawn from these embedding spaces are adequate to support our conclusions.

---

> ### Author Response · Authors · 2024-11-24
>
> > Significance: My concern with the proposed method is that it might not be very useful in practice, as it requires multiple embedded spaces that need to be learned first with deep clustering methods. This makes it quite expensive to compare clustering solutions.
>
> R: Our method does not require learning multiple embedding spaces. Instead, the multiple embedding spaces are generated from different runs of deep clustering algorithms. Specifically, when you run various deep clustering algorithms or even multiple runs of the same algorithm, you obtain different embedding spaces. These embedding spaces are then used as inputs to our algorithm. Our approach focuses solely on generating a unified embedding space from these inputs. As a result, all computational effort is dedicated to the unification process, which has a computational and memory complexity of $O(Mn^2)$ (Section 5). This complexity is comparable to that of calculating internal measure scores, and thus does not introduce significant additional computational cost. Therefore, the computational concerns raised by the reviewer do not apply to our evaluation framework.

---

> ### Author Response · Authors · 2024-11-24
>
> > Questions:
> > Are the internal cluster measures in the comparison representations (“all spaces”, “coupled spaces” and “raw space”) computed in a t-SNE reduced space or in the higher dimensional representation space?
>
> R: None of these scores are calculated in the t-SNE-reduced space. In our paper, only the "Unified space" is computed within the unified embedding space, which is estimated using a t-SNE-based method. As shown in Figure 1, "All spaces" and "Coupled spaces" are evaluated in the embedding spaces generated by the deep clustering algorithms, while "Raw space" is calculated in the original high-dimensional input space.
>
> > How many embedding spaces are needed to learn a sufficiently representative “unified embedding”?
>
> R: There is no strict requirement for the number of candidate embedding spaces needed to learn a sufficiently representative unified embedding space. As demonstrated in Theorem 3.7, under the weak consistency assumption for the weights, the unified embedding space can align with an informative space, ensuring reliable evaluation.
>
> > Please justify the selection of JULE and DEPICT for your main experiments. If possible, add further deep clustering methods to your evaluation. See discussed weakness.
>
> R: Please see our response to the corresponding weakness.
>
> > Please explain how your approach relates to the results in Lowe et al. (2024). I would like to see a clear motivation of why your method is needed and a simpler baseline like UMAP reduced embeddings does not work. See also the corresponding discussed weakness.
>
> R: Please see our response to the corresponding weakness.

---

> > ### Comment · Reviewer_QSGy · 2024-11-25
> >
> > Thank you for the additional experiments and explanations. I appreciate the additional comparison to UMAP. However, I will keep my initial score and decided against giving a higher score, due to two reasons.
> >
> > First, I highly disagree that JULE and DEPICT are representative deep clustering algorithms, e.g., DEC is more cited and established. Further, the field has advanced significantly since the introduction of JULE and DEPICT. Their exist now many more proposed methods that are built using different representation learning methods, like contrastive learning, or entirely different clustering methodologies, like hierarchical clustering, see DeepECT [1] or Tree VAEs [2]. The authors should expand their analysis in a principled manner and compare more recent/representative algorithms.
> >
> > Second, my concern regarding the significance of the proposed work was not properly adressed. I was referring to the requirement of conducting multiple runs of the same deep clustering algorithm that seems to be necessary to achieve good results with your method. These runs are costly and it would be desirable that your method works with as few embeddings generated through such runs as possible. I could not find an ablation in the paper on how many candidate embedding spaces are required and your response to my corresponding question only answered this question partially without providing additional results.
> >
> > While promising, I think this paper should not be accepted in its current form.
> >
> > -----
> >
> > [1] Mautz, D., Plant, C., & Böhm, C. (2019, November). Deep embedded cluster tree. In 2019 IEEE International Conference on Data Mining (ICDM) (pp. 1258-1263). IEEE.
> >
> > [2] Manduchi, L., Vandenhirtz, M., Ryser, A., & Vogt, J. (2023). Tree variational autoencoders. Advances in Neural Information Processing Systems, 36, 54952-54986.

---

### Official Review · Reviewer_FTn3 · 2024-11-03

**Soundness:** 3
**Presentation:** 3
**Contribution:** 3
**Rating:** 5
**Confidence:** 4

**Summary:**

This paper proposes a new deep clustering evaluation framework, which aims to solve the problem that different deep clustering algorithms are difficult to compare and evaluate in high-dimensional space. Experimental results show that this method outperforms traditional methods in terms of accuracy and consistency of internal evaluation, which helps to more reliably evaluate clustering performance.

**Strengths:**

1. By unifying the embedding spaces of different models into a common space, the evaluation bias caused by different algorithms or parameters can be reduced, making the evaluation results more consistent.

2. Through experimental verification, the method in the paper shows higher reliability when using internal evaluation indicators (such as Silhouette score, Calinski-Harabasz index, etc.), and is highly correlated with external evaluation indicators (such as clustering accuracy).

3. Compared with traditional embedding methods that require frequent parameter adjustment, the main steps of the unified embedding space method do not rely on specific parameters, are simple to operate and easy to promote.

**Weaknesses:**

1) The font of the text in the figure should be consistent with the font of the text;

**Questions:**

1) this work relies on Euclidean distance as a similarity metric. In some deep clustering tasks, other distance metrics (such as cosine similarity) may perform better. Can your evaluation framework maintain consistent results under different similarity metrics?

2) the aothurs focus on preserving the local structure of the data to improve clustering accuracy. However, on some datasets, preserving the global structure may be equally important. In the process of generating the unified embedding space, have you considered balancing the impact of local and global structures? Does this method have limitations on datasets with particularly complex data distribution?

---

> ### Author Response · Authors · 2024-11-24
>
> > Weaknesses: The font of the text in the figure should be consistent with the font of the text.
>
> R: Thank you for bringing this to our attention. We have addressed this in the revised version.

---

> > ### Author Response · Authors · 2024-11-24
> >
> > > Question 2: the aothurs focus on preserving the local structure of the data to improve clustering accuracy. However, on some datasets, preserving the global structure may be equally important. In the process of generating the unified embedding space, have you considered balancing the impact of local and global structures? Does this method have limitations on datasets with particularly complex data distribution?
> >
> > R: Firstly, we prioritize preserving local structure, as this enhances the consistency between internal evaluations and external measures. As discussed in Section 5, “we focus on local structure because it is generally more crucial for clustering accuracy. Clustering fundamentally involves grouping similar objects together, making the preservation of local data structure more relevant for differentiating between clusters (Rosales et al., 2004; Yang et al., 2016; Guo et al., 2017)” (L511–515 in the initial submission, L515-520 in the revised version). Additionally, we pointed out that “to achieve this, we benchmark our method against external measures to ensure better alignment with actual performance. Internal measures are typically designed to evaluate clustering quality, which may not fully reflect the correctness of clustering results. This discrepancy underscores our approach: preserving local structure in internal evaluations enhances alignment with external evaluations, given that clustering accuracy is less concerned with global geometry aspects like cluster size and distance” (L515–520 in the initial submission, L520-525 in the revised submission).
> > Furthermore, as discussed in Section 4.3 (L481–486 in the initial submission, L486-491 in the revised submission), increasing the number of dimensions can improve the recovery of global structure. However, this comes at the cost of sacrificing the preservation of local structure. Nevertheless, as highlighted in Section 4.3, the findings in Tables A8 through A11 show that lower dimensionalities still yield similarly strong performance. This suggests that this trade-off may even compromise the consistency and reliability of result evaluation.
> > In our experiments, we have conducted evaluations on deep clustering tasks involving multiple high-dimensional image datasets. To our knowledge, the consistency of clustering result evaluation for the proposed methods is not significantly affected by the data distribution. Therefore, we believe there should be no limitations when applying our methods to datasets with complex data distributions.

---

> ### Author Response · Authors · 2024-11-24
>
> > Question 1: this work relies on Euclidean distance as a similarity metric. In some deep clustering tasks, other distance metrics (such as cosine similarity) may perform better. Can your evaluation framework maintain consistent results under different similarity metrics?
>
> R: If the reviewer is referring to the distance metrics used in internal measures, our evaluation framework may not ensure consistent results. For instance, employing the Silhouette score with cosine similarity tends to yield less consistency with external measures compared to using Euclidean distance, as discussed in Section 5 (L522–525 in the initial submission, L526-529 in the revised version ). However, this issue is specific to the choice of distance function within the internal measures for evaluation and is unrelated to the selection of distance metrics used for performing the deep clustering tasks themselves.
> Typically, most internal measures default to Euclidean distance. For instance, the definitions of the Calinski-Harabasz index and the Davies-Bouldin index explicitly rely on Euclidean distance for calculating pairwise distances. While the Silhouette score allows flexibility in the choice of distance function, Euclidean distance remains the standard option.
> More importantly, we observed that when evaluating results within a single embedding space, both cosine similarity and Euclidean distance produce similarly consistent evaluation results. This suggests that in naturally formed embedding spaces from deep neural networks, the choice of distance function for internal measures has only a minor impact on evaluation outcomes. Our method favors Euclidean distance because of the way it produces the unified embedding. Therefore, we recommend that users of our method select internal measures that utilize Euclidean distance. Incorporating internal measures with alternative distance functions is an avenue for future work, as noted in Section 5.

---

### Official Review · Reviewer_RgaB · 2024-11-04

**Soundness:** 2
**Presentation:** 2
**Contribution:** 2
**Rating:** 5
**Confidence:** 4

**Summary:**

This paper proposed a model that searches the common representations from multiple learned representations of different methods via clustering. Additionally, this architecture can serve as an evaluation metric for comparing various clustering methods. The paper is well-organized and easy to follow. However, I have some concerns. The techniques used in this paper, including all modules and evaluation metrics, do not appear novel.

**Strengths:**

1. The whole paper is easy to follow and well-organized.

2. The motivation is clear.

**Weaknesses:**

1. The model itself lacks originality; the unified similarity matrix learning module appears to be derived from [1], and the unified embedding space learning module closely resembles IDEC [2].

2. Equation (4) means that $U$ should more closely approximate $S^{(m)}$ as their Euclidean distance decreases. But all $S^{(m)}$ is learned during the optimization process, relying on the unreliable metric to decide their optimization trends, does this point make sense? It could cause performance to depend heavily on how to initialize the weight $w$.

3. Lacking clear evaluation details. The paper does not specify which variables were used to calculate the NMI and ACC scores.

4. Why do results from all spaces sometimes outperform those from the unified space, while in other cases, the unified space outperforms all spaces? Please analyze this point clearly.

5. The t-SNE visualization comparing the unified embedding with the coupled embeddings should be included.

**References:**

[1] Feiping Nie, Jing Li, Xuelong Li, et al. Self-weighted multiview clustering with multiple graphs.

[2] Guo X, Gao L, Liu X, et al. Improved deep embedded clustering with local structure preservation. IJCAI. 2017, 17: 1753-1759.

**Questions:**

Please see Weaknesses.

---

> ### Author Response · Authors · 2024-11-24
>
> > Weakness 1: The model itself lacks originality; the unified similarity matrix learning module appears to be derived from [1], and the unified embedding space learning module closely resembles IDEC [2].
>
> R: While the optimization algorithm employed in our evaluation strategy is inspired by existing work from other domains, the strategy itself is entirely new for deep clustering models. The originality of this work lies in identifying two critical pitfalls often overlooked in the current practices of model validation and results evaluation for deep clustering algorithms. Additionally, the novel evaluation strategy we proposed ensures more consistent and reliable assessments of these models. It addresses key challenges associated with existing evaluation routines (specifically the use of "raw space" and "coupled space"), as outlined in our paper. Many studies utilizing deep clustering techniques rely on internal validation methods to compare clustering approaches, optimize model configurations, and choose the optimal number of clusters. Proper implementation of internal validation is crucial for drawing reliable conclusions and ensuring accurate findings. As such, both the proposed evaluation strategy and the identified challenges represent significant and original contributions. It is important to note that no formally published work has previously addressed these challenges or provided solutions. Thus, we disagree with the reviewer’s comment regarding a lack of originality. Furthermore, while our unified similarity matrix learning module shares some similarities with [1], it is distinct in both the formulation of the optimization problem and the approach to its solution. More importantly, while the optimization procedure for unifying similarity matrices is inspired by multi-view methods, the foundation of our approach differs fundamentally from these methods. Our algorithm is grounded in Theorem 3.7, which guarantees the consistency of our unified approach in estimating an informative space for reliable evaluation, given a reasonable collection of embedding spaces and a reliable weight. This is different from multi-view learning methods that aim to learn a better unified similarity matrix. In this regard, although applying other multi-view techniques will also achieve unified similarity matrices, we do not anticipate the same theoretical guarantee.

---

> ### Author Response · Authors · 2024-11-24
>
> > Weakness 2: Equation (4) means that $U$ should more closely approximate $S^{(m)}$ as their Euclidean distance decreases. But all $S^{(m)}$ is learned during the optimization process, relying on the unreliable metric to decide their optimization trends, does this point make sense? It could cause performance to depend heavily on how to initialize the weight $w$.
>
> R: Firstly, we would like to clarify that $ S^{(m)} $ is not learned during the optimization process. Instead, $ S^{(m)} $ represents the pairwise similarity matrix for all observations, corresponding to the embedding data $ Z^{(m)} $, which is generated by model $ m $. Thus, $ S^{(m)} $ is provided as input at the start of the optimization. The only quantities learned during the optimization process are $ U $ and $ w^{(m)} $.
> Secondly, we acknowledge that the estimated unified embedding data may depend on the initialization of $ w^{(m)} $, as the algorithm seeks a local optimum. However, consistent with standard practices in ensemble learning, we initialize the contributions $ w^{(m)} $ uniformly across all embedding spaces. The use of Equation (4) is inspired by prior work [1] and ensures the convergence of the algorithm.

---

> ### Author Response · Authors · 2024-11-24
>
> > Weakness 3: Lacking clear evaluation details. The paper does not specify which variables were used to calculate the NMI and ACC scores.
>
> R: NMI and ACC scores are calculated by comparing the estimated cluster labels obtained from the clustering results of each model to the true cluster labels. This calculation is straightforward and aligns directly with the natural definitions of these two metrics. Additionally, we have provided detailed explanations of this process in Section D of our appendix (Page 18).

---

> ### Author Response · Authors · 2024-11-24
>
> > Weakness 4: Why do results from all spaces sometimes outperform those from the unified space, while in other cases, the unified space outperforms all spaces? Please analyze this point clearly.
>
> R: Unlike the evaluation strategies using 'Raw space' and 'Coupled space,'  'All spaces' represents a valid and effective evaluation strategy that avoids the two pitfalls highlighted in our paper. This is why we designed and proposed 'All spaces' as a baseline for comparison—it serves as a simple yet effective evaluation approach, yielding good performance in certain settings. As noted in the paper (L404-405 in the initial submission, L407-408 in the revised submission), the choice of evaluation measure critically influences rank consistency, making it a key factor in internal validation. Consequently, 'All spaces' can achieve comparable or even better performance for specific indices in some scenarios. For instance, in Table 2, for DEPICT, the 'Unified space' strategy delivers the better performance across all datasets when using the Calinski-Harabasz index and Silhouette score, while it performs similarly to 'All spaces' in terms of rank consistency with the Davies-Bouldin index.

---

> ### Author Response · Authors · 2024-11-24
>
> > Weakness 5: The t-SNE visualization comparing the unified embedding with the coupled embeddings should be included.
>
> R: All t-SNE visualizations for the coupled embeddings were included in a zip folder uploaded as part of the supplementary material in our initial submission. Additionally, this inclusion is referenced in Line 1287 of the manuscript.

---

> > ### Comment · Reviewer_RgaB · 2024-11-29
> >
> > Thank you for the authors' rebuttal, but my concerns remain unresolved. Firstly, while the goal of establishing a consistent evaluation framework for multiple clustering methods is novel, the motivation is not strongly justified. Furthermore, the methods employed are based on existing approaches, and no new evaluation metric is proposed. Additionally, I am seeking more details about the rank consistency of NMI and ARI, rather than just an explanation of how these metrics are calculated. Finally, regarding Weakness 4, the authors should provide a clear explanation of the underlying reasons behind this phenomenon. So, for authors' response to Weaknesses 5, I'll improve my score. However, I still maintain that the overall quality of this paper falls below the acceptance threshold.

---

### Meta-Review · Area_Chair_r15B · 2024-12-18

**Metareview:**

The paper proposes a unified evaluation framework for different clustering methods based on a specifically learned common space. While the paper is well-motivated, it has several notable weaknesses. First, the proposed approach for learning the embedding is somewhat similar to existing multi-view clustering methods but lacks sufficient discussion to distinguish it, as pointed out by reviewers 8u7M and RgaB. Second, the empirical studies are not thorough. Only JULE (2016) and DEPICT (2017) are used as baselines, while more recent methods such as Contrastive Clustering (Li et al., 2021), SCAN (Van Gansbeke et al., 2020), or SeCu (Qi, 2023) should have been included to better validate the effectiveness of the approach. Additionally, widely-used datasets in deep clustering, such as CIFAR-10 and CIFAR-100, are not used in the experiments. Given the insufficient novelty in the approach and the limited empirical validation, I regret to recommend rejection in its current form.

**Additional Comments On Reviewer Discussion:**

The paper received four borderline rejections from expert reviewers. The main concerns raised by the reviewers revolve around the novelty of the approach, which appears similar to existing multi-view clustering methods, and the lack of sufficient modern clustering methods as baselines. During the rebuttal period, the authors explained their choice of JULE (2016) and DEPICT (2017) as baselines and clarified the differences with multi-view clustering methods. However, most reviewers indicated that their concerns were not fully addressed and maintained a negative stance.

---

### Decision · Program_Chairs · 2025-01-22

Reject